# Targeting SUMOylation promotes cBAF complex stabilization and disruption of the SS18::SSX transcriptome in synovial sarcoma

Synovial Sarcoma (SS) is driven by the SS18::SSX fusion oncoprotein and is ultimately refractory to therapeutic approaches. SS18::SSX alters ATP-dependent chromatin remodeling BAF (mammalian SWI/SNF) complexes, leading to the degradation of canonical (cBAF) complexes and amplified expression of SS18::SSX-containing non-canonical BAF (ncBAF or GBAF) complexes that drive an SS-specific transcription program and tumorigenesis. We demonstrate that SS18::SSX activates the SUMOylation program. The small molecule SUMOylation inhibitor, TAK-981, de-SUMOylates the cBAF/PBAF component, SMARCE1, stabilizing and restoring cBAF on chromatin, shifting SS models away from SS18::SSX-driven transcription. The result is DNA damage, cell death and tumor inhibition across both human and mouse SS tumor models. TAK-981 synergizes with cytotoxic chemotherapy through increased DNA damage, leading to tumor regression. Targeting the SUMOylation pathway in SS restores cBAF complexes and blocks the SS18::SSX transcriptome, identifying an unappreciated role of SUMOylation in SS and a subsequent therapeutic vulnerability.

Synovial sarcoma (SS) is an aggressive soft tissue sarcoma (STS) that occurs frequently in children and young adults. Metastatic SS remains incurable with little benefit observed even with newer combination chemotherapies[1]. Given the lack of durable responses to chemotherapies, development of other strategies and targeted therapies for SS has been a priority.

The pathognomonic chromosomal rearrangement in SS fuses the DNA segment coding for the C-terminal 78 amino acids of SSX1, SSX2, or SSX4, all located on the X Chromosome, to the region coding for amino acids 1–379 of the *SS18* gene on chromosome 18, to generate *SS18::SSX1*, *SS18::SSX2*, or *SS18::SSX4* fusions, each capable of driving SS[2]. Genetic inhibition of SS18::SSX results in growth arrest and cell death in SS cells in vitro[3] and inhibition of tumor growth in vivo[4]. The SS18::SSX fusion oncoprotein is dominant and there appears to be infrequent secondary genetic events in SS[5]. The known reliance of SS on SS18::SSX to drive and maintain tumorigenicity makes SS18::SSX an appealing therapeutic target; however, direct small-molecule inhibitors or degraders for SS18::SSX have yet to be developed[5].

SS18::SSX includes no typical DNA-binding domain[6] but does act as a transcriptional modifier[7]. WT SSX is thought to function in the repression of gene transcription, likely through polycomb repressive complex interactions 1 and 2, but its function has not been deeply explored[8]. WT SS18 is a dedicated member of the chromatin remodeling mammalian SWI/SNF (mSWI/SNF, also called BAF) complex[5,7]. In humans, there are several BAF complex subtypes, each consisting of 13 or more subunits of both shared and unique members[9]. The major BAF complexes in most cells include canonical BAF (cBAF, also referred to as BAF), polybromo-associated BAF (PBAF), and non-canonical or GLTSCR1-containing BAF (ncBAF or GBAF). Each of these three complexes impacts gene transcription uniquely and, depending on the context, can both positively and negatively regulate gene expression. In recent years, it has become apparent that the oncogenicity of SS18::SSX can largely be attributed to the aberrant interaction of SS18::SSX with BAF complexes[5,10–12].

Like WT SS18, SS18::SSX can stably incorporate into BAF complexes; when this happens, it competes with and leads to the

✉e-mail: kevin.jones@hci.utah.edu; senthil.radhakrishnan@vcuhealth.org; acfaber@vcu.edu

degradation of SS18[13]. In addition, we have recently demonstrated that SS18::SSX leads to whole complex cBAF degradation, the result of which is the increased relative prevalence of other BAF-family subtypes, in particular ncBAF with the fusion, altering the cellular composition from a dominant cBAF phenotype to a mix of ncBAF, PBAF and cBAF[11]. Furthermore, overexpression of SMARCB1 is sufficient to restore cBAF genome-wide occupancy and to block the growth of SS cells[14], while disruption of the ncBAF-only components BRD9 or GLTSCR1 abrogates ncBAF function, which also results in blocking the growth of SS cells[10,12]. Thus, SS is defined by a rebalancing of BAF complexes on chromatin leading to an ncBAF dominant transcriptome and reversing this rebalance by restoration of cBAF complexes and/or depletion of ncBAF complexes is toxic to SS.

SUMOylation is a highly conserved eukaryotic post-translational modification (PTM) of proteins that results in the covalent attachment of SUMO (small ubiquitin-like modifier) to lysine residues of the target protein. SUMOylation functions in a wide array of biological processes. These include chromatin modeling, transcription, repair of DNA, cell death, and metabolism[15–18]. Of note, there is very little known about the role of SUMOylation in SS[18].

Subasumstat (TAK-981) is a first-in-class SUMOylation inhibitor that disrupts SAE (SUMO-Activating Enzyme) function by covalently binding to SUMO, resulting in the formation of an adduct[19]. TAK-981 has advanced through phase I trials in patients with late-stage cancers where a recommended phase 2 dose (RP2D) was established (NCT03648372) and also was investigated in combination with pembrolizumab (NCT04381650) in metastatic solid tumors. Thus, TAK-981 allows for clinical evaluation of SUMOylation inhibition as a tractable pathway in cancer.

In the current study, we aim to develop a new treatment for SS based on exceptional response to gene perturbations. This approach leads to the discovery that SS models are significantly sensitive to genetic targeting of the SUMOylation pathway, or with the small molecule SUMOylation inhibitor, subasumstat. Mechanistically, SS18::SSX upregulates several SUMO pathway genes and subasumstat leads to stabilization of the cBAF complex on chromatin and a shift away from the SS18:SSX-driven transcriptome, inducing DNA damage and death. Our findings support the investigation of subasumstat for SS.

## Results

### Synovial sarcoma is highly sensitive to disruption of the SUMOylation pathway

To explore the reliance of SS on SUMOylation for survival, we interrogated the Broad Institute depository (DepMap) of genome-wide RNAi screens across more than 800 cancer cell lines. Strikingly, SS was the most sensitive subtype of cancer to knocking down *UBE2I* (*UBC9*), *SUMO2* (the most widely expressed SUMO in humans[20]), *PIAS1*, *SAE1 and UBA2* (*SAE2*) (Supplementary Fig. 1A–E), molecules encompassing the entire SUMOylation process (Supplementary Fig. 1F) and pointing to a selective dependency of SS on this posttranslational modification (PTM). Indeed, analysis of the effective RNAi in the screen against a 12 gene "SUMO signature" created from the Biocarta SUMO pathway[21], corroborated pathway sensitivity (Fig. 1A). Next, we determined whether the expression of SUMOylation pathways proteins in SS depends on the presence of SS18::SSX. For this, we parsed data from a recent study[5] and evaluated the mRNA expression of key members of the SUMOylation cycle (Supplementary Fig. 1F), following knockdown of short hairpins against the fusion oncogene (sh*SSX*). Essential components of the SUMOylation pathway were significantly reduced after silencing *SS18::SSX* (Fig. 1B), demonstrating a role of SS18::SSX to increase SUMOylation genes in SS. We corroborated these data by silencing *SS18::SSX* in HS-SY-II cells and the ex vivo cell line, SS.PDX, and noting SUMO modified proteins decreased in both cell lines. As expected, these cell lines also underwent cell death (as evidenced by

cleaved PARP) following depletion of SS18::SSX (Fig. 1C). Additionally, markedly reduced SUMOylation was demonstrated in a third SS cell line, Yamato, following inducible knock down of SS18::SSX (Supplementary Fig. 2A). A decrease in the abundance of SS18::SSX after treatment with TAK-981 was also noted (Supplementary Fig. 2A), likely a result of a feedback loop attempting to restore cellular SUMOylation levels. To investigate whether the fusion protein directly induces the expression of the SUMOylation pathway genes, we performed an SS18::SSX ChIP-seq assay in HS-SY-II cells, either untreated or treated with TAK-981. Analysis of these data demonstrated that SS18::SSX binds to the promoter of the key SUMOylation genes *SUMO2*, *SUMO3*, *PIAS1*, and *UBE2I* (encoding the UBC9 protein) (Fig. 1D). Altogether, these data demonstrate SS18::SSX drives the SUMOylation program in SS.

To verify the pathway dependency on SUMOylation in SS, we pharmacologically blocked global SUMO modifications, utilizing the novel SUMOylation inhibitor, TAK-981 (subasumstat)[19]. Cell viabilty of the SS cell lines tested was robustly reduced after treatment with increasing concentrations of TAK-981, as demonstrated by CellTiter-Glo and crystal violet survival assays (Fig. 1E and Supplementary Fig. 2B). To further corroborate the dependence of SS on SUMOylation, we depleted characteristic components of the SUMO pathway using sgRNAs in the HS-SY-II cells. As a simple and reliable means to detect a potential effect by CRISPR/CAS9 perturbation, we evaluated cells by crystal violet staining[22–25]. Genetically targeting of SUMOylation resulted in nearly complete death of the SS cells (Fig. 1F). To determine whether the toxicity of TAK-981 in SS was at least partially due to presence of SS18::SSX, we transfected two SS cell lines, HS-SY-II and the SS.PDX ex vivo cells, with siRNA targeting SS18::SSX. We then incubated the cells with increasing concentrations of TAK-981 and performed CellTiter-Glo viability assays[26–29]. A significant rescue of TAK-981 induced toxicity by SS18::SSX depletion was noted in both cell lines (Fig. 1G). Enhanced cleavage of PARP was detected in TAK-981 treated HS-SY-II, SS.PDX and SYO-1 cells (Fig. 1H), evidencing cell death. Of note, downregulation of the well-established histological marker of SS, the prosurvival BCL2 protein[30–32], was also observed following TAK-981 treatment, without changes in other BCL2 family member proteins (Fig. 1H and Supplementary Fig. 2C). Interestingly, we have shown BCL2 is upregulated in the presence of SS18::SSX in human and mouse SS models[33], which suggested to us that TAK-981 may impact the SS18::SSX-driven transcriptome.

### TAK-981 de-SUMOylates SMARCE1, an integral subunit of the cBAF and PBAF complex

We next sought to gain mechanistic insights into why blocking SUMOylation in SS is so effective. To identify the members of SS proteome that were de-SUMOylated upon TAK-981 treatment, a proteome-wide detection of the SUMOylation sites[34] in HS-SY-II SS cells before and after TAK-981 treatment (Fig. 2A) was performed. The assay utilizes a wild-type α-lytic protease (WaLP) to digest SUMOylated proteins resulting in the production of peptides carrying SUMO-remnant diglycyl-lysine (KGG) at the site of SUMO modification (performed by Cell Signaling technology). Using specific antibodies for the isolation of the KGG-containing peptides and followed by mass spectrometry for sequencing of the captured peptides, we identified 61 SUMO-modified proteins in the control HS-SY-II cells that were reduced (greater than two-fold) by TAK-981 (Supplementary Data 1). Among these was the cBAF/PBAF complex member, SMARCE1. SMARCE1 constitutes a fundamental component of the BAF core module along with SMARCC1, SMARCC2, SMARCD1/2/3 and SMARCB1[35]. SMARCE1 incorporates into cBAF and PBAF, but not ncBAF[35]. Importantly, it has been recently demonstrated that in clear cell meningioma (CCM), a cancer driven by loss of SMARCE1, the cBAF complex fails to stabilize on chromatin, attenuating its activity[36], underlying the role of SMARCE1 in sustaining the integrity of the cBAF

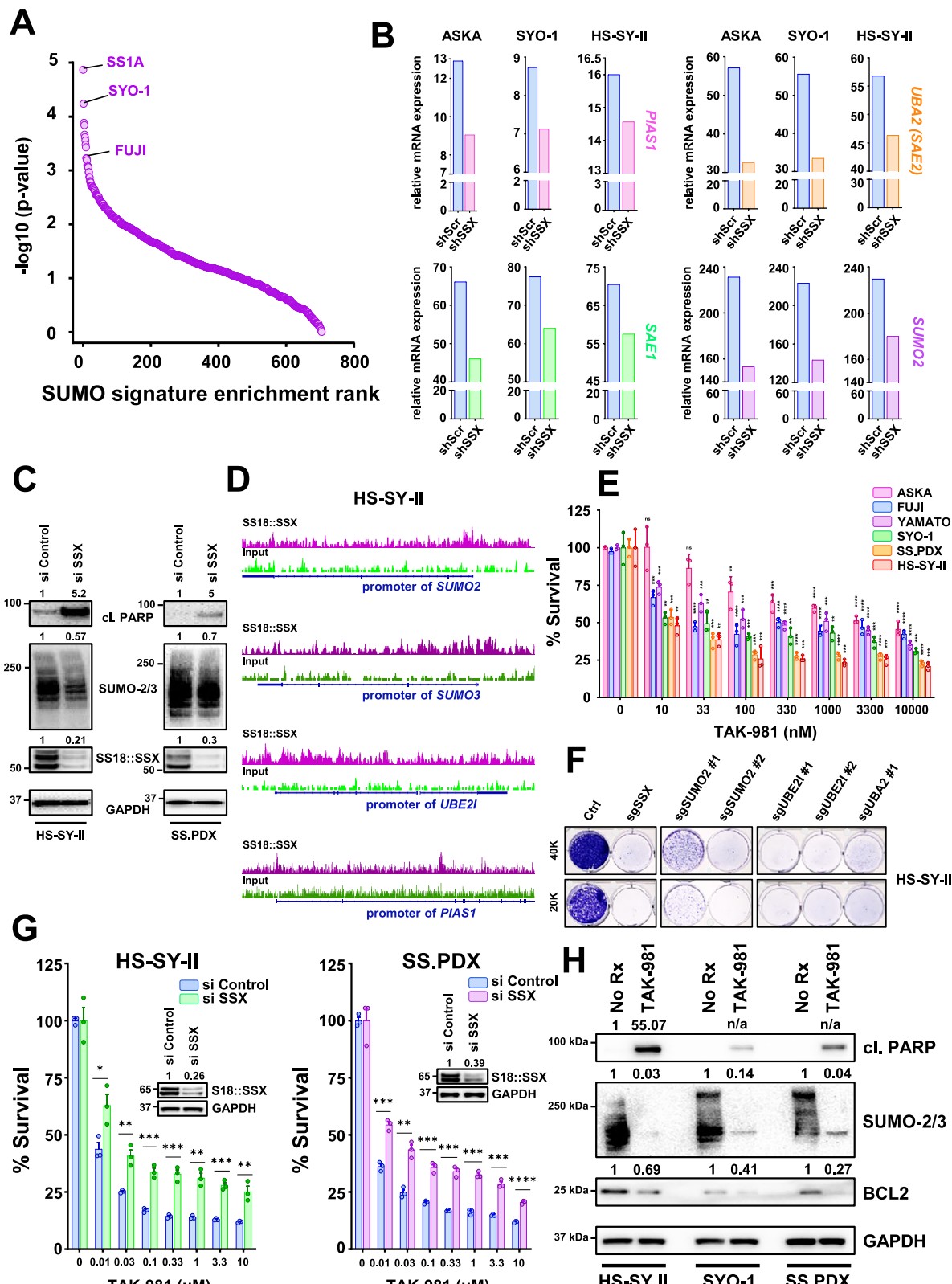

complex, by driving cBAF complex stabilization on chromatin through cross-link interactions with other cBAF components, like ARID1A, but not PBAF subunits[36]. To validate the mass spectrometry data, we performed pull-down assays following transient overexpression of SUMO2 in the transfection amenable HS-SY-II and Yamato SS cells. We immunoprecipitated whole cell lysates with either SUMO-2/3 ab (Fig. 2B, D), or SMARCE1 ab (Fig. 2C, E) and investigated the

SUMOylation status of various BAF components. In both cell lines, SUMOylated SMARCE1 (Fig. 2B, C and Fig. 2D, E), and SUMOylated BAF47 (Supplementary Fig. 2D and Supplementary Fig. 2E) were identified (upper band) along with unmodified SMARCE1, ARID1A and SMARCB1 (BAF47) binding to SUMOylated proteins or free SUMO molecules (lower band) likely via their SUMO interacting motifs (SIMs) (Fig. 2B, D and Supplementary Fig. 2D, Supplementary Fig. 2E). The

**Fig. 1 | Synovial sarcoma is sensitive to disruption of the SUMOylation pathway.**
**A** Each cell type from the DepMap database was analyzed for enrichment in the custom SUMO signature ("SAE1", "SUMO1", "SUMO2", "SUMO3", "UBA2", "UBE2I", "RANBP2", "CBX4", "PIAS1", "PIAS2", "PIAS3", "PIAS4"). Cell type-specific genes were ranked by the DEMETER2 v5 dependency scores ("D2_combined_gene_dep_scores.csv" file downloaded 09/18/2019) and analyzed using GSEA (clusterProfiler v.3.14.3 R package [PMID: 22455463]). The enrichment $p$ values were $-\log 10$ transformed and used to rank the cells and plot the transformed $p$ values on Y-axis. **B** RNA levels of genes involved in SUMOylation following sh*SSX* knockdown were obtained from an RNA-seq analysis performed in SS cell lines[5] **C** SS cells were transfected with si*SSX* for 36 h and whole cell lysates were probed with the indicated antibodies. **D** ChIP-seq enrichment tracks of SS18::SSX in untreated (No Rx = no drug) HS-SY-II cells compared to the tracks of the "input" at the loci of the indicated SUMO components. The combined depth-normalized signal was visualized using IGV browser[103] and ensuring the same signal range (Y-axis) for each region of interest. **E** SS cells were treated with increasing concentrations of TAK-981, and cell viability was assessed 72 h later. **F** HS-SY-II cells underwent CRISPR/Cas9-mediated targeting of the indicated SUMOylation genes and cells were counted and plated at low density and stained with crystal violet 12 days later. **G** SS cells were transfected with siRNA directed against *SSX* for 36 h, reseeded, and treated with increasing concentrations of TAK-981 for 72 h before cell viability was assessed; inset is western blotting confirming knockdown. Protein band intensities were quantified like in (**C**). **H** SS cells were treated with 100 nM TAK-981 for 36 h or left untreated (No Rx) and whole cell lysates were probed with the indicated antibodies. **H** shares the same GAPDH blot with Supplementary Fig. 7C. For (**E**, **G**) $n = 3$ biological replicates and data are presented as mean values + SD. Unpaired two-tailed $t$ tests were performed for comparisons between each TAK-981 treatment and No Rx for each cell line in (**E**) and for comparisons between si*Control* and si*SSX* in (**G**). Exact $p$ values and source data can be found in the Source Data.

model of SUMO molecules binding to potential SIMs in the hydrophobic groove of the specific BAF complex proteins[37] is illustrated in Supplementary Fig. 2F. These results are consistent with the SMARCE1 SUMOylation detected in native conditions (Fig. 2A and Supplementary Data 1) and indicate that other BAF complex proteins in addition to SMARCE1 may constitute SUMO substrates in SS. Importantly, significant upregulation of all three cBAF members (ARID1A, SMARCE1 and BAF47) following TAK-981 treatment was noted in the input lysates (nuclear lysates prior to immunoprecipitation) (Fig. 2B–E). In addition, the interaction between SMARCE1 and both BAF47 and ARID1A was enhanced, but not the interaction between SMARCE1 and the unique PBAF subunit, PBRM1 (Fig. 2C, E). This is consistent with the study by St Pierre et al.[36], demonstrating SMARCE1 can stabilize cBAF complexes, but not PBAF complexes.

Consistent with the literature that the attachment of SUMO moieties can prime the target proteins for ubiquitination by E3 ubiquitin ligases[38,39] and subsequent proteasome degradation, we studied the potential integration of STUbLs (SUMO-targeted Ubiquitin Ligases) to ubiquitinate SUMO-SMARCE1. Specifically, there has been a reported collaborative ubiquitination of SUMOylated substrates by the STUbLs RNF4 and TOPORS[39]. RNF4 is a well-established STUbL that plays a central role in protein regulation bridging the crosstalk between SUMOylation and ubiquitination[40–42]. TOPORS is a recently identified SUMO1-selective STUbL that complements RNF4 in creating ubiquitin landscapes across SUMOylated targets[39]. We found that silencing of RNF4 via siRNA was sufficient to increase SMARCE1 levels at baseline conditions in HS-SY-II and Yamato cells (Fig. 2F–I), demonstrating its role as a STUbL for SMARCE1 in SS. In the YAMATO cells, co-silencing of both *RNF4* and *TOPORS* with siRNA led to a higher level of accumulation of the SUMO-conjugated SMARCE1 protein than RNF4 silencing alone (Fig. 2H). At baseline conditions in the HS-SY-II and YAMATO cells, depletion of RNF4 resulted in the accumulation of a band with a size consistent with the SUMO-conjugated form of SMARCE1, likely representing species of SMARCE1 that could no longer be degraded by the proteasome due to the absence of RNF4. As expected, this band became undetectable in the presence of TAK-981 (Fig. 2F, H). These data altogether demonstrate that RNF4 degrades SUMOylated SMARCE1 in SS cells, with an additional, albeit less pronounced, contribution from TOPORS. Next, we sought to confirm that RNF4/TOPORS-mediated SMARCE1 stabilization is a significant contributor to TAK-981-mediated toxicity in SS. Strikingly, when we interrogated the Broad Institute depository (DepMap) of genome-wide RNAi screens, we noted SS is the most sensitive cancer to RNF4 silencing among 30 subtypes of cancer (Fig. 2J), consistent with the effects of silencing SUMO components in SS (Fig. 1F). To confirm these data, we next silenced *RNF4/TOPORS* by siRNA and examined SS viability. *RNF4/TOPORS* silencing resulted in toxicity of the HS-SY-II cells (Fig. 2K left), with a more marked toxicity in the Yamato and SYO-1 cells, in line with our hypothesis (Fig. 2K right and Supplementary

Fig. 2G). The mechanism of TAK-981 toxicity is graphically depicted in Fig. 2L.

## SUMOylation inhibition stabilizes the cBAF complex
We next interrogated four different SS cell lines (HS-SY-II, Yamato, SYO-1 and FUJI) and conducted a time course of TAK-981 treatment, to comprehensively determine the changes in the protein as well as the mRNA expression levels of several main cBAF members. The results show a significant, enhanced expression of the cBAF proteins ARID1A, SMARCE1 and BAF47 over time, where no increase was seen at the mRNA level (Fig. 3A–C). No consistent changes were observed either in the PBAF unique subunit, PBRM1, or in the ncBAF unique subunit, BRD9, or that of the fusion protein SS18::SSX (Fig. 3A, B). This data is in line with the hypothesis that TAK-981 leads to a selective cBAF complex stabilization.

We next sought to determine the potential alterations in the size of the BAF complexes following SUMOylation inhibition, by conducting density sedimentation studies. Interrogation of glycerol gradients of the nuclear extracts revealed a marked upshift of the cBAF members SMARCE1, ARID1A and BAF47 by ~3 fractions after TAK-981 treatment in SYO-1 cells (Fig. 3D, F and Supplementary Data 2). These data are consistent with the expected gain in cBAF complex mass caused by TAK-981, leading to incorporation of greater amounts of SMARCE1, ARID1A and BAF47 proteins. Both inhibiting ubiquitination with TAK-243[43,44] and blocking the proteasome activity with bortezomib[45] resulted in consistent expression changes in the three cBAF complex members, resembling the changes detected following TAK-981 treatment (Fig. 3D, F and Supplementary Data 2). These data are consistent with the hypothesis that SUMO-conjugated cBAF proteins undergo ubiquitination and subsequent proteasomal degradation leading to degradation of cBAF complexes. In addition, TAK-981 induced a modest, yet clear, increase of SS18::SSX abundance in the cBAF corresponding gradients (Fig. 3D and Supplementary Fig. 2H), in line with cBAF stabilization. In contrast, no significant changes were observed in the PBRM1 (PBAF) or BRD9 (ncBAF) gradients following TAK-981 treatment (Fig. 3D and Supplementary Fig. 2H).

We have previously demonstrated that SS18X::SSX reduces the integrity of the cBAF complex[11]. Here, we ran glycerol gradient gels in the HS-SY-II cells following knock down of the SS18X::SSX fusion protein, in the presence and absence of TAK-981 treatment. In the control cells (shREN knockdown), we noted a significant accumulation of ARID1A, SMARCE1 and BAF47 in the cBAF corresponding fractions following TAK-981 treatment. This is consistent with cBAF stabilization following SUMOylation inhibition (Fig. 3E, G and Supplementary Data 3). Expectedly, shSSX knockdown[11] cells demonstrated enhanced expression of these cBAF proteins compared to control shREN knock down cells (Fig. 3E, G and Supplementary Data 3). As in the SYO-1 cells, TAK-981 treatment in the shREN knockdown control HS-SY-II cells was also sufficient to enhance expression of these cBAF members in the

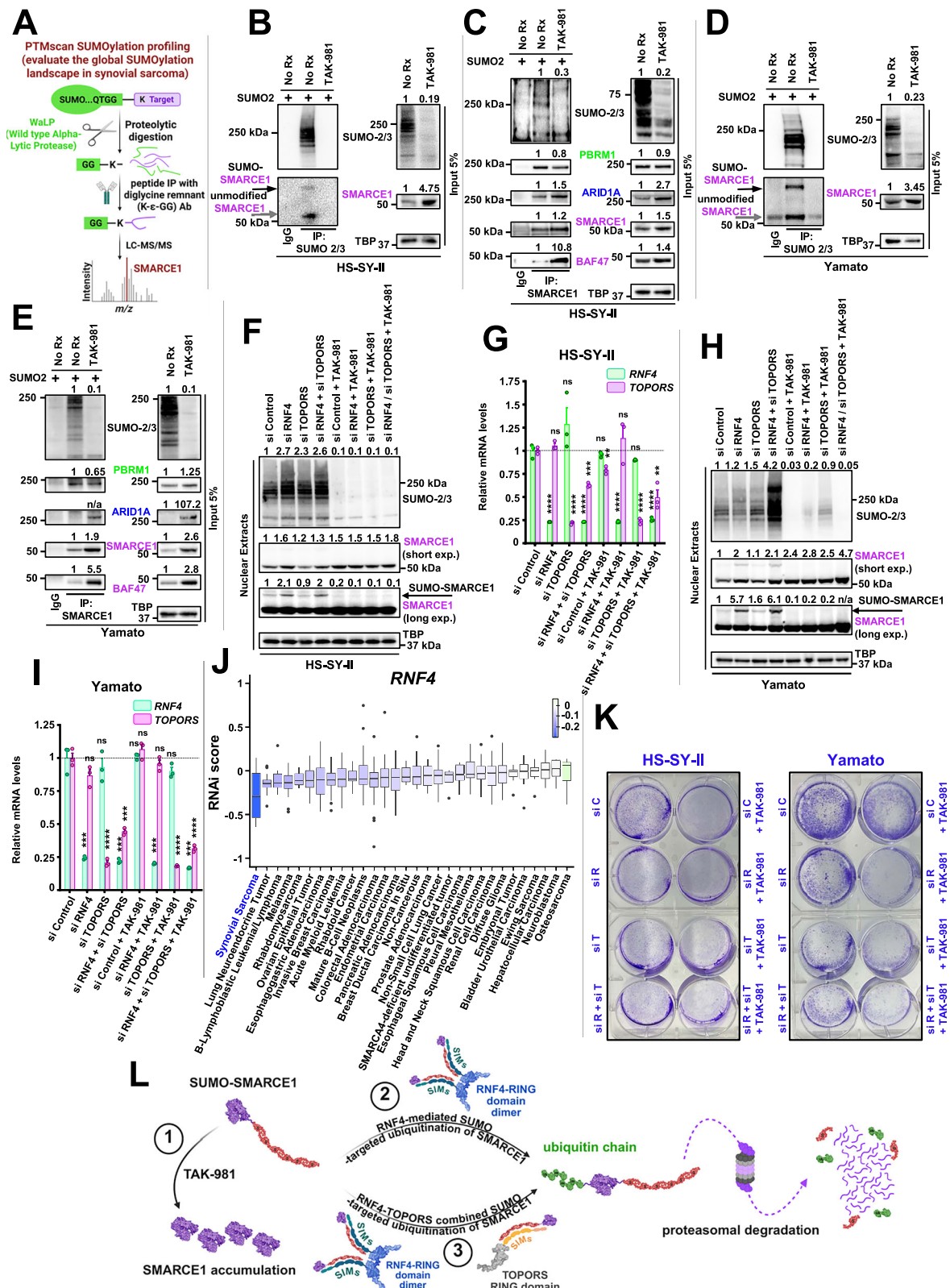

cBAF corresponding fractions (Fig. 3). In contrast to the changes seen with cBAF complex members following TAK-981 treatment, there was a slight reduction in the stability of ncBAF complexes and no significant changes in PBAF complexes, as evidenced by the elution patterns of BRD9 and PBRM1, respectively (Fig. 3E and Supplementary Fig. 2H). The TAK-981-induced changes in cBAF complexes are summarized in Fig. 3H.

While these data demonstrated that TAK-981 enhanced cBAF protein complexes, which was also seen following knockdown of SS18::SSX (Fig. 3D, E), the assays did not speak to any changes of these complexes at chromatin. We therefore measured relative chromatin affinities in the absence and presence of TAK-981 treatment, performing differential salt extraction in SYO-1, HS-SY-II and Yamato cells. The higher affinity of the cBAF complex for chromatin after rescue of

**Fig. 2 | TAK-981 de-SUMOylates SMARCE1, preventing its proteasomal degradation. A** Illustration depicting PTMscan SUMOylation profiling. **B** SUMO-2/3 complexes from untreated (No Rx) or TAK-981 treated (100 nM, 36 h) HS-SY-II cells were immunoprecipitated following transfection with exogenous SUMO2 and 36 h treatment with 100 nM of TAK-981 (or untreated cells; No Rx). Black arrow indicates SUMOylated SMARCE1. Gray arrow indicates unmodified SMARCE1 (see also Supplementary Fig. 2E). **B** shares the same western blots for SUMO-2/3 and TBP with Supplementary Fig. 2D. **C** SMARCE1 complexes were immunoprecipitated from HS-SY-II cells, following the same treatment as in (**B**). **D** Same as in (**B**), for Yamato cells. **D** shares the same western blots for SUMO-2/3 and TBP with Supplementary Fig. 2E. **E** Same as in (**C**), for Yamato cells. **F** HS-SY-II cells were transfected with si*Control*, si*RNF4*, si*TOPORS* or the si*RNF4*/si*TOPORS*, followed by treatment with 100 nM of TAK-981 for 36 h (or no treatment; No Rx) and nuclear lysates were probed with the indicated antibodies. **G** HS-SY-II cells from F) were analyzed by qPCR for *RNF4* and *TOPORS* abundance. Data are values relative to the control sample and normalized to *ACTB*; n = 3 biological replicates; data are values + SEM. Unpaired two-tailed *t* tests comparing each sample with si*Control*. **H** Same as (**F**), for Yamato cells. **I** same as (**G**) for Yamato cells. **J** DepMap consortium RNAi screen data for shRNAs targeting *RNF4* among 666 cancer cell lines. The RNAi score quantifies the impact of knocking down of a gene on the viability of a cell line, with more negative scores indicating a stronger dependency on the targeted gene. The boxplots indicate median and interquartile range. The whiskers extend to the minimum and to the maximum values excluding outliers. **K** HS-SY-II (left) and Yamato cells (right) were transfected and treated as indicated and stained with crystal violet (siR: si*RNF4*; siC: si*Control*; siT: si*TOPORS*). **L** Model of SUMO-SMARCE1 protein processing. (1) TAK-981 blocks SUMOylation and protects SMARCE1 from proteasomal degradation. (2) RNF4 tags ubiquitinated SUMO-SMARCE1. (3) In some SS cell lines, TOPORS together with RNF4 induces greater SUMO-SMARCE1 ubiquitination and proteasomal degradation. The illustrations were created in BioRender. Floros, K. (2025). Exact *p* values and source data can be found in the Source Data.

SMARCB1/BAF47 has been delineated in the past[12]. Blocking SUMOylation resulted in cBAF complex dissociation from chromatin at higher salt concentrations (300–500 mM NaCl) than prior to drug treatment, as evidenced by ARID1A, BAF47 and SMARCE1 expression levels (Fig. 4A–D). In contrast, the elution of BRD9 was noted at lower NaCl concentrations following TAK-981 treatment in SYO-1 and Yamato cells and at slightly higher NaCl concentrations in HS-SY-II cells (Fig. 4A and Supplementary Fig. 2I). Likewise, ARID2 (and consequently the PBAF complex) does not follow a consistent pattern of chromatin affinity across the three cell lines (Fig. 4A and Supplementary Fig. 2I). These results demonstrate stronger binding of the cBAF complex to chromatin following TAK-981 treatment in SS, not seen with the other BAF complexes (Fig. 4E).

## Blocking SUMOylation leads to disruption of the synovial sarcoma signature and to induction of mesenchymal differentiation

Results of a previous study by Banito et al.[22] demonstrated that SS18::SSX binds to and activates the expression of KDM2B-PRC1 target genes. KDM2B is a histone demethylase and component of a non-canonical polycomb repressive complex 1 (PRC1.1), that has been reported by Banito et al. to function as a mediator for the oncogenic activity of SS18::SSX1, contributing to synovial sarcomagenesis and the SS signature. This signature primarily contains genes that encode homeobox transcription factors related to neurogenesis and other developmental processes (Supplementary Fig. 2B). SS18::SSX knockdown reduces this SS signature and rewires the cells towards a mesenchymal phenotype, including the expression of genes encoding extracellular matrix (ECM) proteins and secreted proteins that are highly expressed in human fibroblasts[22]. We evaluated whether the addition of TAK-981 can efficiently impede the activation of the SS18::SSX transcriptome and hence disrupt the SS signature, while inducing ECM-related genes, in total resembling the changes observed after silencing the fusion protein. Due to the heterogeneity of the gene targets of SS18::SSX in SS, we performed RNA-seq in three SS cell lines (SYO-1, the ex vivo SS.PDX, and HS-SY-II), before and after treatment with TAK-981. Strikingly, the up- and down-regulated pathways, based on the expression changes of the related genes, verified our hypothesis as several of the most enhanced or suppressed biological processes are common to both TAK-981 treated and *SS18::SSX* genetically inhibited SS cells (Fig. 5A). In addition, there is a significant overlap of the altered genes between TAK-981 treated cell lines (Supplementary Fig. 3A and Supplementary Fig. 3B), which was verified also after pairwise comparisons (Supplementary Fig. 3C, Supplementary Fig. 3D and Supplementary Fig. 3E). Distribution of the commonly altered genes according to their fold change revealed a positive correlation when interrogating the cell lines in pairwise fashion (Supplementary Fig. 3F, Supplementary Fig. 3G, Supplementary Fig. 3H), verifying their

uniform response to the inhibition of SUMOylation in SS. Consistent with the pathway analysis (Fig. 5A), the overlap between the genes up- and down-regulated after knocking down *SS18::SSX* in HS-SY-II cells[22], and the genes enhanced or suppressed after the addition of TAK-981 to SYO-1 and SS.PDX cells was significant (Fig. 5B, C). Noteworthy, this correlation was even more significant when comparing the SS18::SSX-genetically inhibited HS-SY-II cells and TAK-981-treated HS-SY-II cells (Fig. 5B). Consistently, a significant portion of the 1000 most up- or down-regulated genes after TAK-981 treatment are also up- or down-regulated after silencing *SS18::SSX* in all three SS cell lines (Fig. 5D). Furthermore, addition of TAK-981 promotes silencing of developmental genes and augments the expression of genes related to ECM, cell adhesion and muscle function (Supplementary Fig. 3I and Supplementary Fig. 3K), similar to the changes observed after *SS18::SSX* silencing, and in line with the gene ontology conducted in Fig. 5A. The above data were further corroborated by qPCR (Supplementary Fig. 3J, Supplementary Fig. 3L and Supplementary Data 4) for most of the gene changes identified by RNA-seq.

## The binding of BAF complexes to chromatin is responsible for many of the observed expression changes of the SS transcriptome after blocking SUMOylation

To further investigate how the different BAF complexes contribute to the observed transcription phenotype following suppression of SUMOylation, we performed Chromatin Immunoprecipitation Sequencing (ChIP-seq) using antibodies raised against SMARCA4 (the ATPase found in all three BAF complexes) in HS-SY-II cells, and antibodies raised against SS18::SSX, KDM2B, and H3K27Ac in both HS-SY-II and SYO-1 cells. Gene ontology analysis of the reduced and the enhanced SMARCA4 ChIP-seq signals after TAK-981 treatment in HS-SY-II cells indicated transcriptome changes of genes intimately involved in neurogenesis, DNA replication and cell cycle (for the reduced signals) and genes involved in the ECM and muscle function (for the enhanced signals) (Fig. 6A, D), indicating a correlation between our RNA-seq data (Fig. 5) and the binding profiles of all three BAF complexes.

Moreover, more than half of the genes with a decreased SMARCA4 ChIP-seq signal (155 out of 288) also displayed a reduced SS18::SSX ChIP-seq signal (Fig. 6B, F and Supplementary Fig. 4C). Additionally, 49 out of the 155 genes that had enhanced SMARCA4 signals also had enhanced SS18::SSX binding (Fig. 6E, F, Supplementary Fig. 4E). Furthermore, 15 genes that demonstrated significant downregulation of SS18::SSX binding (Fig. 6B), and 24 genes with significant downregulation of SMARCA4 binding (Fig. 6B, C), were included among the 100 most downregulated genes following *SS18::SSX* knock down from Banito et al.[22]. While we did not see significant overlap between increased SMARCA4 or SS18::SSX binding at the genes that were upregulated after silencing SS18::SSX[22], binding of SMARCA4 at

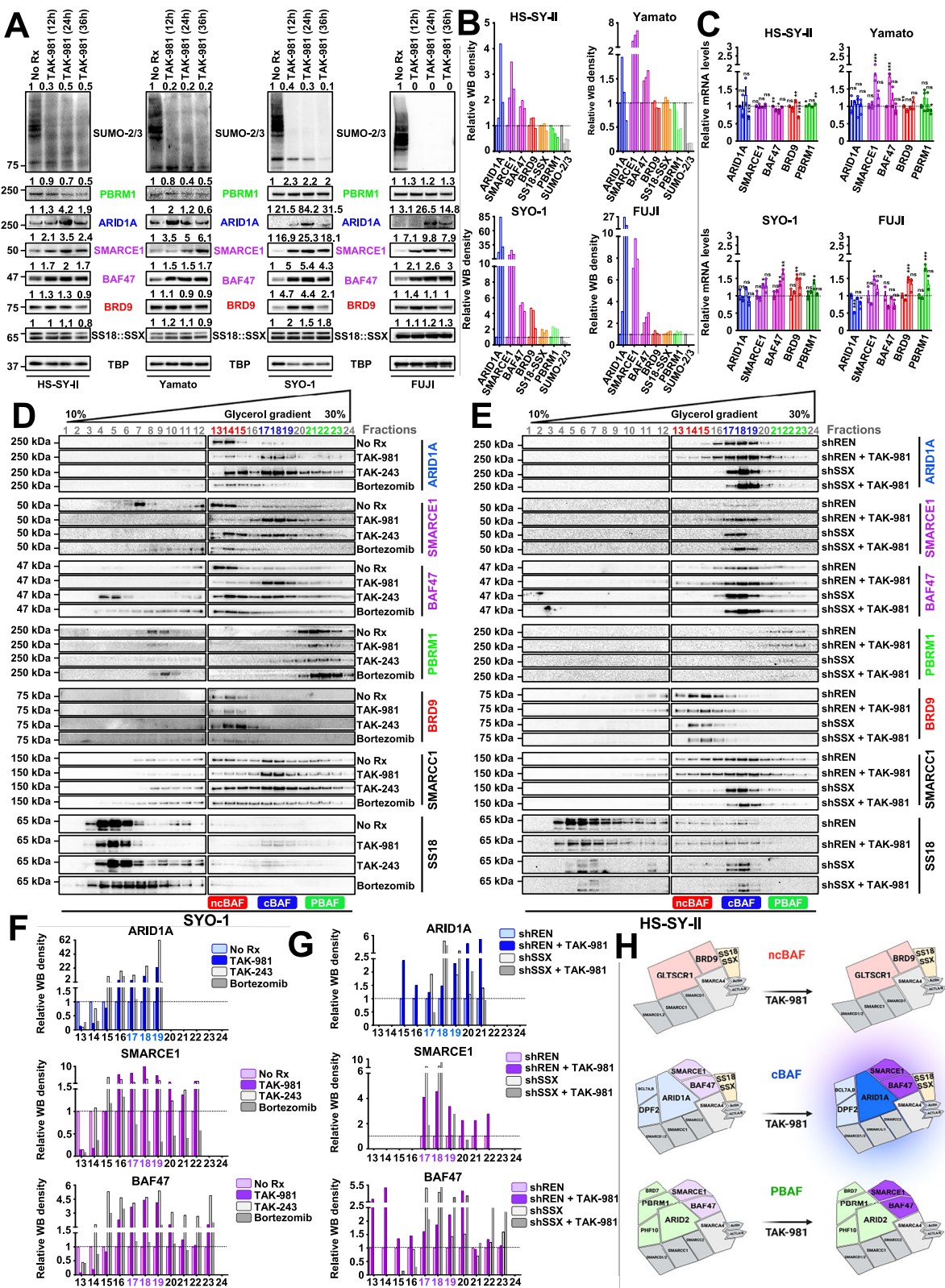

several genes related to the mesenchymal phenotype that were upregulated following SS18::SSX silencing from Banito et al.[22] was noted (Fig. 6G).

Next, in the HS-SY-II cells, we examined the overlap between the up- and down-regulated genes following TAK-981 treatment that were identified by the RNA-seq analysis of our study, with the SMARCA4/SS18::SSX binding profiles, and again found significant overlap

(Supplementary Fig. 4A and Supplementary Fig. 4D). This is consistent with the overlap observed with the gene expression changes following *SS18::SSX* knock down (study by Banito et al.[22]). Additionally, a significant portion of the genes that demonstrated reduced KDM2B binding demonstrated reduced SMARCA4/SS18::SSX binding after TAK-981 treatment (Supplementary Fig. 4A). We then turned our attention to a 30-gene signature that was developed by examining

**Fig. 3 | SUMOylation inhibition stabilizes the cBAF complex. A** Different SS cell lines (HS-SY-II, Yamato, SYO-1 and FUJI) underwent a time-course of treatment with 100 nM of TAK-981 for 0, 12, 24, and 36 h and nuclear lysates were prepared and the expression of the indicated proteins was detected by western blotting. **B** The quantified signal from (**A**) was graphed using the GraphPad Prism software. A line was added to the No Rx levels ("1") to simplify visual comparisons. **C** RNA extracts from the same cells used in (**A**) were analyzed by qPCR and the abundance of the mRNA levels of the indicated BAF complex subunits was quantified (*n* = 3 biological replicates; data are presented as mean values + SEM). Unpaired two-tailed *t* tests were performed for comparisons between No Rx and each time point for each BAF complex member separately. **D** SYO-1 cells were untreated (No Rx) or treated with 100 nM of TAK-981 for 36 h, 1 µM of TAK-243 for 3 h or 100 nM of bortezomib for 3 h and density sedimentation assay (10–30% glycerol gradient) followed by western blotting for each of the indicated BAF family components was performed on nuclear extracts. **E** HS-SY-II cells were infected with control shRNA ("shREN") or sh*SS18::SSX* ("shSSX") and treated for 36 h with 100 nM of TAK-981, or left untreated. Density sedimentation assay (10–30% glycerol gradient) followed by the indicated BAF family components immunoblotting was performed on nuclear extracts. **F** Protein band intensities from D) were quantified using GeneTools software with background subtraction and normalized for each fraction to the value of No Rx for each BAF complex subunit separately (please see also Supplementary Data 2). A line was added to the No Rx levels ("1") to simplify visual comparisons. **G** Protein band intensities from (**E**) were quantified using GeneTools software as previously (please see also Supplementary Data 3). A line was added to the shREN levels ("1") to simplify visual comparisons. **H** Illustration depicting the upregulation of SMARCE1, ARID1A and BAF47 following TAK-981 treatment and the stabilization specifically of cBAF complexes (created in BioRender. Floros, K. (2025)). Exact *p* values and source data can be found in the Source Data.

genes specifically elevated in SS compared to other sarcoma tumors and that were diminished after silencing the fusion protein in HS-SY-II cells; these genes constitute a core "SS signature"[22]. More than half of these genes had weaker SMARCA4 and/or SS18::SSX binding after TAK-981 treatment (Supplementary Fig. 4A and Supplementary Fig. 4B).

We next evaluated genes with significantly enhanced or diminished SMARCA4 and SS18::SSX ChIP-seq peaks following TAK-981 treatment with that of the H3K27ac ChIP-seq, a marker for active gene enhancers and promoters[46]. Noteworthy, the attenuated ChIP-seq profile of SS18::SSX and of H3K27ac at genes overlaid significantly with each other (Supplementary Fig. 4A), as did the enhanced ChIP-seq profile of SMARCA4 and of H3K27ac (Supplementary Fig. 4D and Supplementary Fig. 4F). We next analyzed ChIP-seq profiles of SS18::SSX, KDM2B, and H3K27ac in SYO-1 cells, in the presence and absence of TAK-981. Similar to the data in HS-SY-II cells, we found significant overlap in the changes in ChIP-seq signals with TAK-981 treatment among these different experiments, as well as with these changes and the changes in gene expression following *SS18::SSX* knockdown in the HS-SY-II cells from Banito et al.[22] or TAK-981 treatment detected by RNA-seq in our study, as well as the SS signature[22] (Supplementary Fig. 5A–K). Altogether, the above data suggests that while reduced ncBAF complex plays a central role in the loss of the SS18::SSX-driven transcriptome[10], the changes noted in other BAF complexes (most noteworthy, increased cBAF binding and activity) contributes to the transcriptome changes following SUMOylation inhibition in SS.

## Addition of TAK-981 results in loss of chromatin accessibility to promoters of the SS18::SSX transcriptome

SS18::SSX inhibition in SS causes a large reduction of chromatin accessibility at SS18::SSX binding sites, associated with rearrangement of BAF complexes across the genome[22]. Based on the data above, we reasoned that inhibition of SUMOylation resulted in mitigated accessibility to the promoters of genes in the SS18::SSX transcriptome. Therefore, we performed ATAC-seq (Assay for Transposase-Accessible Chromatin using sequencing) in HS-SY-II and SYO-1 cells before and after TAK-981 treatment. Subsequently, we focused on the reduced ATAC-seq signals at the genes that were also suppressed in HS-SY-II cells after knocking down *SS18::SSX*, from Banito et al.[22] Strikingly, blocking SUMOylation caused a marked reduction of the accessibility in SYO-1 cells, and a smaller but clear attenuation of the signal in the HS-SY-II cells, at the promoters of the genes reduced in expression following *SS18::SSX* knockdown (Fig. 7A, B). In addition, we observed enhancement at introns and at distal intergenic sites of these genes (Fig. 7A, B, Supplementary Fig. 6A and Supplementary Fig. 6B). The mitigation of the number of peaks and hence of accessibility at the transcription starting sites (TSS) of these genes is demonstrated in Fig. 7C. A similar pattern was detected when conducting the same analysis around peaks of down- and

up-regulated genes after knocking down *SS18::SSX* (Banito et al.[22]). Blocking SUMOylation stimulated a reduction in the accessibility in SYO-1 cells (Supplementary Fig. 6C left), and a less significant mitigation of the signal in the HS-SY-II cells (Supplementary Fig. 6D left). Interestingly, TAK-981 addition did not result in any change in the accessibility of the chromatin for upregulated genes following knockdown of *SS18::SSX* in SYO-1 cells (Supplementary Fig. 6C right), while it strengthened the ATAC-seq signal in the HS-SY-II cells (Supplementary Fig. 6D right). Consistent with our hypothesis and the data displayed in earlier figures, blocking SUMOylation induces a switch from an open to a close chromatin state at promoters of genes that promote synovial sarcomagenesis (Fig. 7D).

It has been previously reported that increased cBAF complex occupancy strongly correlates with enhancer activation[14]. We hypothesized that most of the genes that increase in expression upon de-SUMOylation are targets of cBAF enhancers. That is, TAK-981 treatment increases gene expression that is primarily due to cBAF stabilization. To do so, we investigated the correlation between the 1000 most upregulated genes following TAK-981 treatment (detected by our RNA-seq analysis), and the changes in the accessibility of the chromatin at these specific genes (ATAC-seq analysis) and in their H3K27ac ChIP-seq signals. The 1000 most upregulated genes following de-SUMOylation also demonstrated enhanced chromatin accessibility and H3K27ac levels in both the SYO-1 and HS-SY-II cells (Fig. 7E, F). Expectedly, the same analysis for the 1000 most downregulated genes revealed a less consistent pattern (Supplementary Fig. 6E and Supplementary Fig. 6F). The above findings are illustrated in Supplementary Fig. 6G, while the redistribution of the BAF complexes following TAK-981 treatment and deSUMOylation of SMARCE1 is portrayed in Fig. 7G.

## TAK-981 has activity in synovial sarcoma in vitro and in vivo by inducing DNA damage

We next attempted to elucidate the mechanism that leads to SS toxicity after treatment with TAK-981. Consequently, we conducted gene set enrichment analysis[47] of the ATAC-seq data in the SYO-1 and HS-SY-II TAK-981-treated cells and found that several of the top 20 downregulated pathways involved DNA damage. For example, "DNA repair", "cell cycle", "mismatch repair" and "base excision repair" were suppressed (Supplementary Fig. 7A, Supplementary Fig. 7B). Similar data were obtained from the RNA-seq data from experiments performed in the same cell lines (Fig. 5A). We also confirmed these data at the protein level with a well characterized DNA damage marker (p-H2AX (Ser139), also known as γ-H2AX[48]) (Supplementary Fig. 7C).

Seeking a direct link between the observed rebalancing of the BAF complexes and the induction of DNA damage, we focused on the promoters of genes associated with reduced SS18::SSX (in HS-SY-II and SYO-1 cells) and SMARCA4 (in HS-SY-II cells) ChIP-seq peaks. In these analyses, we identified motifs of several TFs (transcription factors) of

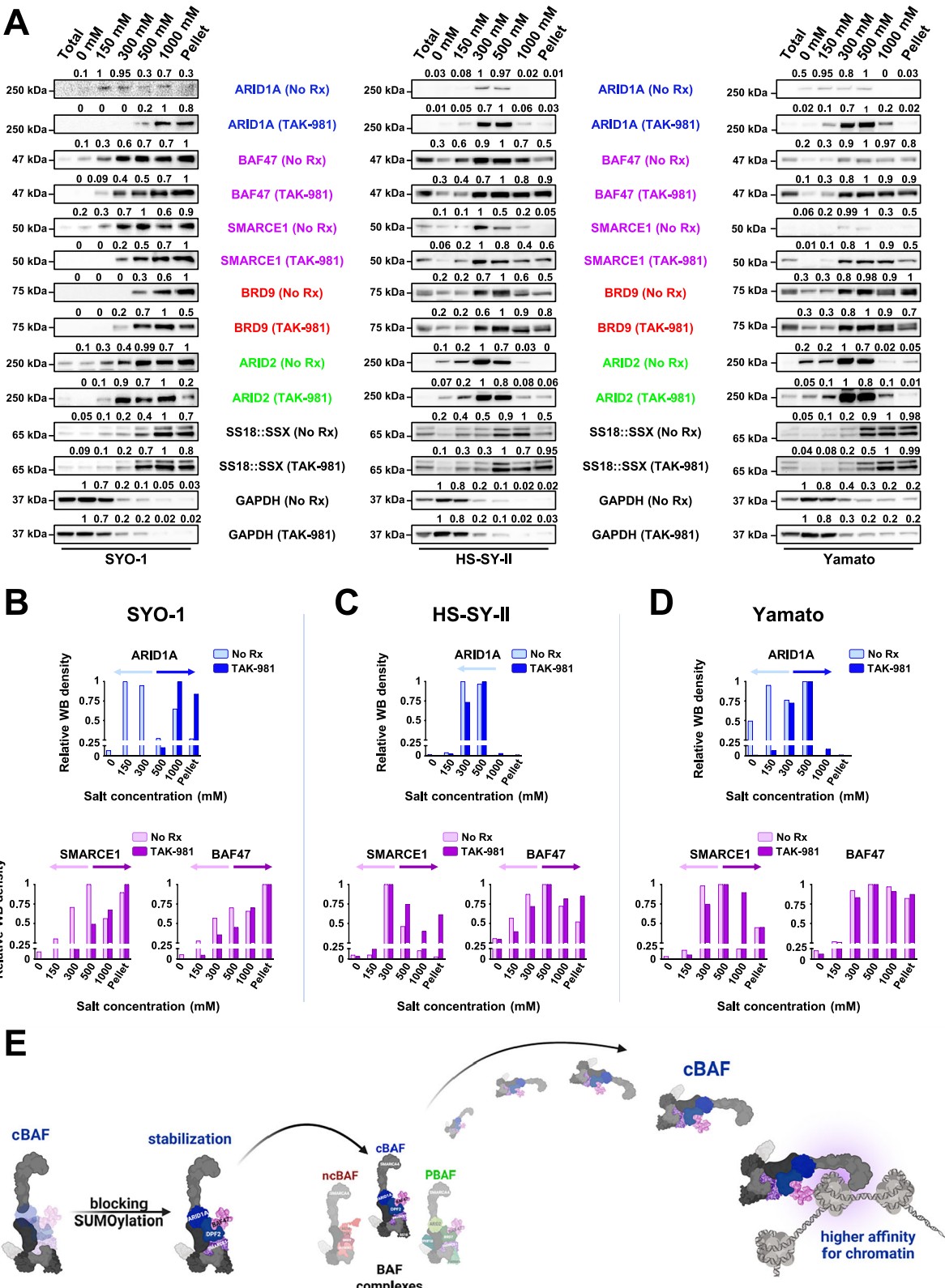

DNA repair related genes (Supplementary Fig. 7D, Supplementary Fig. 7E and Supplementary Fig. 7F) demonstrating a correlation between the binding of the BAF complexes and the attenuation of the DNA repair response.

We next tested TAK-981 in vivo, as dosed previously[19], in two SS cell line-derived xenograft models (HS-SY-II and SYO-1) and one PDX model (SS.PDX). All tumors were grown in NOD Scid Il2r Gamma (NSG)

mice. The HS-SY-II and SYO-1 models were treated with 25 mg/kg and 50 mg/kg of TAK-981 via tail vein injection, three consecutive days per week, and compared to the untreated mice. The HS-SY-II tumors demonstrated tumor growth inhibition at both doses (Fig. 8A and Supplementary Fig. 8A, Supplementary Fig. 8B) while the SYO-1 tumors displayed tumor shrinkage at the low dose and were not detectable by calipers at the high dose of 50 mg/kg (Fig. 8B and Supplementary

**Fig. 4 | TAK-981 treatment induces stronger binding of the cBAF complex to chromatin. A** Immunoblot of SS cells for specific BAF components was performed following treatment with no drug and 100 nM of TAK-981 for 36 h and differential salt extraction (0–1000 mM NaCl). GAPDH was used as a cytoplasmic loading control. Protein band intensities were quantified using GeneTools software with background subtraction and normalized to the expression of the strongest band within each blot, for drawing conclusions regarding the shift of each BAF complex component (elution at lower or higher salt concentrations) and the chromatin affinity before and after TAK-981 treatment, independent of the change in its total expression levels. **B**, **C**, **D** The band densities of ARID1A, SMARCE1 and BAF47 for the three SS cell lines from (**A**) were graphed using the GraphPad Prism software. **E** Illustration depicting the stabilization of the cBAF complex and the gain in its chromatin affinity after blocking SUMOylation compared to ncBAF and PBAF complexes (created in BioRender. Floros, K. (2025)). The colors that were used for the names of the BAF complex components throughout all figures of the manuscript are as follows: For unique cBAF complex subunits (e.g., ARID1A) a blue color has been used. For unique PBAF complex subunits (e.g., PBRM1 or ARID2) a green color has been used. For unique ncBAF complex subunits (e.g., BRD9 or GLTSCR1) a red color has been used. For cBAF/PBAF complex subunits (e.g., SMARCE1 or BAF47) a purple color has been used.

Fig. 8A, Supplementary Fig. 8B). Furthermore, to explore the lowest efficient dose of the drug in vivo, we treated the SS.PDX model with 7.5 mg/kg of TAK-981, for three consecutive days per week, for two cycles (days of treatment: 1, 2, 3, 8, 9, 10) and monitored the tumor growth after the 10th day of the study, for a total treatment period of 4 weeks. Even at these doses, tumor growth was almost completely blocked (Fig. 8C and Supplementary Fig. 8A, Supplementary Fig. 8B). Representative tumors of the SYO-1 model, before and after drug treatment, were subsequently excised and stained by immunohistochemistry (IHC) for cleaved caspase-3 and γ-H2AX to corroborate cell death and the DNA damage induction noted in our in vitro experiments (Fig. 8D and Supplementary Fig. 8C), as well as for SMARCE1, SMARCB1 (BAF47) and ARID1A expression, verifying the elevation of the specific fellow cBAF complex components (Fig. 8D and Supplementary Fig. 8C). The staining was also quantified and the H-score was calculated for the expression of each protein, verifying the observed differences (Fig. 8E). There was no weight loss of the mice for any of our in vivo treatments, indicating that the administration of TAK-981, at all doses tested, is well tolerated (Supplementary Fig. 8D, Supplementary Fig. 8E and Supplementary Fig. 8F).

### TAK-981 sensitizes SS to cytotoxic chemotherapy in vitro and in vivo

Doxorubicin (DOX)-based chemotherapy remains a common first-line therapy for SS, with the addition of ifosfamide (IFO) providing some benefit[49]. Since both DOX and IFO are DNA damaging chemicals, DOX by intercalation and IFO by alkylation[50–52], we hypothesized that the addition of TAK-981 would synergize with these drugs via enhanced DNA damage. We found significant synergy of DOX/IFO with TAK-981 (72 h) using a Bliss-Sum matrix and calculating a percentage over the Bliss score[53] across multiple doses of the drugs (Fig. 8F and Supplementary Fig. 9A)[54,55]. In line with our hypothesis, fragmentation of DNA was more pronounced in combination treated cells than those cells treated with either DOX plus IFO or TAK-981 alone, as assessed by the COMET assay[56]. More specifically, the combination treatment of the two cell lines that exhibited the highest Bliss synergy score (ASKA and HS-SY-II) resulted in greater length of the "tails" escorting the stained nuclei (Supplementary Fig. 9B and Supplementary Fig. 9C), that was also quantified by their tail moments[57] (Supplementary Fig. 9B, right and Supplementary Fig. 9C, right). This data provides a rationale for combining TAK-981 with DOX and/or IFO.

To minimize the adverse effects from the usage of multiple agents, we attempted to test the efficacy of the combination of TAK-981 with IFO in vivo, without the presence of DOX. IFO alone has been used several times for the therapy of STS patients[58–61] and there are cases where DOX and IFO together do not induce greater activity than IFO alone[62]. The administration of IFO + TAK-981 (depicted in Supplementary Fig. 9D) demonstrated impressive combination activity compared to single-agent activity, with the tumors treated with the combination shrinking (Fig. 8G, Supplementary Fig. 9E and Supplementary Fig. 9F). Noteworthy, no weight loss was detected in the combination-treated mice, suggesting tolerability (Supplementary Fig. 9G). Finally, IHC analysis of the tissues and its quantification by calculating the H-score of the combination cohort revealed enhanced

staining for cleaved caspase-3, γ-H2AX, the cBAF component (ARID1A) and the cBAF/PBAF proteins (BAF47, SMARCE1). In contrast, the expression of the ncBAF unique subunits BRD9 and GLTSCR1 remained unchanged (Fig. 8H, I and Supplementary Fig. 9H), in line with our in vitro results (Fig. 3A, B).

### TAK-981 blocks tumor growth in a mouse model with conditional SS18::SSX expression

To further test the effect of TAK-981 in vivo, we utilized a genetic SS mouse model that conditionally expresses SS18::SSX2 with littermate-controlled cohorts of *hSS2* mice (homozygous for *hSS2* at the *Rosa26* locus)[11] (Fig. 9A). TATCre was injected to initiate SS tumors in the *hSS2* mice (Fig. 9A), followed by 25 mg/kg of TAK-981 or vehicle administration. Consistent with the mouse models harboring human SS tumors, TAK-981 treatment led to significantly reduced tumors compared to the vehicle-treated tumors (Fig. 9B–D). RNA-seq analysis of vehicle and TAK-981 treated tumors identified clusters of genes that clearly differentiated between the two cohorts (Fig. 9E) and pathways associated with DNA repair and the cell cycle were significantly downregulated in TAK-981 treated tumors (Fig. 9F, Supplementary Fig. 10A and Supplementary Fig. 10B). In contrast, pathways related to muscle function, including those involved in cardiac muscle contraction and hypertrophic cardiomyopathy, as well as to ECM receptor interaction and focal adhesion, exhibited a marked upregulation in TAK-981 treated tumors (Fig. 9F, Supplementary Fig. 10A and Supplementary Fig. 10C). These data demonstrate that SUMOylation pathway inhibition in a conditionally expressed SS18::SSX2 mouse model leads to a reduction of the neuronal characteristics of SS tumors and a concurrent gain of muscle-related phenotypes, as was also previously demonstrated by the RNA-seq, ChIP-seq and ATAC-seq analyses of our in vitro experiments (Figs. 5A, 6A, D, Supplementary Fig. 7A and Supplementary Fig. 7B). Additionally, a significant overlap was noted between the up- and down-regulated genes of our mouse model and the 100 most up- and down-regulated genes after *SS18::SSX* knockdown by Banito et al.[22] (Fig. 9G, H). TAK-981 induced cell death was corroborated by western blotting of cleaved PARP (Fig. 9I). Furthermore, we noted elevated expression of BAF47 and SMARCE1 as determined by IHC (Fig. 9J, K and Supplementary Fig. 10D), supporting the model of restoration of cBAF complexes across both in vitro and in vivo models in response to the inhibition of SUMOylation in SS. The core findings of our study are summarized in the graphical abstract in Fig. 10.

## Discussion

SS18::SSX is a fusion oncogene found invariably in SS tumors that assembles into BAF complexes and leads to the eviction of the wild-type SS18 protein from these complexes[13].

SS18::SSX exerts its oncogenic effect by rebalancing BAF complexes from a cBAF dominated landscape to a landscape with an increasing relative prevalence of ncBAF, and to a lesser extent, PBAF. As such, reversing this re-balancing is an attractive therapeutic strategy for SS. Indeed, either decreasing ncBAF complex activity or increasing cBAF complex activity has anti-SS activity[10–12]. In this study, we demonstrate that this rebalancing can be achieved through blocking SUMOylation with the small molecule inhibitor, TAK-981.

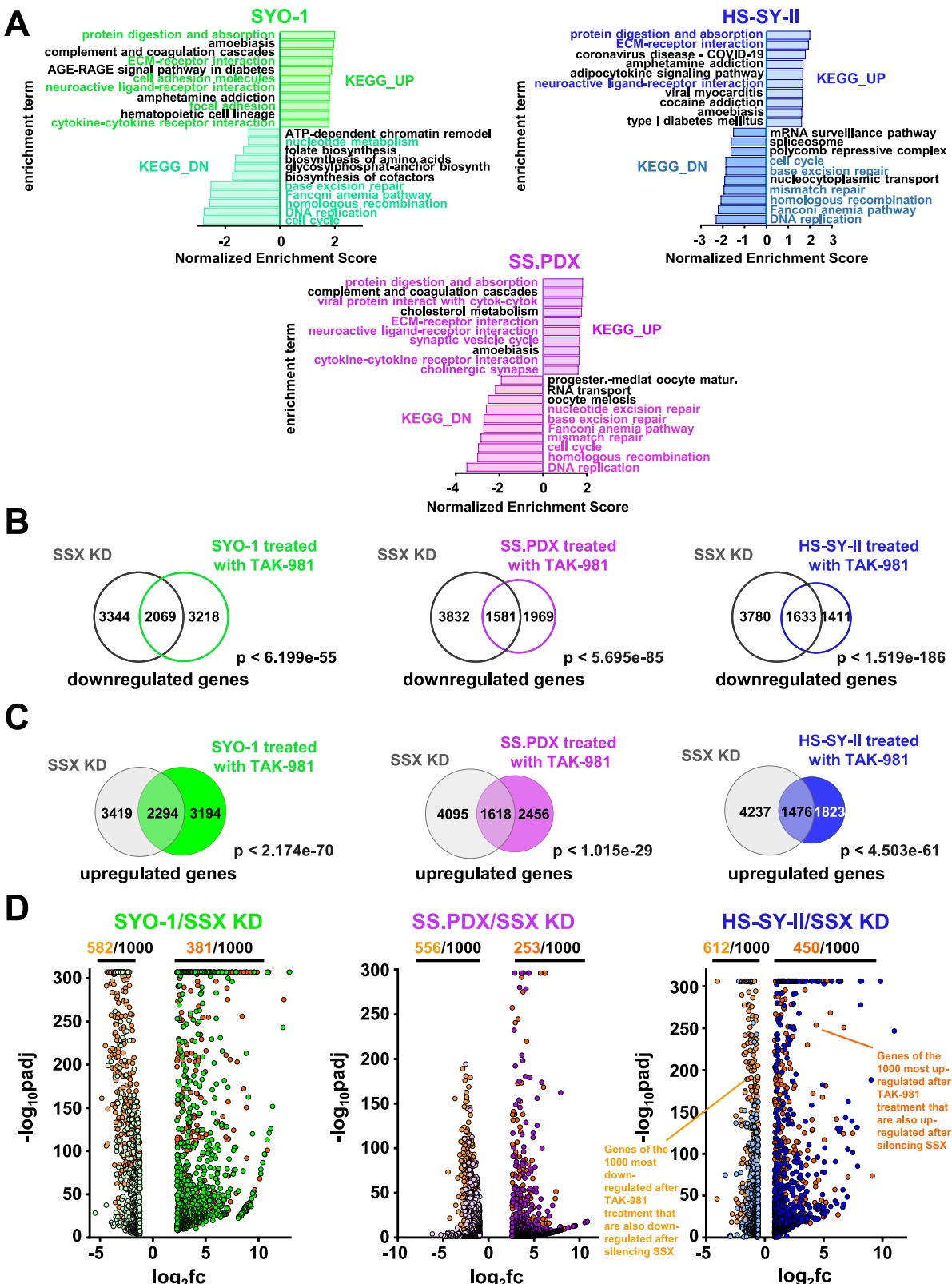

**Fig. 5 | Blocking SUMOylation leads to disruption of the synovial sarcoma signature and to induction of mesenchymal differentiation. A** Normalized enrichment scores after KEGG analysis of RNA-seq data demonstrating pathways altered by 100 nM TAK-981 treatment for 36 h in the indicated SS cells. **B** Venn diagram of genes significantly downregulated following *SS18::SSX* knockdown or treatment with 100 nM TAK-981 for 36 h. The significance of the gene set overlaps (*p* value) was calculated with Chi-square test under the assumption of 17,611 expressed transcripts. **C** Venn diagram of genes significantly upregulated following *SS18::SSX* knockdown or treatment with 100 nM TAK-981 for 36 h. The significance of the gene set overlaps (*p* value) was calculated with Chi-square test under the assumption of 17,611 expressed transcripts. **D** Volcano plot of the most down- or upregulated 1000 genes following TAK-981 treatment (detected by RNA-seq), plotted with the commonly downregulated (light orange dots) or the commonly upregulated genes (dark orange dots) after *SS18::SSX* KD, among the indicated SS cell line. The *p* values were adjusted using a False Discovery Rate (FDR) multiple testing correction method.

Overall, SUMOylation appears to most prominently impact chromatin structure and transcriptional function[63–66]. SUMOylation often negatively affects protein expression, but it can also increase protein expression: For instance, since SUMOylation and ubiquitination compete for lysines, SUMOylation can lead to hinderance of ubiquitination and thus stabilization of the affected proteins[67,68]. Conversely, STUbLs like the SUMO-dependent E3 ubiquitin ligases RNF4 and TOPORS can degrade SUMOylated proteins leading to their depletion[69,70]. Indeed, in our study, we found that SUMO-dependent SMARCE1 depletion was the result of coordinated action of RNF4 and TOPORS. Consistent with recent findings in clear cell meningioma, where SMARCE1 is a stabilizer of the cBAF complex on chromatin[36], we found increased SMARCE1 expression was sufficient to stabilize the cBAF complex in SS, following TAK-981 treatment. In contrast, deSUMOylation of SMARCE1 does not affect the integrity of the PBAF complex, despite being an established member of both complexes (cBAF and PBAF), in line with the findings of the previous study by St Pierre et al.[36]. This was sufficient to shift BAF complexes away from the SS state, with marked loss of the SS signature. Our data strongly points to the redistribution of BAF complexes as the major mechanism of TAK-981 toxicity in SS. Using a genetically engineered mouse model of SS (Fig. 9, Supplementary Fig. 10), we can recapitulate the effect of TAK-981 on cBAF re-activation. However, we cannot rule out that other changes to proteins outside of BAF complexes are contributing to toxicity.

It should be noted that although the PTMScan data did not detect SUMOylation of BAF47 in native conditions, we did detect SUMOylation upon exogenous expression of SUMO2. This raises the possibility that other SUMOylation events in BAF complex members outside of SMARCE1 could occur in certain contexts and could theoretically contribute to the cBAF complex stabilization that we observe following TAK-981 treatment. Consistent with increased SMARCE1 expression ultimately resulting in TAK-981 toxicity in SS, our attempts to reconstitute wild-type SMARCE1 and SMARCE1 mutated at the sites of SUMOylation (K92 and K277) either constitutively or through an induced system resulted in significant toxicity that precluded performing functional assays. However, we have noticed upon interrogation of the Broad Institute depository (Depmap) that *RNF4* silencing is more toxic to SS than it is to 30 other cancer subtypes (Fig. 2J), confirming our hypothesis that increased SMARCE1 expression is the linchpin of the exquisite toxicity of TAK-981 in SS.

Furthermore, we demonstrate that SS18::SSX binds to SUMOylation genes to activate the SUMOylation program, suggesting the SUMOylation of SMARCE1 is an important and newly described oncogenic function of SS18::SSX. Knockdown of SS18::SSX and TAK-981 treatment have largely overlapping transcriptome changes, and knockdown of SS18::SSX can significantly phenocopy the alterations of the cBAF complex stability induced by SUMOylation inhibition. Overall, our study strongly implicates that TAK-981 acts as a surrogate SS18::SSX inhibitor in SS and has substantial anti-SS activity in vitro and in vivo. As such, SUMOylation inhibition should be viewed as an exciting new strategy to treat SS.

Preclinical studies have also demonstrated a role for TAK-981 in immune cells, changes that may promote cancer treatment benefit[71,72]. The current study did not evaluate a role of the immune system in SS, and further studies should be performed to elicit any additional effects TAK-981 may have in this context. Furthermore, while our data suggests weaker binding of SMARCA4 and SS18::SSX to promoters bearing motifs of transcription factors of DNA repair related genes following TAK-981 treatment (Supplementary Fig. 7D, Supplementary Fig. 7E and Supplementary Fig. 7F), SUMOylation also plays a direct role in the DNA damage response through modification of additional DNA damage and repair genes[73,74]. Thus, further studies are needed to more fully characterize the mechanism of DNA damage following TAK-981 treatment in SS.

Metastatic SS continues to have a poor prognosis due to poor disease control by available therapies. DOX-based chemotherapy remains the commonly practiced standard therapy in many centers, and adding IFO to DOX increases overall response rates and progression-free survival, but neither significantly extends overall survival[49,75]. Pazopanib is a pan-kinase inhibitor approved for SS but has a low response rate[76,77]. Merck's PD-1 inhibitor, pembrolizumab, was recently evaluated in STSs including 10 SS cases with only one patient having a response[78]. In addition, BRD9 degraders have been tested clinically, however the most advanced (CFT8364; C4 Therapeutics) has recently been discontinued. In a phase I trial across multiple patients with metastatic solid tumors or refractory/relapsed lymphoma, TAK-981 demonstrated tolerability and a RP2D of 90 mg BIW was established, and TAK-981 has also been evaluated in combination with pembrolizumab (NCT04381650). Of note, to our knowledge, TAK-981 has not been evaluated in SS patients.

Overall, we demonstrate SUMOylation inhibition is exceptionally effective in SS and leads to a relative increased expression and activation of cBAF and subsequent reduction of the SS signature. Our data provides rationale to test TAK-981 in SS patients, either alone or combined with cytotoxic chemotherapy.

## Methods

### Animal studies

All mouse experiments of Fig. 8, Supplementary Fig. 8 and Supplementary Fig. 9D–H were approved and performed in accordance with the Institutional Animal Care and Use Committee at VCU (protocol number: AD10001048). The hSS2 conditional mouse experiment (Fig. 9 and Supplementary Fig. 10) was approved by the University of Utah animal care committee in accordance with international legal and ethical standards (protocol number: 00001442).

### Cell lines

The SS.PDX cell line was established ex vivo from a SS patient-derived xenograft model. To generate a cell suspension from the xenograft tumor, the tissue was minced and digested for 1 h at 37 °C in DMEM (11995-065; Gibco), 1x penicillin/streptomycin (15140-122; Gibco), 2.5 mg/mL collagenase B (11088831001; Roche), 100 units/mL DNase I (10104159001; Roche), and 10% FBS (A52568-02; Gibco), passing through a 70 μm strainer. Red blood cells were lysed using RBC lysis buffer (00-4333-57; Thermofisher). ASKA (RCB3576), HS-SY-II (RCB2231) and Yamato (RCB3577) cell lines were obtained from Riken Bioresource Center (Tsukuba, Japan). SYO-1 and Fuji cell lines were provided by Dr. Torsten Nielsen (University of British Columbia, Vancouver, Canada). Cells were authenticated by quantitative PCR detection of SS18::SSX1/2, which is specific to SS. Routine mycoplasma testing was performed on all cell lines, using the MycoAlert Mycoplasma Detection Kit (LT07–318; Lonza). SS.PDX, SYO-1, HS-SY-II, Yamato and Fuji, cell lines were cultured in DMEM (11995-065; Gibco) with 10% FBS (A52568-02; Gibco) and 1 μg/mL penicillin and streptomycin. ASKA cells were grown in DMEM, with 20% FBS and 1 μg/mL penicillin and streptomycin.

### Reagents

The following drugs were purchased: TAK-981 for in vitro and in vivo studies (CT-TAK981; Chemietek), ifosfamide for in vivo and in vitro studies (I4909; Sigma-Aldrich), doxorubicin (CT-DOXO; Chemietek) for in vitro studies. The antibodies used for western blotting in this study were as follows: anti-BCL2 (D55G8) (human specific) (4223; Cell Signaling), anti-cleaved PARP (Asp214) (D64E10) (5625; Cell Signaling), anti-SMARCE1 (E6H5J) (33360; Cell Signaling), anti-GAPDH (6C5) (sc-32233; Santa Cruz), anti-SMARCB1/BAF47 (D8M1X) (91735; Cell

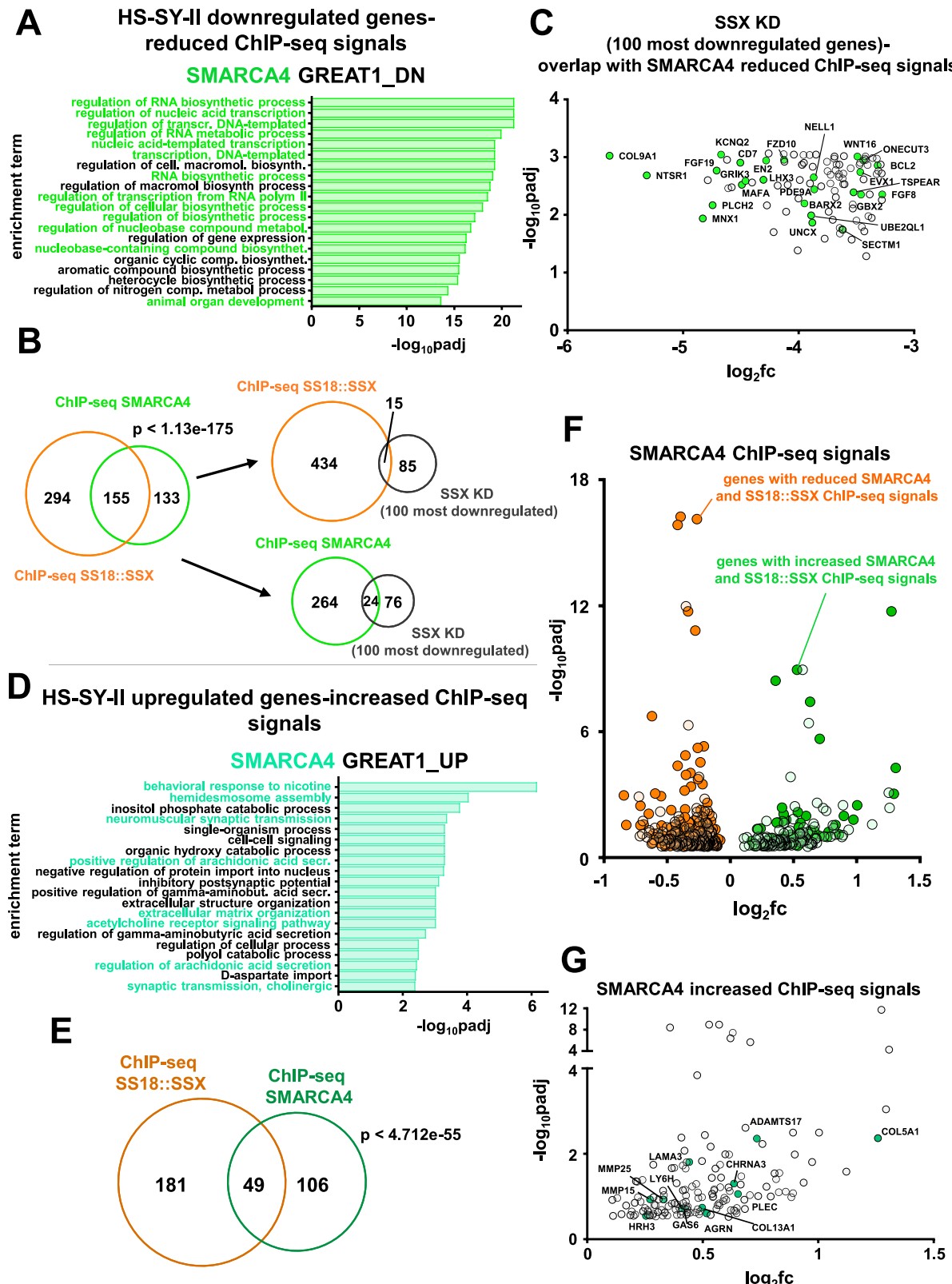

**A** HS-SY-II downregulated genes- reduced ChIP-seq signals

**B**

**C** SSX KD (100 most downregulated genes)- overlap with SMARCA4 reduced ChIP-seq signals

**D** HS-SY-II upregulated genes-increased ChIP-seq signals

**E**

**F** SMARCA4 ChIP-seq signals

**G** SMARCA4 increased ChIP-seq signals

Signaling), anti-BRD9 (24785-1-AP; Proteintech), anti-ARID1A antibody (EPR13501-73) (ab182561; Abcam), anti-PBRM1 (D3F7O) (91894; Cell Signaling), anti-TBP (8515; Cell Signaling), anti-SUMO-2/3 (18H8) (4971; Cell Signaling). For ChIP-seq the following antibodies were used: anti-JHDM1B (KDM2B) (17-10264; Milipore-Sigma), anti-BRG1 (SMARCA4) (EPNCIR111A) (ab110641; Abcam), anti-H3K27ac (pAb)

(39134; Active Motif) and anti-SS18::SSX (E9X9V) XP (72364; Cell Signaling) used for both western blotting and ChIP-seq. For the SUMO detection co-IP, SUMO-2/3 affinity beads (ASM24-Beads; Cytoskeleton) and mouse IgG IP Control Beads (CIG01-Beads; Cytoskeleton) were used. For the SMARCE1 co-IP, anti-BAF57/SMARCE1 antibody (EPR8849) (ab137081; Abcam) and normal rabbit IgG (2729S; Cell

**Fig. 6 | The binding of BAF complexes to chromatin is responsible for many of the observed expression changes of the SS transcriptome after blocking SUMOylation. A** Gene ontology analysis of genes associated with significantly reduced SMARCA4 ChIP-seq signals after TAK-981 treatment in HS-SY-II cells. **B** Venn diagrams overlap in significantly downregulated genes among the different noted experiments. The significance of the gene set overlap (*p* value) was calculated with Chi-squared test under the assumption of 17,611 expressed transcripts. **C** Overlap of the 100 most significantly downregulated genes following *SS18::SSX* knockdown by Banito et al.[22] with significantly reduced SMARCA4 ChIP-seq signals. **D** Gene ontology analysis of genes associated with significantly upregulated

SMARCA4 ChIP-seq signals after TAK-981 treatment in HS-SY-II cells. **E** Venn diagram of the overlap of significantly enhanced SS18::SSX and SMARCA4 ChIP-seq signals in HS-SY-II cells. The significance of the gene set overlap (*p* value) was calculated with Chi-square test under the assumption of 17,611 expressed transcripts. **F** Volcano plot of the reduced and increased SMARCA4 ChIP-seq signals plotted with the genes of commonly reduced (dark orange dots) or commonly enhanced (dark green dots) SS18::SSX ChIP-seq signals. **G** Genes related to the mesenchymal phenotype associated with increased SMARCA4 ChIP-seq signals (green dots). *p*adj = adjusted *p* values. For (**A, C, D, F, G**) the *p* values were adjusted using a False Discovery Rate (FDR) multiple testing correction method.

Signaling) were used. The secondary antibodies that were used in this study were as follows: ECL Anti-mouse IgG Horseradish Peroxidase conjugated antibody (NA931V; Amersham) and ECL Anti-rabbit IgG Horseradish Peroxidase conjugated antibody (NA934V; Amersham). All primary antibodies were used at 1:1000 dilutions. Secondary antibodies were used at 1:10,000 dilutions. The antibodies that were used for IHC staining are listed in the "IHC staining and calculation of the H-score" section.

## Vector construction and lentiviral transduction

*SS18::SSX*-specific shRNA expression vectors were provided by A. Banito and delivered by lentiviral transfection as previously described (shREN-control)[22]. For lentivirus production, $1 \times 10^6$ HEK293T cells were transfected with 3 µg of constructs and helper vectors (2.5 µg of psPAX2 and 0.9 µg of VSV-G). Transfection of packaging cells was performed using polyethyleneimine (23966-2; Polysciences) by mixing with DNA in a 3:1 ratio. Viral supernatants were collected 48 h after transfection, filtered through a 0.45-µm filter and supplemented with 4 µg ml$^{-1}$ polybrene (Sigma) before being added to target cells.

## siRNA knock-down

For the short-interfering RNA (siRNA) experiments, siRNA against *RNF4* (ON-TARGETplus Human RNF4, L-006557-00-0005; Horizon), a negative control siRNA (ON-TARGETplus Non-targeting Control siRNA, D-001810-01-20; Horizon) and a custom siRNA against *TOPORS* using the following sequence: 5′-GUCCUAAGGCCUUCGUAUAAU-3′[39], from Horizon were purchased. For *SS18::SSX* knockdown, duplex oligo (sense, CAAGAAGCCAGCAGAGGAATT; antisense, UUCCUCUGCUGG CUUCUUGTT) *SS18::SSX* siRNAs were designed to target the SSX portion of *SS18::SSX* using the Integrated DNA Technologies RNA interference (RNAi) design tool and synthesized by Integrated DNA Technologies (IDT). All siRNAs were used at a concentration of 50 nM and reverse transfected with Lipofectamine RNAiMAX transfection reagent (13778075; ThermoScientific) using the manufacturer's protocol.

## Western blotting

Whole cell lysates (supplemented with 0.5 M of N- ethylmaleimide (NEM) (23030; Thermofisher)) were prepared in lysis buffer (20 mM Tris-HCl (351-091-101; Quality Biological), 150 mM NaCl (V4221; Promega), 1% Nonidet P-40 (J19628.AP; Thermofisher), 1 mM EDTA (351-027-721; Quality Biological), 1 mM EGTA (BM-151; Boston Bioproducts), 10% glycerol (G5516; Sigma), and protease and phosphatase inhibitors), incubated on ice for 15 min, and centrifuged at max speed for 10 min at 4 °C. Equal amounts of the detergent-soluble lysates were resolved using the NuPAGE Novex Midi Gel system on 4–12% Bis–Tris gels (WG1402BOX; Invitrogen), transferred to PVDF Immobilon-P transfer membranes (IPVH00010; Millipore) in between six pieces of Whatman Cellulose Chromatography paper (21427-524; Avantor) set in 10x Tris/Glycine transfer buffer (1610771; Biorad) with 20% methanol, and following transfer and blocking in 5% nonfat milk in PBS, probed overnight with the antibodies listed above. Representative blots from at least three independent experiments are shown in the figures. Chemiluminescence was detected with the Syngene G: Box camera

(Synoptics). Densitometry was performed using the GeneTools software suite (Syngene). Protein bands intensities were quantified with background subtraction and normalized to the expression of the loading control for each lane. Within each blot, values were normalized to the No Rx levels (n/a: the blots couldn't be quantified due to the extremely low density in the untreated lysates). The unit for all molecular weight markers is kDa. Uncropped scans of all blots are located in the Source Data file.

## Nuclear extraction

Cells were seeded in a 150 mm dish per condition and incubated with or without TAK-981 according to the assay's requirements. Cells were washed with 1x PBS, harvested, pelleted by centrifugation for 5 min at $800 \times g$ and resuspended in lysis buffer (20 mM Tris, 150 mM NaCl, 1% Nonidet P-40, 1 mM EDTA, 1 mM EGTA, 10% glycerol, and protease and phosphatase inhibitors), supplemented with 0.5 M of N- ethylmaleimide (NEM). Afterwards cells were allowed to swell on ice for 20 min and centrifuged at $12,000 \times g$ for 10 min at 4 °C. The pelleted nuclei were resuspended in RIPA buffer (supplemented with Halt Protease and Phosphatase Inhibitor Cocktail (100X) (78442; Thermofisher) and 0.5 M of N-ethylmaleimide (NEM)) and RIPA lysates were sonicated in a Diagenode Bioruptor 300 for 5 min. Nuclear extracts were collected by centrifugation at $12,000 \times g$ for 15 min at 4 °C. Tumors from the SS18::SSX2 conditional mouse model were homogenized with Tissuemiser (Fisher Scientific) in the lysis buffer described previously, incubated for 20 min on ice, and centrifuged at max speed for 10 min at 4 °C, prior to following the protocol for the nuclear extraction described above.

## Cell viability

For the Cell Titer-Glo experiments, 1000–3000 seeded cells per well in 96-well flat-bottom black plates were treated with 25 µL of CellTiter-Glo Luminescent Cell Viability Assay (G7573; Promega), following continuous drug treatment (each time with the indicated drugs at the indicated concentrations), at 37° and 5% atmospheric $CO_2$ and immediately read on an H1 Biotek plate reader according to the Promega protocol. Quantification of no-treatment seeded cells was used to determine the total cell growth number over the experiment. All data are means + SD of three independent experiments ($n = 3$).

## Crystal violet

Cells were seeded at 50,000 cells per well in a six-well dish and treated the following day with TAK-981. Five days later, when untreated cells reached confluency, cells were stained with 0.1% crystal violet (405830250; Thermofisher). For Fig. 1F, HS-SY-II cells underwent CRISPR/Cas9-mediated gene targeting of the indicated SUMOylation genes and cells were seeded at 20,000 or at 40,000 cell per 6- well plate and stained with crystal violet 12 days later.

## Synergy assay

ASKA, HS-SY-II, SS.PDX and Yamato cells were seeded in quadruplicate at $1 \times 10^3$ cells in a 96-well plate. Twenty-four hours after seeding, cells were treated with varied concentrations of TAK-981 (1–100 nM) and of doxorubicin (DOX) (1–7 nM)/ifosfamide (IFO) (1–7 mM) for 72 h,

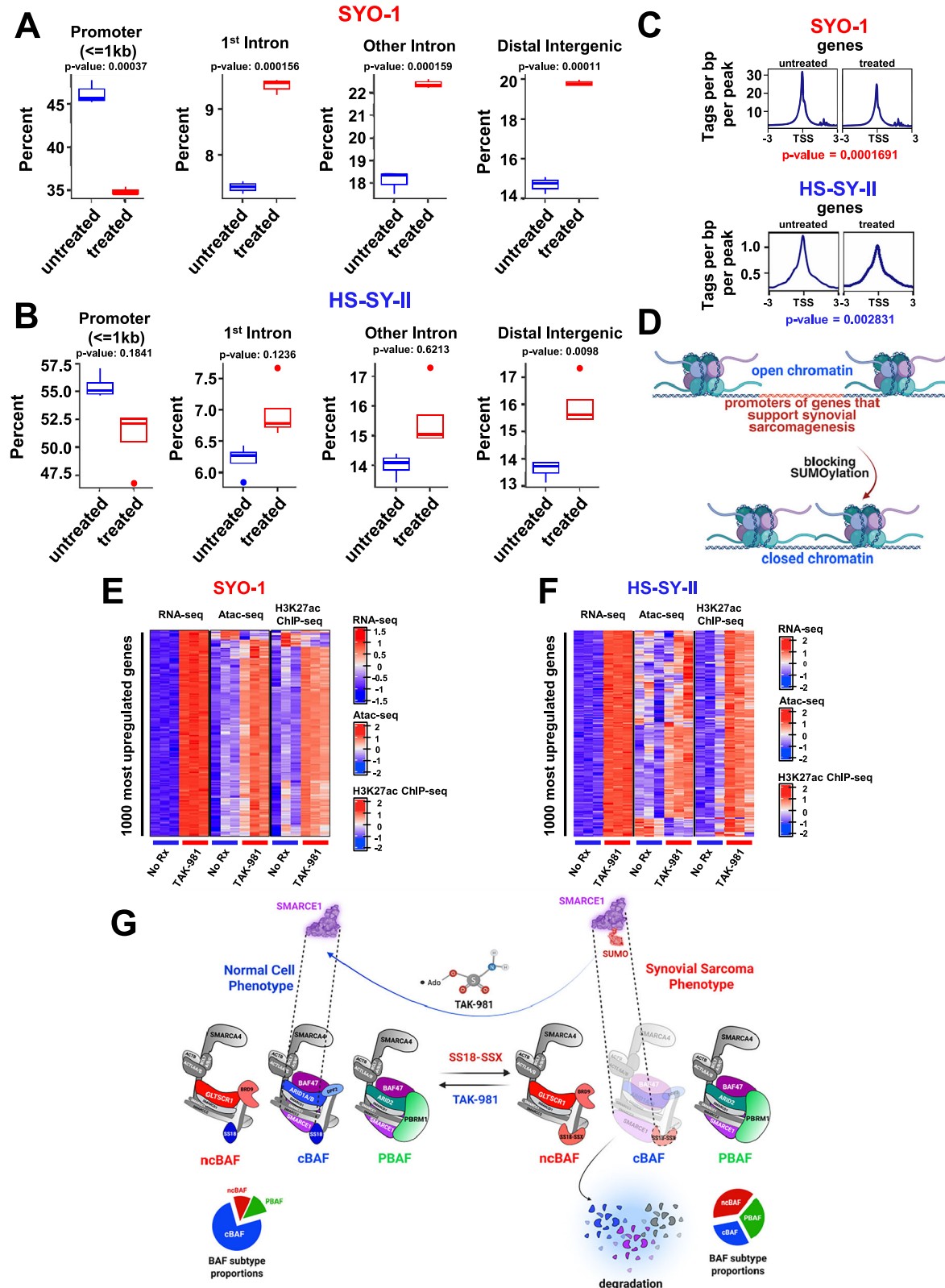

followed by measurement of cell viability by CellTiter-Glo. Percent viability was constrained to a maximum of 100. Percent over the bliss score was calculated as previously described[53].

**CRISPR/Cas9 gene editing**

Single guide RNAs (sgRNAs) against SUMO2, UBE2I and UBA2 were designed using the GUIDES tool (http://guides.sanjanalab.org) and

cloned by annealing two DNA oligos and ligating into a BsmB1-digested pLKO1-puro-U6-sgRNA-eGFP. Transformation was carried out using Stbl3 bacteria (C737303; Invitrogen). The same vector without an sgRNA (empty vector) was used as a control in subsequent experiments. A well validated guide against SSX1[79] was used as a positive control. HS-SY-II cells were transduced with lentiCas9-Blast (52962; Addgene) and selected using 20 μg ml⁻¹ blasticidin to generate stable

**Fig. 7 | Addition of TAK-981 results in loss of chromatin accessibility to promoters of the SS18::SSX transcriptome. A, B** Boxplots demonstrating distribution of SYO-1 and HS-SY-II ATAC-seq peaks genome-wide after 100 nM of TAK-981 treatment (72 h) (*n* = 3 biological replicates for No Rx and *n* = 3 biological replicates for TAK-981-treated SYO-1 cells and *n* = 4 biological replicates for No Rx and *n* = 4 biological replicates for TAK-981 treated HS-SY-II cells), focusing on downregulated RNA-seq normalized genes for HS-SY-II cells transduced with *SS18::SSX* shRNA from Banito et al.[22]. Differential accessibility peaks were annotated using the ChIPseeker v.1.40.0 R package and percentages of peaks associated with various gene-centric annotations were compared using unpaired two-sided *t* test. The boxplots show the median (central line), and the interquartile range of the data. The whiskers extend to the minimum and to the maximum values excluding outliers. **C** SYO-1 and HS-SY-II ATAC-seq profiles after TAK-981 treatment focusing on downregulated RNA-seq normalized genes of HS-SY-II cells transduced with *SS18::SSX* shRNA[22] (the same list of downregulated genes used in (**A**, **B**), centered on TSS (transcription starting sites). For calculating the *p* values two-sided Wilcoxon tests were performed. **D** Schema of changes at the chromatin following TAK-981 treatment in SS. **E, F** Heatmaps of jointly clustered row-normalized gene expression (RNA-seq), and signal (ATAC-seq, ChIP-seq) of the top 1000 most upregulated genes and the associated peaks before and after TAK-981 treatment (number of replicates for each condition, *n* = 3). **G** Model of TAK-981 efficacy. TAK-981 deSUMOylates SMARCE1, leading to its accumulation. SMARCE1 upregulation leads to cBAF complex stabilization and redistribution of the BAF complexes (towards a normal cell phenotype) (**D**, **G**) were created in BioRender. Floros, K. (2025).

---

Cas9-expressing cell lines. Cells were subsequently transduced with sgRNAs. After 3 days of infection, cells were selected with 2 μg ml⁻¹ puromycin. Following selection, cells were counted and plated at low density (20,000 or 40,000 cell per 6 -well plate) and stained with crystal violet 12 days later. The sgRNA sequences were as following: SUMO2_sgRNA1_fwd: CACCGCGATCATATTAATTTGAAGG, SUMO2_sgRNA_1_rev: AAACCCTTCAAATTAATATGATCGC SUMO2_sgRNA_2_fwd: CACCGGGCGGGGCAGGATGGTTCTG

SUMO2_sgRNA_2_rev: AAACCAGAACCATCCTGCCCCGCCC UBA2_sgRNA_1_fwd: CACCGTCAATAAAGTATAGTCCTGG

UBA2_sgRNA_1_rev: AAACCCAGGACTATACTTTATTGAC
UBE2I_sgRNA_1_fwd: CACCGACATTCGGGTGAAATAATGG
UBE2I_sgRNA_1_rev: AAACCCATTATTTCACCCGAATGTC
UBE2I_sgRNA_2_fwd: CACCGGTGCCTGTCCATCTTAGAGG
UBE2I_sgRNA_2_rev: AAACCCTCTAAGATGGACAGGCACC

### RNA extraction and qRT-PCR
RNA was isolated from cultured cells grown at sub-confluency using the Isolate IIRNA Mini kit (BIO-52072, Bioline), and RNA was reverse-transcribed to form cDNA molecules using the High-Capacity cDNA Reverse Transcription kit (4374966; Applied Biosystems) on a 7500 Fast Real-Time PCR System (Applied Biosystems). The expression of the genes was measured using a ProFlex PCR System (Applied Biosystems) that quantified bound SYBR Green (Perfecta SYBR Green FastMix, 95072-250; QuantaBio). *ACTB* was used as internal control. The primers for the human genes used for the study are listed on Supplementary Data 4. To determine relative abundance of the genes in relation to *ACTB*, the Delta-Delta CT (cycle threshold) method was utilized. All data are means ± SEM of three independent experiments (*n* = 3).

### Chromatin immunoprecipitation (ChIP)
For chromatin immunoprecipitation (ChIP) experiments, indicated cells were harvested following 36 h exposure to 100 nM of TAK-981. ChIP experiments were performed as previously[80]. Briefly, ~20 million cells from SYO-1 or HS-SY-II cells per sample were cross-linked for 10 min with 1% formaldehyde at 37 °C. The reaction was subsequently quenched with 125 mM glycine for 5 min. Chromatin from fixed cells was fragmented by sonication with a Diagenode Bioruptor 300 and the solubilized chromatin was incubated with the indicated antibodies (listed in the "Reagents") overnight at 4C. Antibody-chromatin complexes were pulled down by incubation with Protein A Sepharose CL-4B beads (GE17-0963-03; Sigma) for 2 h at 4 °C, washed and eluted. The samples finally underwent crosslink reversal, as well as RNase A (EN0531; Thermofisher), and proteinase K (20-298; Millipore) treatment. Following washing, DNA was purified using ethanol precipitation.

### ChIP-seq
If concentration allowed, ChIP-DNA was checked for size distribution using Fragment Analyzer High Sensitivity DNA assay (Agilent Technologies, Santa Clara, CA).

ChIP-seq library preparation and next generation sequencing: Approximately 0.1 ng to 20 ng ChIP DNA was fragmented to size range around 200 bps using Covaris (shearing parameter 2 min in LV tubes using 15ul microTUBE-15 AFA Beads Screw-cap, Cat: #520145), and used for ChIP-seq library preparation by following the NEBNext Ultra II DNA library preparation Kit from Illumina (E7645L, New England Biolabs Inc, Ipswich, MA, USA). This kit utilizes a proprietary adapter attachment chemistry which minimizes bias and supports very low amount of input sample. ChIP-seq libraries were subjected to quantification process by the combination of Qubit and Bioanalyzer, pooled and subsequent sequenced with either Illumina NovaSeq 6000 or NextSeq 2000 platform. After the sequencing run, demultiplexing with Bcl2fastq2 was employed to generate the fastq file for each sample. The SYO-1 and HS-SY-II SS18::SSX and KDM2B (JHDM1B) ChIP-seq data were quality controlled at each processing step using FASTQC, alignment was performed using STAR v.2.7.9a[81] using the UCSC's hg38 genome assembly. Peaks were called with MACS2[82] and differential peaks were identified using DESeq2 v.1.42.1[83] and FDR correction for multiple testing. The HS-SY-II SMARCA4 and H3K27ac ChIP-seq data was analyzed using the Nextflow ChIP-seq (https://nf-co.re/chipseq/2.0.0) pipeline. Downstream analyses included (differential) motif detection (MEME suite[84]), peak annotation (ChIPPeakAnno v.3.36.1[85]), and functional enrichment analysis of the associated genes (clusterProfiler v.4.10.1[86]).

### RNA-seq
Total RNA was isolated from SYO-1, HS-SY-II and SS.PDX cells using the Isolate IIRNA Mini kit (BIO-52072, Bioline). The quality of total RNAs was determined by Agilent Tape Station (Agilent Technologies, Santa Clara, CA), and only RNAs with RQN > 7 were used for subsequent mRNA-seq library preparation and sequencing. Stranded mRNA-seq library preparation and next generation sequencing: Approximately 500 ng of total RNA was used for stranded mRNA-seq library preparation by following the NEB Directional mRNA-seq sample preparation guide (New England Biolabs, Ipswich, MA). The first step in the workflow involved purifying the poly-A containing mRNA molecules using poly-T oligo-attached magnetic beads. Following purification, the mRNA was fragmented into small pieces using divalent cautions under elevated temperature. The cleaved RNA fragments were copied into first strand cDNA using reverse transcriptase and random primers. This was followed by second strand cDNA synthesis using DNA Polymerase I and RNase H. Strand specificity was achieved by replacing dTTP with dUTP in the Second Strand Marking Mix (SMM). These cDNA fragments then went through an end repair process, the addition of a single "A" base, and then ligation of the adapters. The products were then purified and enriched with PCR to create the final RNA-seq library. After RNA-seq libraries were subjected to quantification process, pooled and subsequent 100 bp paired read sequencing run with Illumina NovaSeq 6000 platform. After the sequencing run, demultiplexing with Bcl2fastq2 was employed to generate the fastq file, with the average of 30 M reads per sample. The SYO-1 data were aligned to the GRCh38 human genome assembly using bwa v. 0.7.17-r1188[87] and

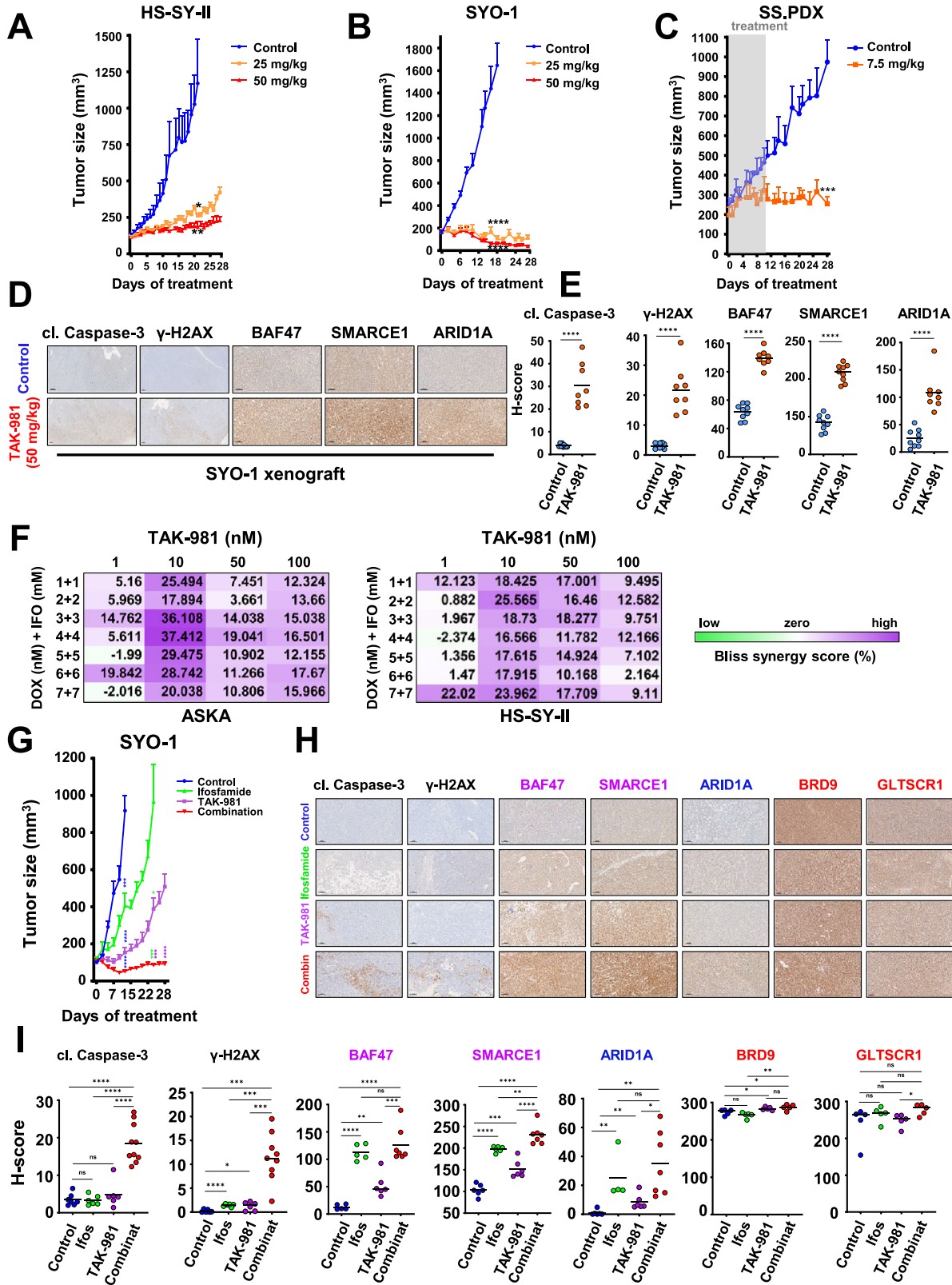

gene expression was quantified using featureCounts v. 2.0.1[88]. The HS-SY-II data were processed with the Nextflow RNA-seq pipeline (https://nf-co.re/rnaseq/3.14.0). Gene expression was quantified using the STAR-RSEM strategy. The SS.PDX RNA-seq data were analyzed by Novogene with us performing differential gene expression analysis. Differentially expressed genes were detected using edgeR v.4.0.16[89]. We also reanalyzed data from Banito et al.[22] with limma[90] to detect

genes differentially expressed between SSX1 and Ren. p values were adjusted using a False Discovery Rate (FDR) multiple testing correction method[91].

**ATAC-seq**

ATAC-seq library preparation was done by following the Active Motif ATAC-Seq Kit Manual (53150, Active Motif). ATAC-seq libraries were

**Fig. 8 | TAK-981 has activity in SS in vivo and sensitizes the cancer cells to cytotoxic chemotherapy. A** HS-SY-II tumor-bearing NSG mice were treated with 25 mg/kg or 50 mg/kg TAK-981 three consecutive days (Tuesday-Thursday) and tumors were monitored by caliper at least 3/w (*n* = 6 for control, 5 for 25 mg/kg and 6 for 50 mg/kg). **B** SYO-1 tumor-bearing NSG mice were treated with 25 mg/kg or 50 mg/kg TAK-981 3/w and tumors were monitored as before (*n* = 8 for control and 25 mg/kg and 9 for 50 mg/kg). **C** SS.PDX tumor-bearing NSG mice were treated with 7.5 mg/kg TAK-981 three consecutive days, the first and second week of the experiment. Drug administration was paused following the second round of treatment (day 10) and the mice were monitored for tumor growth for a total of 28 days (*n* = 6 for control, 5 for TAK-981). For (A–C), the data are presented as mean values + SEM. Unpaired two-tailed *t* tests were performed for the time point at which control mice were euthanized. **D** Representative images from IHC analysis of the indicated antibodies from experiment in (**B**). Scale bars = 100 μm. **E** H-scores of staining from (**D**). **F** Bliss Sum synergy scores obtained following treatment with the indicated drugs and concentrations were calculated as previously described[53]. **G** SYO-1 tumor-bearing NSG mice were treated with TAK-981 (7.5 mg/kg) three consecutive days per week, for 4 weeks (28 days) and ifosfamide via intraperitoneal injection (30 mg/kg), for three consecutive days the first and 4th week of the experiment. The average tumor volume + SEM for each cohort is displayed (*n* = 7 for control, 9 for ifosfamide, 8 for TAK-981 and 8 for combination treatment). Unpaired two-tailed *t* tests compared control and treatment mice (ifosfamide, TAK-981, combination - blue asterisks, for the 13th day of treatment). Comparisons to ifosfamide (green asterisks) and to TAK-981 (purple asterisks) are depicted for the 24th day of treatment. **H** Representative images of IHC analyses of the indicated antibodies in SYO-1 tumors. Scale bars = 100 μm. **I** H-scores from (**H**). For (**E**, **I**), unpaired two-tailed *t* tests were performed. Exact *p* values can be found in the Source Data.

subjected to quantification process by the combination of Qubit and Bioanalyzer, pooled and subsequent sequenced with 100 bp paired read sequencing using Illumina NovaSeq 6000 platform. After the sequencing run, demultiplexing with Bcl2fastq2 was employed to generate the fastq file for each sample. The SYO-1 ATAC-seq data was quality controlled at each processing step using FASTQC, alignment was performed using STAR v.2.7.9a[81] using the UCSC's hg38 genome assembly. Peaks were called with MACS2[82] and differential peaks were identified using DESeq2 v.1.42.1[83] using FDR correction for multiple testing.

The HS-SY-II ATAC-seq data was analyzed using the Nextflow ATAC-seq (https://nf-co.re/atacseq/2.1.2) pipelines. Downstream analyses included (differential) motif detection (MEME suite[84]), peak annotation (ChIPPeakAnno v.3.36.1[85]), and functional enrichment analysis of the associated genes (clusterProfiler v.4.10.1[86]).

### Xenograft studies
Six to eight week old NOD/SCID gamma (NSG) male mice were injected with ~5 × 10^6 SYO-1, or HS-SY-II cells per 200 μL of 1:1 (cells: Matrigel) into both flanks and monitored for tumor growth. When tumors reached approximately 150–200 mm³, the tumor-bearing mice were randomized to a no-treatment control group, a 25 mg/kg TAK-981 group and 50 mg/kg TAK-981 group. Mice were treated with TAK-981 via tail vein injection. The solvent for TAK-981 was 20% 2-hydroxy propyl-β-cyclo dextrin (Sigma-Aldrich). The tumors were measured every other day by electronic caliper, in two dimensions (length and width), and with the formula $v = (l \times w^2)/2$, where $v$ is the tumor volume, $l$ is the length, and $w$ is the width (the smaller of the two measurements). The drug schedule was 3 consecutive days a week (Tuesday to Thursday). For the chemotherapy combination in vivo model, TAK-981 and ifosfamide were administered as described in the legend of Fig. 8G. Ifosfamide was formulated in saline (10 mg/ml). The maximum total combined tumor volume permitted by the Institutional Animal Care and Use Committee of VCU (IACUC) is 2000 mm³ per mouse. This limit has not been exceeded in any of the animal studies of this manuscript that were conducted at VCU. Mice are typically housed under a 12-h light/12-h dark cycle, with ambient temperatures maintained between 68 and 75 °F (20-24 °C) and humidity around 45–65% and adhere to established guidelines and regulations to ensure animal welfare and the validity of research.

### Synovial sarcoma PDX model
PDX tumors were serially passaged in NOD scid gamma (NSG) mice (injected subcutaneously into each flank of 6–8 week old NOD/SCID gamma (NSG) male mice). Details about the schedule and dosage of the SS.PDX model can be found in the legend of Fig. 8C. For IHC staining, tumors were harvested 2 h following the last drug treatment. The maximum total combined tumor volume permitted by the Institutional Animal Care and Use Committee of VCU (IACUC) is 2000 mm³

per mouse. This limit has not been exceeded in the specific animal study at VCU.

### *hSS2* conditional mouse model
TATCre was injected at 8 days to initiate tumors in *hSS2* mice[92]. Littermate-controlled cohorts of *hSS2* mice were treated with 25 mg/kg TAK-981 or vehicle. Caliper measurements started when tumors became visible. Tumor volumes were calculated using the following formula: tumor volume = $(D \times d^2)/2$, in which $D$ and $d$ refer to the long and short tumor diameter, respectively. Treatment started once the tumors attained a size of ~0.5 cm³. The maximum total combined tumor volume permitted by the University of Utah animal care committee is 2000 mm³ per mouse. This limit has not been exceeded in the specific animal experiment. Standard vivarium conditions for mice at Huntsman Cancer Institute include 12-h light/12-h dark cycles with the temperature typically be maintained between 20 and 26 °C (68–79 °F) and the relative humidity generally at 45–50%.

### RNA-seq performed in *hSS2* tumors
Fresh frozen tumor tissue was disrupted using Tissue-Tearor (BioSpec). Total tumor RNA was obtained from snap frozen tumor samples using the Animal Tissues protocol from the RNeasy Kit (74104; Qiagen). Libraries were prepared with the Illumina TruSeq Stranded RNA Kit with Ribo-Zero Gold kit and sequenced on a HiSeq 2500 instrument for ~25 million reads per sample of 50 Cycle Single-Read Sequencing (version 4).

### IHC staining and calculation of the H-score
All tissues were fixed in 10% neutral buffered formalin for at least 5 days before paraffin embedding and sectioning. IHC staining was preformed similar to Jacobs et al.[44]. Cleaved Caspase 3 (CC3) (1:500; Cell Signaling 9664), ARID1A (1:500; GeneTex GTX129433) and SMARCE1 (1:1000; Abcam ab131328) stainings were performed with the Leica Bond RX autostainer using heat-induced epitope retrieval buffer 1 (Leica, Sodium Citrate, pH 6.0) for SMARCE1 or buffer 2 (Leica, EDTA pH 8.0) for ARID1A and CC3 for 20 min. For all three antibodies (CC3, ARID1A, and SMARCE1) a 20 min blocking (using Leica BioSciences Power Vision Universal IHC blocking diluent PV6123) was performed before the primary antibody incubation for 45 min. BAF47 (1:1500; 91735; Cell Signaling) and γ-H2AX (1:400; 9718; Cell Signaling) were also stained with the Leica Bond RX autostainer using heat-induced epitope retrieval buffer 2 (Leica, EDTA pH 8.0) for 20 min with no blocking and an antibody incubation for 15 min (γ-H2AX) or 30 min (BAF47). BRD9 (ab259839; Abcam) and GLTSCR1 (ab302712; Abcam) stainings were also performed on Leica Bond RX autostainer using Bond Polymer detection system (Leica biosystems DS9800). For both antibodies, the primary Ab dilution was 1:500, the incubation time 30 min, and EDTA based HIER buffer was used. All the IHC stainings were performed in the VCU Tissue and Data

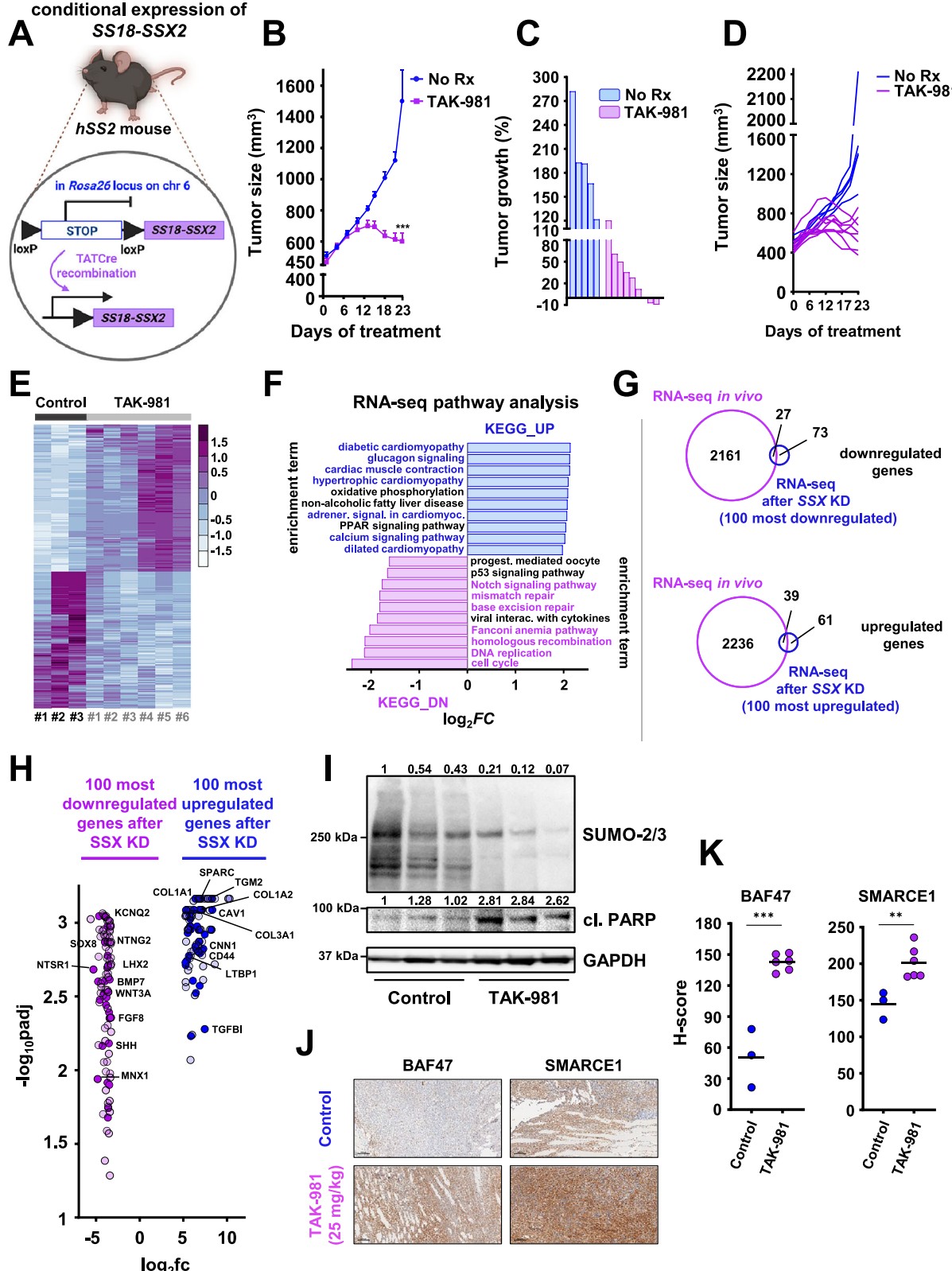

Acquisition and Analysis Core. Stained slides were then imaged on the Phenoimager HT (Akoyabiosciences) and scored. The H-score was calculated using QuPath[93]. Cells were categorized into weakly-, moderately-, and strongly stained thresholds using Qupath, in consultation with a pathologist. The "Positive Cell Detection" function in QuPath was subsequently used to calculate the final H-scores from these thresholds.

## PTMScan SUMO profiling of SS cells

$2 \times 10^8$ HS-SY-II cells per sample were grown in the appropriate number of 15 cm dishes (<80% confluency−this produces ~20−40 mg of total protein). The cells were treated with No Rx (one sample) and 100 nM of TAK-981 for 36 h (one sample) in 15 cm dishes. The medium was aspirated, and the adherent cells were washed with PBS. 10 ml of Urea Lysis Buffer (that contains the following components: 20 mM HEPES

**Fig. 9 | TAK-981 blocks tumor growth in a mouse model with conditional SS18::SSX expression. A** Schema of the design of the mouse experiment[92] (created in BioRender. Floros, K. (2025)). **B** hSS2 mice were treated with 25 mg/kg 3/w or vehicle (control) and tumors were monitored by caliper at least 3/w (*n* = 5 for control and *n* = 9 for the TAK-981 treatment cohort) (*n*: number of tumors). The data were presented as mean values + SEM. Unpaired two-tailed *t* test was performed and the *p* value for the comparison between control and TAK-981-treated tumors was calculated, *p* = 0.0002. **C** Waterfall plot of data from (**B**). **D** Individual tumor growth from the experiment shown in (**B**, **C**). **E** Heatmap of the RNA expression profiles of the 1000 most differentiated genes following TAK-981 treatment, after RNA-seq analysis in the indicated hSS2 tumors (*n* = 3 for control and n = 6 for TAK-981 treatment). **F** Pathway analysis of RNA-seq data from hSS2 tumors. **G** Venn diagrams of significant gene changes caused by TAK-981 in hSS2

mice and the 100 most up- or down-regulated genes in vitro following *SS18::SSX* knockdown in HS-SY-II cells by Banito et al.[22]. **H** Volcano plot of the 100 most down- or upregulated genes following *SS18::SSX* knockdown in vitro by Banito et al.[22] plotted with the commonly downregulated genes (dark purple dots) or commonly upregulated genes (dark blue dots) after RNA-seq in the hSS2 tumors. The *p* values were adjusted using a False Discovery Rate (FDR) multiple testing correction method. **I** Representative tumors from the control and the TAK-981-treated cohort were harvested 2–3 h after the last TAK-981 administration and tumor lysates were subjected to western blot analysis and probed for the indicated proteins. **J** Representative images of IHC analysis of the indicated antibodies for control and TAK-981-treated tumors from the hSS2 mouse experiment. Scale bars = 100 μm. **K** H-scores of staining with the indicated antibodies from (**J**). Exact *p* values and source data can be found in the Source Data.

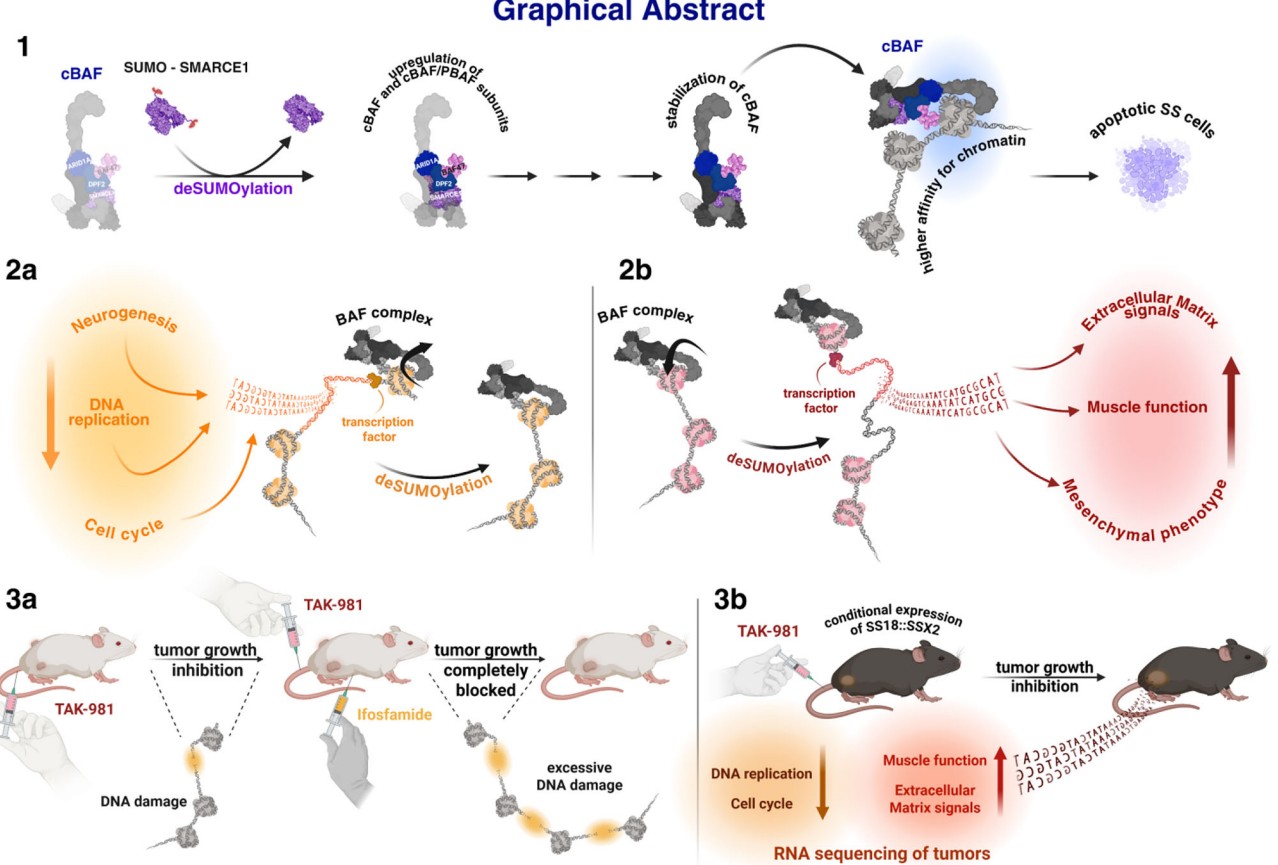

**Fig. 10 | Graphical abstract of the study.** Summary of the main results of the study. **1** In SS models, blocking SUMOylation prevents SMARCE1 degradation, therefore stabilizing cBAF complexes on chromatin and leading to redistribution of BAF complexes and subsequent cell death. **2a** Blocking of SUMOylation leads to disruption of the SS signature that primarily contains genes that encode homeobox transcription factors related to neurogenesis and other developmental processes and rewires the SS cells towards a mesenchymal phenotype that contains genes encoding extracellular matrix (ECM) proteins and proteins related to muscle

function (**2b**). **3a** TAK-981 sensitizes SS to cytotoxic chemotherapy (ifosfamide) in vivo by inducing DNA damage. **3b** TAK-981 blocks tumor growth in a mouse model with conditional SS18::SSX2 expression. RNA-seq analysis demonstrated pathways associated with DNA repair and the cell cycle were significantly downregulated in TAK-981-treated tumors. In contrast, pathways related to muscle function, and to ECM receptor interaction exhibited a marked upregulation in TAK-981 treated tumors. (created in BioRender. Floros, K. (2025)).

(pH 8.0), 9.0 M urea (PTMscan qualified, 60055S; Cell Signaling), 1 mM sodium orthovanadate, 2.5 mM sodium pyrophosphate and 1 mM ß-glycerol-phosphate) per 15 cm dish were added. The lysate from each dish (the cell lysate yield from 10 dishes was ~12 ml) was scraped and transferred into a 50 ml conical bottom tube. The 50 ml sample tubes were place on dry ice/ethanol for at least 30 min to freeze completely and shipped to Cell Signaling Technology for analysis.

Cells were sonicated and centrifuged to remove insoluble material. Protein content was determined by Bradford assay and equal protein quantities from all samples were used for analysis. Samples were reduced with DTT and alkylated with iodoacetamide. Samples were digested with wild type alpha-lytic protease (wALP, 33036; Cell Signaling), purified over C18 columns (Waters) and enriched using the PTMScan HS Ubiquitin/SUMO Remnant Motif (K-ε-GG) Kit (59322; Cell

Signaling) as previously described[94]. Samples were desalted over C18 tips prior to LC-MS/MS analysis.

LC-MS/MS analysis was performed using a Thermo Orbitrap Fusion™ Lumos™ Tribrid™ mass spectrometer[94,95] with replicate injections of each sample. Briefly, peptides were separated using a 50 cm × 100 μM PicoFrit capillary column packed with C18 reversed-phase resin and eluted with a 90-min linear gradient of acetonitrile in 0.125% formic acid delivered at 280 nl/min. Tandem mass spectra were collected in a data-dependent manner using a 3 s cycle time MS/MS method, a dynamic repeat count of one, and a repeat duration of 30 s. Real time recalibration of mass error was performed using lock mass[96] with a singly charged polysiloxane ion m/z = 371.101237. MS spectra were evaluated by Cell Signaling Technology using Comet and the GFY-Core platform (Harvard University)[97–99]. Searches were performed against the most recent update of the Uniprot *Homo sapiens* database with a mass accuracy of ±20 ppm for precursor ions and 0.02 Da for product ions. Results were filtered to a 1% peptide-level FDR with mass accuracy ±5ppm on precursor ions and presence of a K-GG residue for enriched samples. All quantitative results were generated using the Skyline software[100] to extract the integrated peak area of the corresponding peptide assignments. Accuracy of quantitative data was ensured by manual review in Skyline or in the ion chromatogram files.

## Co-immunoprecipitation

SS cells were seeded in 2 × 15 cm dishes per condition and transfected the next day with 7.5 μg of a SUMO2 vector (pEFIRESpuro-6His-SUMO2 (a gift from Dr Ronald Hay, University of Dundee))[101] per dish, using Lipofectamine 3000 Transfection Reagent (L3000015; Thermofisher Scientific). TAK-981 (100 nM) was added 36 h post-transfection for another 36 h. Nuclear lysates were prepared as above using the Pierce IP Lysis Buffer (87787; Thermofisher Scientific), supplemented with 0.5 M of N-ethylmaleimide (NEM); 500 μg of nuclear lysates were incubated each time with a proper anti-BAF57/SMARCE1 antibody (5 μg), or a normal rabbit IgG antibody (5 μg). Following the addition of 25 μL of 1:1 PBS: prewashed Protein A Sepharose CL-4B beads (cat. no. 17-0963 03; GE Healthcare Life Sciences) to the antibody/lysate mix, samples were incubated with rotating motion at 4°C overnight. Equal amounts of nuclear extracts (5% of immunoprecipitated protein) were prepared in parallel.

## SUMO detection by co-IP

The same protocol was used as for the co-IP and 1 μg of nuclear lysate per sample (in IP lysis buffer as described above) was incubated with 40 μl of Anti-SUMO-2/3 (ASM24-Beads; Cytoskeleton) (50% bead slurry) and equivalent volume of IgG control bead slurry (CIG01-Beads; Cytoskeleton), following manufacturer's instructions with rotating motion at 4°C, overnight. Equal amounts of nuclear extracts (5% of immunoprecipitated protein) were prepared in parallel.

## Nuclear extraction before density sedimentation

For Fig. 3E, an SS18::SSX-specific shRNA expression vector was provided by A. Banito and delivered by lentiviral transfection to the HS-SY-II cells as previously described (sh*REN*-control)[22] (see also "Vector Construction and Lentiviral Transduction"). The cells were cultured in 150 mm dishes, expanded according to assay requirements and treated with the indicated reagents (see Fig. 3D, E). Afterwards cells were scraped in cold PBS with phosphatase inhibitors and centrifuged for 5 min at 500 × g. The cells were washed with 5 ml ice-cold Buffer A (20 mM HEPES pH 8.0, 12 mM MgCl₂, 10 mM KCl, 0.25% NP-40, 0.5 mM DTT, and 2x protease inhibitor cocktail) incubated for 10 min on ice and homogenized using a douncer (Wheaton). The homogenate was centrifuged for 5 min at 5000 rpm and the supernatant was discarded. The pelleted nuclei were resuspended in Buffer C (20 mM HEPES pH 8.0, 25% glycerol, 1.5 mM MgCl₂, 420 mM KCl, 0.25% NP-40, 0.2 mM EDTA, 0.5 mM DTT, and 2x protease inhibitor cocktail) and

homogenized using a douncer. Nuclei were subjected to rotation for 30 min at 4 °C, then centrifuged for 30 min at 20,000 × g.

## Density sedimentation gradients

Nuclear extracts were diluted 1:1 with dilution buffer (20 mM HEPES pH 8.0, 1.5 mM MgCl₂, 0.2 mM EDTA, 0.5 mM DTT, and 2x protease inhibitor cocktail). The samples were loaded on the top of 11 ml 10–30% glycerol gradients containing 20 mM HEPES pH 8.0, 1.5 mM MgCl₂, 200 mM NaCl, 0.2 mM EDTA, 0.5 mM DTT, and 2x protease inhibitor cocktail. Tubes were loaded into SW41 rotor and centrifuged at 40,000 rpm for 20 h at 4 °C. 25 fractions were manually collected from the bottom of the gradient. Samples were run on an SDS-PAGE gel for western blot analysis.

## Differential salt extraction

The assay was performed as previously described[14]. Briefly, SYO-1, HS-SY-II and Yamato cells were grown under standard conditions. Following treatment with TAK-981, ~20 million cells per condition were collected. Then cells were suspended in elution buffer (50 mM Tris-HCl at pH 7.5, 1 mM EDTA, 0.1% NP40) supplemented with protease inhibitor mixture (Roche) and 1 mM PMSF, incubated on ice for 5 min, and centrifuged. Supernatant was collected, and the pellet was suspended in elution buffer with 150 mM NaCl. This process was repeated sequentially with increasing concentrations of NaCl to collect 0, 150, 300, 500, and 1000 mM NaCl soluble fractions. Each fraction, including the total (~4 × 10⁶ cells in elution buffer) and pellet (material remaining following 1000 mM NaCl extraction) fractions, was prepared in SDS and analyzed by immunoblotting.

## Single-cell gel electrophoresis (COMET)

1 × 10⁴ cells were plated in 24-well plate with 1 mL media, 1 day prior to harvesting. The next day, cells were trypsinized and resuspended in a mixture of 0.5% wt/vol low molecular weight agarose (50101; Lonza) and PBS at a ratio of 10:1. Suspension was immediately pipetted onto R&D Systems COMET Slides (4250-050-03) and allowed to dry for 30 min at 4 °C. Slides underwent lysis for 90 min at 4 °C in the dark (Lysis buffer: 10 mM Tris, 100 mM EDTA, 2.5 M NaCl, 1% TritonX100, 10% DMSO titrated to pH 10.0). Afterwards slides were placed in Alkaline buffer for 25 min at 4 °C in the dark (Alkaline buffer: 1 mM EDTA, 200 mM NaOH, pH > 13.0). Slides were transferred to an agarose gel electrophoresis box filled with additional alkaline buffer. Electrophoresis was performed at 25 V for 20 min at room temperature (RT) in the dark. Slides were then washed twice in dd (double distilled) H₂O for 5 min at RT and then placed in neutralization buffer for 20 min at RT in the dark (Neutralization buffer: 400 mM Tris-HCl titrated to pH 7.5). Neutralized slides were then left to dry at 37 °C in the dark. Dried slides were stained with DAPI (1:10,000 in dd H₂O) for 15 min at RT then washed twice with dd H₂O for 5 min. Stained and rinsed slides were left to dry overnight. Slides were imaged using the Keyence imaging system at 20x with >5 images taken per replicate. Quantification was achieved using the CaspLab Comet Assay Analyzer software[57]. For each cell/comet, the length and intensity of the head and the tail were measured. The DNA damage was quantified as tail moment. The tail moment was calculated as the length of the tail x tail DNA %. The tail DNA % was measured as tail DNA intensity/total cell DNA intensity × 100. 40 cells were counted for each image (except for the combination group where only 20 cells were counted due to low cell density). The data is presented as mean tail moment +SEM.

## Statistics and reproducibility

Figure 1C: This experiment has been repeated independently three times with similar results. Figure 1H: This experiment has been repeated independently three times with similar results. Figure 2B: This experiment has been repeated independently three times with similar results. Figure 2C: This experiment has been repeated independently

three times with similar results. Figure 2D: This experiment has been repeated independently three times with similar results. Figure 2E: This experiment has been repeated independently three times with similar results. Figure 2F: This experiment has been repeated independently four times with similar results. Figure 2H: This experiment has been repeated independently four times with similar results. Figure 3A: This experiment has been repeated independently four times with similar results. Figure 3D: This experiment has been repeated independently three times with similar results. Figure 3E: This experiment has been repeated independently three times with similar results. Figure 4A: This experiment has been repeated independently three times with similar results. Figure 9I: This experiment has been repeated independently four times with similar results. Supplementary Fig. 2A: This experiment has been repeated independently four times with similar results. Supplementary Fig. 2A: This experiment has been repeated independently four times with similar results. Supplementary Fig. 2C: This experiment has been repeated independently three times with similar results. Supplementary Fig. 2D: This experiment has been repeated independently three times with similar results. Supplementary Fig. 2E: This experiment has been repeated independently three times with similar results. For all experiments conducted in the present study no statistical method was used to predetermine sample size and no data were excluded from the analyses. In all the experiments, samples were allocated to different groups using random assignment to ensure fairness and to minimize bias. For the mouse experiments, when tumors reached the indicated size, the tumor-bearing mice were randomized to a no treatment/control group, and the indicated TAK-981 treatment groups. Mice were randomized according to their tumor sizes to ensure that the experimental groups are as comparable as possible at the start of the study. At least five tumors per cohort were enrolled in each experiment to gain meaningful insights.

All the details/methods that were used to calculate the statistical significance for each figure were added to the corresponding figure legends. Functional enrichment of genes associated with ChIP-seq or ATAC-seq peaks (False Discovery Rate FDR < 0.3) was performed using Gene set enrichment analyses (MSigDB gene signatures, clusterProfiler v.4.2.16 R package) or GREAT (Genomic Regions Enrichment of Annotations Tool, rGREAT v.2.6.0 R package). Pairwise Venn diagram overlap significance was calculated using Chi-Square test under the assumption of 17,611 expressed transcripts. For all calculated $p$ values throughout the entire manuscript: differences were considered statistically significant if $p < 0.05$. * $p < 0.05$, ** $p < 0.01$, *** $p < 0.001$, **** $p < 0.0001$.

**Reporting summary**

Further information on research design is available in the Nature Portfolio Reporting Summary linked to this article.

## Data availability

All genomics data generated by this study is publicly available in the GEO (Gene Expression Omnibus) database under accession code: GSE266135. The mass spectrometry proteomics data has been deposited to the ProteomeXchange Consortium via the PRIDE[102] partner repository and is publicly available with the dataset identifier: PXD059936. The remaining data are available within the Article, Supplementary Information or Source Data file. Sequences of the qPCR primers used in the current study have been provided as separate Supplementary Data. Source data are provided with this paper.

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

## Acknowledgements

We thank Nicolo Riggi, Gaylor Boulay, Adam Crystal and Ronald Hay for invaluable support, feedback and conversation. This work was supported by the National Institutes of Health/National Cancer Institute (NIH/NCI) grant number 1R01CA272710-01A1 (to A.C.F.), a Department of Defense Rare Cancer Research Program grant, award number HT9425-23-1-1017 (to A.C.F.), and a Department of Defense Rare Cancer Research Program grant, award number W81XWH-22-1-0938 (to S.K.R.). A.C.F. is also supported by a Natalie N. and John R. Congdon endowment and the Meryl and Charles Witmer Charitable Foundation. A.B. received funding from the European Research Council (ERC) under the European Union's Horizon 2020 research and innovation program (grant agreement number 805338) and from the NIH/NCI U54CA231652. R.L. was supported by an NIH/NIAID grant (R01AI141410), Research Scholar Grant (134703-RSG-20–054-01-MPC) from the American Cancer Society and

start-up funds from the University of Pittsburgh Medical Center (UPMC) Hillman Cancer Center. T.S. was supported by NIH grant R01 AI51051. Data was generated in the Genome Sequencing Facility, which is supported by UT Health San Antonio, NIH-NCI P30 CA054174 (Cancer Center at UT Health San Antonio), NIH Shared Instrument grant 1S10 OD030311-01 (S10 grant), and CPRIT Core Facility Award (RP220662). Z.L. is supported by NIH NCI R50 CA265339. Services and products in support of the current research project were also generated by the Virginia Commonwealth University Cancer Mouse Models Core Laboratory and the Tissue and Data Acquisition and Analysis Core Laboratory, both supported, in part, with funding to the Massey Cancer Center from NIH-NCI Cancer Center Support Grant P30 CA016059.

## Author contributions

Conceptualization, K.V.F., S.K.R. and A.C.F.; formal analysis, K.V.F., C.K.F., J.L., K.Z., J.L.R., R.K., B.H., V.K., N.H., S.S., M.M.I., K.S.-F., L.L., A.S., K.M.D., A.J., E.I.A., Y.X., R.D.H., J.M.S., M.S., A.C.D., M.R.L., M.R.H., N.T., T.K., V.A.K., F.G.S., B.R.B., Z.L., S.A.B., A.M.S., J.P.L., M.H.M., K.V., R.L., A.B., A.P., J.E.K., T.S., M.G.D., K.B.J., S.K.R., A.C.F.; funding acquisition, K.B.J., S.K.R., T.S., A.C.F.; investigation, K.V.F., R.L., A.B., M.L.N., D.P., J.E.K., H.E., T.S., M.G.D., K.B.J., S.K.R., A.C.F.; resources, K.V.F., M.G.D., K.B.J., S.K.R., A.C.F.; supervision, A.C.F.; writing—original draft, K.V.F., A.C.F.; writing—review and editing, K.V.F., K.B.J., S.K.R., J.E.K., A.B., M.G.D., A.C.F.

## Competing interests

A.C.F. is a consultant and equity holder in Treeline Biosciences and has previously served as a scientific advisor for AbbVie and has received research funding from IDP Pharma. K.V. receives support from Astra-Zeneca. R.L. is an inventor of a provisional patent on targeted killing of EBV-positive cancer cells by CRISPR/dCas9-mediated EBV reactivation. S.A.B. is Consultant for Caris Lifescience, has received honoraria from SpringWorks for an educational lecture and is senior clinical program leader at Boehringer-Ingelheim. The authors declare that these listed activities have no relationship to the present study.

## Additional information

Konstantinos V. Floros [1,2], Carter K. Fairchild Jr.[1,3,29], Jinxiu Li[4,29], Kun Zhang[1,2], Jane L. Roberts[1,2], Richard Kurupi[1,5], Durga Paudel[1,2], Yanli Xing[1,2], Bin Hu[6], Vita Kraskauskiene[6], Nayyerehalsadat Hosseini[6], Shanwei Shen[6], Melissa M. Inge[7,8], Kyllie Smith-Fry[4], Li Li[4], Afroditi Sotiriou [9,10,11], Krista M. Dalton[1,2], Asha Jose[1,12], Elsamani I. Abdelfadiel[1,2], Ronald D. Hill[1,2], Jamie M. Slaughter[1,2], Mayuri Shende[6], Madelyn R. Lorenz[6], Noritaka Tanaka[13], Taisuke Kajino [13], Mary L. Nelson [4], Mandy R. Hinojosa[14,15], Victor A. Kehinde [1,2], Benjamin R. Belvin[1], Febri G. Sugiokto [16,17,18], Zhao Lai[14,15], Alexandros C. Dimopoulos [19], Sosipatros A. Boikos[20], Angeliki M. Stamatouli[21], Janina P. Lewis[1,22,23], Masoud H. Manjili [23], Hiromichi Ebi [12,24], Kristoffer Valerie[25], Renfeng Li [16,17,18], Andrew Poklepovic[26], Jennifer E. Koblinski[6], Trevor Siggers [7,8,27], Ana Banito [9,10], Mikhail G. Dozmorov[28], Kevin B. Jones [4] ✉, Senthil K. Radhakrishnan [6] ✉ & Anthony C. Faber [1,2] ✉

[1]VCU Philips Institute, Virginia Commonwealth University School of Dentistry and Massey Comprehensive Cancer Center, Richmond, VA, USA. [2]Department of Pediatrics, Virginia Commonwealth University School of Medicine, Richmond, VA, USA. [3]Pauley Heart Center, Division of Cardiology, Department of Internal Medicine, Virginia Commonwealth University, Richmond, VA, USA. [4]Departments of Orthopaedics and Oncological Sciences, Huntsman Cancer Institute, University of Utah, Salt Lake City, UT, USA. [5]Division of Oncology, Department of Internal Medicine, Washington University School of Medicine, St. Louis, MO, USA. [6]Department of Pathology, Virginia Commonwealth University and Massey Comprehensive Cancer Center, Richmond, VA, USA. [7]Department of Biology, Boston University, Boston, MA, USA. [8]Biological Design Center, Boston University, Boston, MA, USA. [9]Soft Tissue Sarcoma Research Group, Hopp Children's Cancer Center, Heidelberg (KiTZ), German Cancer Research Center (DKFZ), Heidelberg, Germany. [10]National Center for Tumor Diseases (NCT), NCT Heidelberg, A Partnership between DKFZ and Heidelberg University Hospital, Heidelberg, Germany. [11]Faculty of Biosciences, University of Heidelberg, Heidelberg, Germany. [12]Renal Section, Department of Medicine, Boston Medical Center, Boston University School of Medicine, Boston, MA, USA. [13]Division of Molecular Therapeutics, Aichi Cancer Center Research Institute, Nagoya, Japan. [14]Greehey Children's Cancer Research Institute, University of Texas Health San Antonio, San Antonio, TX, USA. [15]Department of Molecular Medicine, University of Texas Health San Antonio, San Antonio, TX, USA. [16]Program in Microbiology and Immunology, University of Pittsburgh, Pittsburgh, PA, USA. [17]Department of Microbiology and Molecular Genetics, University of Pittsburgh, Pittsburgh, PA, USA. [18]Cancer Virology Program, Hillman Cancer Center, University of Pittsburgh Medical Center, Pittsburgh, PA, USA. [19]Department of Informatics and Telematics, School of Digital Technology, Harokopio University, Athens, Greece. [20]Georgetown Lombardi Comprehensive Cancer Center,

Washington, DC, USA. [21]Division of Endocrinology, Diabetes, and Metabolism, Department of Internal Medicine, Virginia Commonwealth University School of Medicine, Richmond, VA, USA. [22]Department of Biochemistry and Molecular Biology, and Massey Comprehensive Cancer Center, Virginia Commonwealth University, Richmond, VA, USA. [23]Department of Microbiology & Immunology and Massey Comprehensive Cancer Center, Richmond, VA, USA. [24]Division of Advanced Cancer Therapeutics, Nagoya University Graduate School of Medicine, Nagoya, Aichi, Japan. [25]Department of Radiation Oncology and Massey Comprehensive Cancer Center, Virginia Commonwealth University, Richmond, VA, USA. [26]Department of Internal Medicine, Division of Oncology, Massey Comprehensive Cancer Center, Virginia Commonwealth University, Richmond, VA, USA. [27]Bioinformatics Program, Boston University, Boston, MA, USA. [28]Department of Biostatistics, Virginia Commonwealth University, Richmond, VA, USA. [29]These authors contributed equally: Carter K. Fairchild Jr., Jinxiu Li. ✉e-mail: kevin.jones@hci.utah.edu; senthil.radhakrishnan@vcuhealth.org; acfaber@vcu.edu

