## [Transparent Peer Review file · Nature Communications]

Targeting SUMOylation promotes cBAF complex stabilization and disruption of the SS18::SSX transcriptome in Synovial Sarcoma

Corresponding Author: Dr Anthony Faber

Version 0:

Reviewer comments:

Reviewer #1

(Remarks to the Author)

The authors investigate synovial sarcoma, which is driven by the SS18:SSX fusion oncoprotein and is currently challenging to treat. The fusion oncoprotein changes chromatin remodeling BAF complexes leading to the degradation of the canonical BAF (cBAF) complex. The authors found that the fusion oncoprotein activates sumoylation and that synovial sarcoma is sensitive to the SUMO E1 inhibitor TAK-981. The authors show that cBAF subunit SMARCE1 is sumoylated in synovial sarcoma and degraded. TAK-91 synergizes with chemotherapy in a mouse model. Overall, this is an interesting study, but several aspects remain to be strengthened.

1. Mechanistically, it is unclear how the SS18:SSX fusion oncoprotein activates the sumoylation program. The fusion oncoprotein increases transcription of SUMO genes and sumoylation machinery components, but the molecular mechanism is not clear. Please investigate the molecular mechanism underlying this change in transcriptional program.
2. The authors need to strengthen SUMO modification of cBAF components using SUMO pulldown-Western approaches, because the employed mass spectrometry approach can easily lead to false positives as it can misidentify ubiquitinated proteins.
3. The authors should solidify the SUMO—SUMO Targeted Ubiquitin Ligase (STUbL)-Ubiquitin-proteasome axis of the study by including the specific ubiquitin E1 inhibitor TAK-243 and by including proteasome inhibitors in their experiments, particularly in Figures 2C and 3A-D.
4. The authors investigate the STUbL RNF4, but ignore other STUbLs including the recently identified STUbL TOPORS (Liu et al. 2024 Nat Struct Mol Biol doi: 10.1038/s41594-024-01294-7). Please include knockdown of TOPORS and double knockdown of RNF4 and TOPORS in Figure 2C.
5. Does the drug ifosfamide alter sumoylation and degradation of the cBAF complex? Please address by SUMO pull down-Western approach.
6. The authors need to verify RNF4 knockdown by immunoblotting.
7. The authors need to submit the mass spectrometry data to the PRIDE repository and provide access.

Reviewer #2

(Remarks to the Author)

In this manuscript titled "Targeting of SUMOylation leads to cBAF complex stabilization and disruption of the SS18::SSX transcriptome in Synovial Sarcoma", the authors (Floros et al.) strive to demonstrate the following observations:

- 1) The synovial sarcoma (SS) fusion SS18-SSX activates the sumoylation pathway as part of its oncogenic program.

- 2) Inhibiting sumoylation with the TAK-981 compound suppresses SS18-SSX programs and SS cells and tumor growth.
- 3) Part of TAK-981 mechanism is desumoylation and thereby stabilization of the BAF component SMARCE1, which restores normal cBAF complex on chromatin, thus destabilizing the oncogenic ncBAF complex formed by SS18-SSX in the sarcoma cells.
- 4) SS cell and tumor death by TAK-981 could be the result of DNA damage caused by the drug.
- 5) TAK-981 synergizes with SS standard therapeutic agents and therefore is an appropriate option for SS combination therapy.

To support their findings, the authors generated 7 CHIP-seq screens in 2 SS cell lines, 2 ATAC-seq analyses, 4 RNA-seq studies, and utilized 6 different SS cell lines, a xenograft, a PDX, and a genetic SS tumor models.

This is an extensive multidisciplinary study that presents an alternative therapy for an up to now incurable malignancy. The authors convincingly demonstrate an involvement of SS18-SSX in protein sumoylation, and their synergy and in vivo modeling data are strong. However, the biochemical evidence for direct SMARCE1 sumoylation by SS18-SSX and desumoylation by TAK-981, and the important role it appears to play in shifting BAF complexes need to be strengthened.

General concerns:

- Add quantification of relative changes in western blots: Figs.1C, S2A, S2C, 3A, 8I
- Add statistics to all relevant graphs and include details in the legends, not in the Methods section, or preferably convert to histograms or dot plots or box plots with individual data points: Figs.1D, 1F, S3J, S3L, 6A, 7E, 8K
- Manuscript needs a detailed Statistics section
- Please make sure that all the panel labels correspond, in the text, figure legends and figures.

Major concerns:

- Fig.S2A: discuss the apparent reduction in SS18-SSX levels by TAK-981 in the inducible Yamato cells. Is there a feedback loop between SS18-SSX expression and sumoylation?
- Fig.1E: explain why crystal violet staining was chosen over a quantifiable cell viability assay in the CRISPR experiment.
- Fig.1E: discuss why SMARCE1 levels and SS18-SSX levels in SS cells were not measured upon depletion of SUMO components by CRISPR.
- Fig.2B: It is important to confirm that the increase in SMARCE1 levels is post-transcriptional with qRT-PCR in the TAK-981 treated cells.
- Fig.2C: the "sumo-SMARCE1" band is weak and inconclusive. It could just be a non-specific background band. After identification of SMARCE1 in the PTMscan, the next thing would be to verify whether SMARCE1 is indeed sumoylated in SS. This could be shown with more than one method, for example:
 - IP/western: IP with sumo2 and or sumo3 Ab, then blot with SMARCE1 Ab, using very sensitive ECL reagents such as super signal ATTO
 - and/or the reverse with IP SMARCE1 and blot for SUMO
 - if endogenous levels are still undetectable, then accumulation of SUMO species with a reagent such as TAK-243, MG132, or ALLN, etc. might enhance SUMO-SMARCE1 detection.
 - For general significance, it is advisable to test for SMARCE1 sumoylation in all 6 SS cell lines.
 - one could also express exogenous sumo and test for SMARCE1 modification etc.
- Fig.3A: the authors assume that, based on previous studies, increased SMARCE1 is the cause of enhanced ARID1A and BAF47 expression. However, it is essential to rule out that, like SMARCE1, this upregulation of ARID1A and BAF47 is due to desumoylation by TAK-981 as well, even though they were not identified in the PTMscan. This could be tested in the IP/western described above.
- Fig.3B, 3C, 3D: adequate glycerol gradients and salt extractions. Was SS18-SSX left out for a reason?
- Optional: it would add to the findings if co/IP experiments were conducted to test whether any changes in the composition of the cBAF, ncBAF, and PBAF complexes had occurred in the TAK-981 treated SS cells.

Minor concerns:

- Methods: why are Miapaca-2, Panc-1, RPE.1, and SW982 cells included?
Why is the XP antibody listed?

- Introduction-2d paragraph, unclear first sentence. Suggest replacing with: "The pathognomonic chromosomal rearrangement in synovial sarcoma fuses the DNA segment coding for the C-terminal 78 amino acids of SSX1, SSX2, or SSX4, all located on the X Chromosome, to the region coding for amino acids 1–379 of the SS18 gene on chromosome 18, to generate SS18::SSX1, SS18::SSX2, or SS18::SSX4 fusion, each capable of driving SS (Clark et al., 1994)".

- Introduction-3d paragraph- Suggest changing to: "The major BAF complexes in most cells include canonical BAF (cBAF, also referred to as BAF), polybromo-associated BAF (PBAF), and non-canonical or GLTSCR1-containing BAF (ncBAF or GBAF).

- Supplemental Fig.1 Legend: it is "this analysis" or "these analyses". Also, the Schema is panel F)

- Results: The binding of BAF complexes to chromatin is responsible for many of the observed expression changes of the SS transcriptome after blocking SUMOylation
This section text and figures are hard to follow, it would gain from clearer writing and better organization of the results description and figure layout in a systemic fashion. it should be specified at the beginning that the SMARCA4 CHIP-seq was conducted in HS-SY-II cells and the SS18-SSX, KDM2B, and H3K27Ac CHIP-seqs in HS-SY-II and SYO-1 cells etc... Also, the use of imported RNA-seq data and the ones generated by the authors should be clearly explained and sorted.

In the same section, is it meant to be: "Additionally, 49 out of the 155 genes that had enhanced SMARCA4 signals also had enhanced SS18-SSX binding?"

- Results: TAK-981 has activity in synovial sarcoma in vitro and in vivo by inducing DNA damage

Fig.S7F in the text is S7C
Fig. S7C, S7D and S7E in the text are Fig. S7D, S7E, and S7F

The corresponding legend needs same corrections

- Fig.S9B: the comet assay result needs some quantification and statistics. How many tumors were looked at?

- Include reference for the hSS2 mice in the text.

Reviewer #3

(Remarks to the Author)
Comments to the Authors

In this study, the authors aim to define how SS18-SSX fusion protein can activate the SUMOylation program, rendering these tumors sensitive to a small molecules SAE1/2 inhibitor, TAK-981. They show that TAK-981 deSUMOylates SMARCE1 that they claim then restabilizing cBAF and reduces ncBAF on chromatin. However, the major issue is that the work does not convince the reader that TAK-981 impact on SMARCE1 and cBAF (or any chromatin remodeler/regulator) is the cause of its mechanisms. Rather, it would appear clear that TAK-981 will impact MANY SUMOylation events on many cellular proteins and reduce viability of SS (and other cell) lines in this manner. It is also impossible to conclude from the data presented that besides overt inhibition of many pathways in highly proliferative cancer cells lines like SS that the mechanism is via cBAF/ncBAF as the authors claim and hypothesize, leading to significantly reduced enthusiasm for the study by this reviewer.

Major comments:

Knockdown of the SS18-SSX fusion, in many other studies, completely reorganizes activated and repressed regions of chromatin, by mechanisms that have been well described in the literature (Kadoch lab many paper and Ana Banito lab, Kevin Jones lab, others). Many genes, including those involved in protein degradation, are both controlled by SS18-SSX on chromatin and enable protein level control. Crabtree and Kadoch even show in 2013 (Kadoch, Crabtree Cell 2013) that fixing cBAF such as by forced overexpression of BAF47 is not able to reduce or combat the effects of SS18-SSX because of likely affinity changes that I noted are also shown at biochemical and biophysical level in McBride et al NSMB 2020 paper which the authors cite.

In Figure 2, which is also very sparse, the authors identify SMARCE1 but don't show blots in these experiments for HS-SY-II or Yamato cell lines for any other cBAF or ncBAF subunit component. Where are the protein levels for all the other subunits in this experiment? This makes it very challenging for the reader to interpret; just because the mass spectrometric shows one subunit, it does not mean that that is the only subunit potentially having impact. Also, a time course set of experiments with TAK-981 for total protein as well as RNA of each BAF subunit would be needed to better interpret the data.

In the Figure 3, The authors show that several subunits are “stabilized” at the protein level. They would likely be stabilized in non-SS cells as well but these data are not shown (where are gradients and blots for many other cancer cell lines or normal cells? I anticipate that most proteins would increase in levels. One can see also that PBAF subunits (which have been shown to not be impacted by the fusion and do not contain the SS18-SSX fusion) also are majorly stabilized by TAK-981. Similarly BRD9 of ncBAF complex is stabilized and there looks to be even more of it with the TAK-981 treatment. Many proteins including probably chromatin repressive complexes would show all the same.

The panels in Fig 3D do not tell us anything except further that levels are increased of everything... if this is an attempt to show changed chromatin stability, this does not show it; this is just at the mercy of antibody detection levels/affinities and overall protein levels.

The authors then look at gene expression in the Figure 4. Here again, the figures as presented do not help the case—clearly, TAK-981 does a lot of other things as does KD of fusion. If TAK-981 really impacted the complexes, one would expect this to look much more like SS18-SSX KD and likely other conditions like BRD9 knockout or knockdown for ncBAF.

In Fig 5, the authors perform ChIP-seq but there is no comprehensive analysis of these data shown—no heatmaps, few volcano plots, few tracks over important genes, and it is not clear if indeed in their own data, there is really any alteration in cBAF and ncBAF binding to SUMOylation genes and SUMOylation target genes. Overlap in 5E is very limited and numbers of peaks are low altogether....

There are just as possible other mechanisms of degradation at play for SS18-SSX complexes, including very standard ones (i.e. even MG-132 has been show similar results), especially for proteins that cannot assemble into complexes because of fusion protein and what it structurally changes in complex.

The fact that SMARCE1 and BAF47 levels are so much more markedly increased by the combination of TAK-981 and ifosfamide (and even ifosfamide alone!), i.e. in panel 7H, just further show that this is a global cell poisoning mechanism. If the authors had stained for BRD9 or BIRCA as ncBAF members, they would also likely show the same, indicating that ncBAF is not decreased and ncBAF-caused transcription is not the main mechanism that is attenuated. Evaluation of other chromatin proteins and protein complexes would show similar, given these data. In fact, nowhere in this paper do the authors show that BRD9 is reduced in protein levels, and hence the authors conclusions that their TAK-981 “shifts away from SS18-SSX ncbAF-driven transcription” is just not well founded from the data. There is a difference of “ncBAF transcriptome” and “genes that are on in cancer cells” and nowhere in here do the authors specifically show any changes to ncBAF on chromatin or genes defined to be altered by BRD9 degradation or KD/KO.

If the authors were to ever convince that SUMOylation is a major factor here for these remodelling complexes then a screen would be needed (i.e. to treat cells, synovial sarcoma cells and others including normal cells without fusion) with TAK-981 and identify genes needed for TAK-981-induced killing.

Version 1:

Reviewer comments:

Reviewer #1

(Remarks to the Author)

The authors have thoroughly revised their paper and my concerns have been properly addressed.

Reviewer #2

(Remarks to the Author)

Congratulations to the authors for significantly strengthening their study.

The notable modifications are:

- 1- The mechanistic/biochemical studies to demonstrate sumoylation of the BAF components by the SS18-SSX fusion and its effect on SS tumor growth, thereby rendering this process an appropriate target for SS therapy.
- 2- Including the necessary statistics and quantifications to confirm the significance of the findings
- 3- Clarifying several points of confusion found in the initial version of the manuscript

With diverse sound high throughput gene expression and genomics approaches and by using in vitro and preclinical SS in vivo models, the authors have supported their claim regarding the potential benefit of including desumoylating agents to SS drug regimens.

Reviewer #4

(Remarks to the Author)

Synovial sarcoma is a soft tissue sarcoma with dismal outcome upon metastasis that occurs in children. There has been no major therapeutic breakthroughs over the last decades. Thus, new treatment modalities are urgently needed. Floros et al provide data that indicates the SUMOylation inhibitor TAK-981 has promising anti-SS efficacy primarily by deSUMOylating SMARCE1 subunit of the canonical BAF complex (cBAF). The authors have performed extensive biochemical characterization of the impact of TAK-981 treatment in vitro and in vivo as well as genetic manipulations of key intermediary genes in the sumoylation and BAF complex.

While the findings are highly intriguing, there are several major concerns that dampen overall enthusiasm for the study. The fundamental premise of the work originated from interrogation of the genome scale shRNA knockdown dependency scores generated from the DepMap portal which suggested the sumoylation pathway is a top dependency in SS. However, the authors failed to consider that key components of the sumoylation pathway including SAE1 (the sumo activating enzyme and target of TAK-981), UBE2I and SUMO2 are common essential genes as documented by DepMap CRISPR knockout dependency study. The study has not addressed the fact that the SUMO pathway, and more relevant for the present study the target of the compound the study is almost completely dependent on, is an essential gene with pleiotropic effects. While the biochemical data regarding sumoylation of SMARCE1 is solid and convincing, the authors have not entertained other proteins in their sumoylation proteomic study such as HSP90, tubulin (table S1) that are more likely to underlie the cytotoxic phenotype of TAK-981. In fact, these targets are also most likely to explain the essentiality of the sumoylation pathway and SAE1 across the DepMap cell lines. The absence of a strong rescue experiment using SMARCE1 overexpression, knockout and complementation or similar assays are additional weak points. Taken together, the study does not convincingly show that the deSUMOylation of SMARCE1 is the primary mechanism of cytotoxicity of TAK-981.

Other concerns

RNF4 and TOPORS have been suggested to mediate sumo dependent ubiquitination and their knockdown was as effective as TAK-981 in stabilizing SMARCE1. Does this knockdown phenocopy TAK-981 in its cytotoxicity?

Fig 3E does not convincingly show TAK-981 administration with or without shSSX shifts the BAF complex at all.

Fig 4A, while the authors state that only components of the BAF complex show alteration in salt extraction, ARID2 (PBAF subunit) clearly shows very similar behavior to the BAF components in SYO-1 cell line. In HS-SY-II and Yamato cell lines, the BAF complex shows inconsistent and unimpressive salt extraction change by TAK-981.

Fig 7: the BAF complex and its association with enhancers has been extensively shown to be highly relevant for tumorigenesis. Thus, it would be highly desirable to plot average peak intensities and heatmaps centered around peaks in addition to TSS centered plots.

Version 2:

Reviewer comments:

Reviewer #4

(Remarks to the Author)

The authors have addressed the key concerns raised.

Overview: We have added a substantial number of new figures: Fig. 1D, 2B, 2C, 2D, 2E, 2F, 2G, 2H, 3A, 3B, 3C, 3D, 3E, 3F, 3G, 3H, 4B, 4C, 4D, 4E, 10, new Sup. Fig. S2D, S2E, and S2G. Furthermore, new data have been added to the original figures: 7H (new Fig. 8H), 7I (new Fig. 8I), S9B (new Fig. S9B and S9C) and S9G (new Fig. S9H).

Please find below our response to the reviewer's comments:

Reviewer #1 – SUMOylation, cancer

The authors investigate synovial sarcoma, which is driven by the SS18:SSX fusion oncoprotein and is currently challenging to treat. The fusion oncoprotein changes chromatin remodeling BAF complexes leading to the degradation of the canonical BAF (cBAF) complex. The authors found that the fusion oncoprotein activates sumoylation and that synovial sarcoma is sensitive to the SUMO E1 inhibitor TAK-981. The authors show that cBAF subunit SMARCE1 is sumoylated in synovial sarcoma and degraded. TAK-91 synergizes with chemotherapy in a mouse model. Overall, this is an interesting study, but several aspects remain to be strengthened.

1. Mechanistically, it is unclear how the SS18:SSX fusion oncoprotein activates the sumoylation program. The fusion oncoprotein increases transcription of SUMO genes and sumoylation machinery components, but the molecular mechanism is not clear. Please investigate the molecular mechanism underlying this change in transcriptional program.

Answer: We thank the reviewer for prompting us to investigate the mechanism of activation of the SUMOylation program by the SS18::SSX fusion protein. Based on the data in the original manuscript, we hypothesized that the fusion protein acts as a transcription modifier to induce the expression of the proteins of the SUMOylation machinery. To investigate this possibility, we interrogated the SS18::SSX ChIP seq data in the HS-SY-II cells (other data from this experiment is being presented in Fig. 6 and Fig. S4). Here, we found that SS18::SSX binds at the promoter of the SUMO components *SUMO2*, *SUMO3*, *PIAS1*, and *UBE2I* (encodes UBC9 protein) (new Fig. 1D). This new data links the increase in transcript levels (presented in Fig. 1B) to SS18::SSX at the *SUMO2*, *SUMO3*, *PIAS1*, and *UBE2I* loci. We appreciate the reviewer prompting this investigation.

2. The authors need to strengthen SUMO modification of cBAF components using SUMO pulldown-Western approaches, because the employed mass spectrometry approach can easily lead to false positives as it can misidentify ubiquitinated proteins.

Answer: We thank the reviewer for this important point. To validate the mass spectrometry data, following the reviewer's suggestion, we have now performed pull-down western assays after transiently overexpressing SUMO2 in HS-SY-II and Yamato

cells. Anti-SUMO-2/3 antibody conjugated beads were used for the immunoprecipitation assay (new Fig. 2B and 2D). By pulling down SMARCE1 (antibody ab137081, Abcam), we were able to confirm the SUMOylation of SMARCE1. (new Fig. 2C and new Fig. 2E), supporting our hypothesis of the critical role of the deSUMOylation of SMARCE1 in stabilizing the cBAF complex.

In addition, by pulling down SUMO2/3 in this same system, SUMOylated SMARCE1 (new Fig. 2B and 2D) and BAF47 (new Fig. S2D) were identified (upper bands) along with unmodified SMARCE1, ARID1A and BAF47 binding to SUMOylated proteins or free SUMO molecules (lower band), due to the potential SUMO Interacting Motifs (SIMs) they carry (please see also new Fig. S2E for potential SIMs in the proteins of interest, SMARCE1, ARID1A and BAF47). As we could not detect SUMO-conjugated ARID1A, another interpretation of these data is that ARID1A binds to SUMOylated proteins but does not get SUMOylated itself. As the SUMOylated SMARCE1 and BAF47 proteins appear at the proper MWs in our SUMO IP western blots in both cell lines, this again confirms the SMARCE1 mass spectrometry data and SMARCE1 pulldown assay, as well as indicates that there may be additional BAF proteins other than SMARCE1 that can be SUMOylated, at least when SUMO is exogenously expressed and its levels exceed those of the native conditions (as detected by mass spec, Fig. 2A and Sup. Table 1). We have included this possibility in the discussion.

3. The authors should solidify the SUMO—SUMO Targeted Ubiquitin Ligase (STUbL)-Ubiquitin-proteasome axis of the study by including the specific ubiquitin E1 inhibitor TAK-243 and by including proteasome inhibitors in their experiments, particularly in Figures 2C and 3A-D.

Answer: As suggested by the reviewer, we repeated the density sedimentation assay in the SYO.1 cells (old Fig. 3B) and included treatments with both the ubiquitin inhibitor TAK-243 and the proteasome inhibitor, bortezomib (new Fig. 3D), and investigated the alterations in the expression of the BAF subunits in the corresponding cBAF, ncBAF and PBAF complexes. The addition of both bortezomib or TAK-243 leads to an upshift of the cBAF complex members (ARID1A, BAF47 and SMARCE1), partially resembling the changes detected after TAK-981 treatment. The density of the western blot bands was quantified, and the above changes were graphically depicted in new Fig. 3F and new Fig. S2F. These results support our proposed mechanism of SMARCE1 deSUMOylation leading to cBAF complex stabilization and are now described in the results section of the manuscript (highlighted in yellow).

4. The authors investigate the STUbL RNF4, but ignore other STUbLs including the recently identified STUbL TOPORS (Liu et al. 2024 Nat Struct Mol Biol doi: 10.1038/s41594-024-01294-7). Please include knockdown of TOPORS and double knockdown of RNF4 and TOPORS in Figure 2C.

Answer: We thank the reviewer for bringing the recently identified STUbL to our attention. We silenced *TOPORS* along with *RNF4* and included the data in the **new Figures 2F** and **2G**. In this experiment, we reproduced the effect of *RNF4* KD on *SMARCE1* abundance that we originally reported in the initial manuscript. In the YAMATO cells, we found silencing *TOPORS* in addition to *RNF4* resulted in substantially greater *SMARCE1* expression, than *RNF4* alone. It seems that also in our case the two STUbLs can act in coordination with each other to complete the SUMO-ubiquitin crosstalk observed in other cases (*Liu et al. 2024 Nat Struct Mol Biol doi: 10.1038/s41594-024-01294-7*). The suggested mechanism of SUMO-SMARCE1 ubiquitination is now depicted in **new Fig. 2H**. We have added appropriate text and also mentioned these data in the discussion: “**Conversely, STUbLs like the SUMO-dependent E3 ubiquitin ligases *RNF4* and *TOPORS* can degrade SUMOylated proteins leading to the depletion of their presence. Indeed, in our study we found that SUMO-dependent *SMARCE1* depletion was the outcome of the coordinated action of *RNF4* and *TOPORS*.**”

For reviewer only Figure 1. SS cell lines were treated with the indicated concentrations of ifosfamide and TAK-981, as well as their combination overnight and expression levels of SUMO-2/3, SMARCE1, ARID1A, BAF47 and TBP (nuclear loading control) were detected by western blotting.

levels.

5. Does the drug ifosfamide alter sumoylation and degradation of the cBAF complex? Please address by SUMO pull down- Western approach.

Answer: As per the reviewer’s request, we treated three different SS cell lines (HS-SY-II, SYO.1 and YAMATO) with ifosfamide, TAK-981, and their combination, and compared these conditions to a no drug (No Rx) control (**for reviewer only Figure 1**). Although SMARCE1 and BAF47 were elevated in two out of the three cell lines (moderate increase in HS-SY-II and slight increase in SYO.1) following ifosfamide treatment, and ARID1A slightly upregulated in HS-SY-II cells, the SUMOylation status of the cells was not significantly altered (there was a small enhancement in HS-SY-II and YAMATO cells), indicating the observed SMARCE1 upregulation following ifosfamide treatment. These data demonstrate ifosfamide does not decrease SUMOylation

6. The authors need to verify *RNF4* knockdown by immunoblotting.

Answer: We apologize for not including the RNF4 levels in the old Figure 2C. Since we continued to face difficulties detecting the protein levels of RNF4 and TOPORS in nuclear extracts by western blotting, despite trying several antibodies, we evaluated the mRNA expression of *RNF4* and *TOPORS* in **new Figure 2F (right)** and **2G (right)**, by qRT-PCR, verifying *RNF4* and *TOPORS* were silenced by siRNA in these experiments.

7. *The authors need to submit the mass spectrometry data to the PRIDE repository and provide access.*

Answer: Following the reviewer's advice, we have now submitted the mass spectrometry data to PRIDE repository, providing also access to it.

Reviewer access details:

Project accession: PXD059936

Token: R65Awr9ZdeLc

We have added the related information to the Materials and Methods:

"The mass spectrometry proteomics data have been deposited to the ProteomeXchange Consortium via the PRIDE¹ partner repository with the dataset identifier PXD059936."

We thank the reviewer for all their suggestions, there is no doubt they have helped improve the manuscript.

Reviewer #2 – Synovial sarcoma

In this manuscript titled "Targeting of SUMOylation leads to cBAF complex stabilization and disruption of the SS18::SSX transcriptome in Synovial Sarcoma", the authors (Floros et al.) thrive to demonstrate the following observations:

- 1) *The synovial sarcoma (SS) fusion SS18-SSX activates the sumoylation pathway as part of its oncogenic program.*
- 2) *Inhibiting sumoylation with the TAK-981 compound suppresses SS18-SSX programs and SS cells and tumor growth.*
- 3) *Part of TAK-981 mechanism is desumoylation and thereby stabilization of the BAF component SMARCE1, which restores normal cBAF complex on chromatin, thus destabilizing the oncogenic ncBAF complex formed by SS18-SSX in the sarcoma cells.*
- 4) *SS cell and tumor death by TAK-981 could be the result of DNA damage caused by the drug.*
- 5) *TAK-981 synergizes with SS standard therapeutic agents and therefore is an appropriate option for SS combination therapy.*

To support their findings, the authors generated 7 CHIP-seq screens in 2 SS cell lines, 2

ATAC-seq analyses, 4 RNA-seq studies, and utilized 6 different SS cell lines, a xenograft, a PDX, and a genetic SS tumor models.

This is an extensive multidisciplinary study that presents an alternative therapy for an up to now incurable malignancy. The authors convincingly demonstrate an involvement of SS18-SSX in protein sumoylation, and their synergy and in vivo modeling data are strong. However, the biochemical evidence for direct SMARCE1 sumoylation by SS18-SSX and desumoylation by TAK-981, and the important role it appears to play in shifting BAF complexes need to be strengthened.

General concerns:

- Add quantification of relative changes in western blots: Figs. 1C, S2A, S2C, 3A, 8I

Answer: As per the reviewer's request quantification has been added to almost all western blots across the manuscript (Figs. 1C, 1H, 2B, 2C, 2D, 2E, 2F, 2G, 3A (graphically depicted in new Fig. 3B), Fig. 3D and 3E (Sup. Table 2 and graphically depicted in new Fig. 3F, 3G, and S2E), Fig. 4A (graphically depicted in new Fig. 4B, 4C, 4D and S2F), Figs. 9I, S2A, S2C, S2D and S7F).

- Add statistics to all relevant graphs and include details in the legends, not in the Methods section, or preferably convert to histograms or dot plots or box plots with individual data points: Figs. 1D, 1F, S3J, S3L, 6A, 7E, 8K

Answer: We have now added statistics to all graphs throughout the manuscript and included the details of such in the figure legends.

- Manuscript needs a detailed Statistics section

Answer: A detailed Statistics section has been added to the Materials and Methods ("statistical considerations").

- Please make sure that all the panel labels correspond, in the text, figure legends and figures.

Answer: We have double checked, making sure that all panel labels correctly correspond to the text and to the figures/figure legends throughout the whole manuscript, as suggested by the reviewer.

Major concerns:

- Fig.S2A: discuss the apparent reduction in SS18-SSX levels by TAK-981 in the

inducible Yamato cells. Is there a feedback loop between SS18-SSX expression and sumoylation?

Answer: As correctly pointed out by the reviewer, there is a decrease in the abundance of SS18::SSX after treatment with TAK-981 in YAMATO cells (Fig. S2A). This can indeed happen due to a potential feedback loop in order for the cells to restore their SUMOylation status, by rebalancing the SS18::SSX levels. We have added this possibility in the results of the manuscript: "A decrease in the abundance of SS18::SSX after treatment with TAK-981 was also noted (Fig. S2A), likely a result of a feedback loop attempting to restore cellular SUMOylation levels."

- Fig. 1E: explain why crystal violet staining was chosen over a quantifiable cell viability assay in the CRISPR experiment.

Answer: Despite not being quantitative, crystal violet has been our preferred method to visualize viable cells following a drug or genetic perturbation, in particular when we are looking for a large drug effect or severe genetic perturbation effect (and not, for instance, an IC50 or EC50). Indeed, we have used this assay for this purpose across different studies¹⁻⁴). To clarify this, we have added in the text, "As a simple and reliable means to detect a potential effect by CRISPR/CAS9 perturbation, we evaluated cells by crystal violet staining".

- Fig. 1E: discuss why SMARCE1 levels and SS18-SSX levels in SS cells were not measured upon depletion of SUMO components by CRISPR.

Answer: We appreciate the comment. We considered that the expression of both BAF complex subunits following SUMO inhibition has been exhaustively assessed throughout the entire manuscript. More specifically, SMARCE1 expression has been evaluated *in vitro*, in 12 different western blots, in Fig. 2B, 2C, 2D, 2E, 2F, 2G, 3A (quantified in Fig. 3B) and measured also *in vivo* in Fig. 8E and 8I by IHC staining, after blocking SUMOylation with the selective SUMO inhibitor TAK-981. In all these cases its levels were clearly elevated, in line with our hypothesis. Similarly, SS18::SSX expression levels have been determined in Fig. 3A, 3D and 3E following TAK-981 treatment, showing no significant changes. We thank the reviewer for giving us the opportunity to further explain our rationale.

- Fig. 2B: It is important to confirm that the increase in SMARCE1 levels is post-transcriptional with qRT-PCR in the TAK-981 treated cells.

Answer: We have added quantifications of the differences of SMARCE1 levels (among other proteins) by western blot (new Fig. 3A and 3B) and have performed qRT-PCR (new Fig. 3C) to evaluate the impact of TAK-981 on SMARCE1 over time in four SS cell lines (HS-SY-II, YAMATO, SYO-1 and FUJI). The changes at the mRNA level were

minimal and not consistent with the increased abundance assessed by western, consistent with SMARCE1 changes occurring at the post-translational levels and not at the mRNA levels.

- *Fig.2C: the “sumo-SMARCE1” band is weak and inconclusive. It could just be a non-specific background band.*

After identification of SMARCE1 in the PTMscan, the next thing would be to verify whether SMARCE1 is indeed sumoylated in SS. This could be shown with more than one method, for example:

- *IP/western: IP with sumo2 and or sumo3 Ab, then blot with SMARCE1 Ab, using very sensitive ECL reagents such as super signal ATTO - and/or the reverse with IP SMARCE1 and blot for SUMO*

- *if endogenous levels are still undetectable, then accumulation of SUMO species with a reagent such as TAK-243, MG132, or ALLN, etc. might enhance SUMO-SMARCE1 detection.*

- *For general significance, it is advisable to test for SMARCE1 sumoylation in all 6 SS cell lines.*

- *one could also express exogenous sumo and test for SMARCE1 modification etc.*

Answer: As described in detail above, in reviewer 1 response, we have now performed the requested pull-down western assays, after transiently overexpressing SUMO-2 in HS-SY-II and Yamato cells. We immunoprecipitated the nuclear lysates with both, a SUMO-2/3 (new Fig. 2B and 2D) and a SMARCE1 mAb (new Fig. 2C and 2E, and please see also a detailed answer to comment #2 of Reviewer #1), The results verified the SUMOylation of the SMARCE1 protein.

- *Fig.3A: the authors assume that, based on previous studies, increased SMARCE1 is the cause of enhanced ARID1A and BAF47 expression.*

However, it is essential to rule out that, like SMARCE1, this upregulation of ARID1A and BAF47 is due to desumoylation by TAK-981 as well, even though they were not identified in the PTMscan. This could be tested in the IP/western described above.

Answer: We thank the reviewer for prompting us to detect the potential SUMO modifications of ARID1A and BAF47 as well. We have included expression of both proteins in the SUMO-2/3 IP performed in two SS cell lines (new Figure 2B and 2D).

Please see also answer to comment # 2 of Reviewer #1.

- *Fig.3B, 3C, 3D: adequate glycerol gradients and salt extractions. Was SS18-SSX left out for a reason?*

Answer: We apologize for not having included the SS18::SSX expression in the original manuscript. We have now added its expression to all sedimentation and salt extraction assays (new Fig. 3D, 3E, 4A and parallel quantification in new Fig. S2E and S2F). Even

though no change of SS18::SSX was detected in the salt extracts after blocking SUMOylation, a slight increase of SS18::SSX has been noted in the sedimentation assay of the SYO-1 cells (new Fig. 3D) at fractions 17, 18 and 19 (that correspond to the eluted gradients of the cBAF complex), due to the stabilization of the cBAF complex. Similar changes were also noticed in the specific gradients after TAK-243 treatment, in line with our hypothesis of the ubiquitin dependent degradation of SUMO-SMARCE1. The new findings have been added to the results (highlighted in yellow): “In addition, TAK-981 induced a modest, yet clear, increase of SS18::SSX abundance in the cBAF corresponding gradients (Fig. 3D and S2E), in line with cBAF stabilization.”

- *Optional: it would add to the findings if co/IP experiments were conducted to test whether any changes in the composition of the cBAF, ncBAF, and PBAF complexes had occurred in the TAK-981 treated SS cells.*

Answer: We have now conducted SMARCE1 pull downs in SUMO-2 overexpressed nuclear lysates of HS-SY-II and Yamato cells treated with TAK-981 (new Fig. 2C and 2E). Our data revealed a clear enhancement in the interaction between SMARCE1 and ARID1A and SMARCE1 and BAF47, consistent with cBAF stabilization, following TAK-981 treatment. In contrast, there was not an increase (if anything a slight decrease) in the interaction between SMARCE1 and PBRM1 (unique member of the PBAF complex), verifying the data of a previous study that demonstrated SMARCE1 can stabilize cBAF complexes but not PBAF complexes⁵.

Minor concerns:

- *Methods: why are Miapaca-2, Panc-1, RPE.1, and SW982 cells included? Why is the XP antibody listed?*

Answer: We apologize for mistakenly including those cell lines in the methods section. They have now been removed from the Materials and Methods. Regarding the “XP” antibody, this is part of the name of the specific SS18::SSX antibody we used in this study. Its full name is: **SS18::SSX (E9X9V) XP** (Cat #72364; from Cell Signaling).

- *Introduction-2d paragraph, unclear first sentence. Suggest replacing with: “The pathognomonic chromosomal rearrangement in synovial sarcoma fuses the DNA segment coding for the C-terminal 78 amino acids of SSX1, SSX2, or SSX4, all located on the X Chromosome, to the region coding for amino acids 1–379 of the SS18 gene on chromosome 18, to generate SS18::SSX1, SS18::SSX2, or SS18::SSX4 fusion, each capable of driving SS (Clark et al., 1994)”.*

Answer: We have now replaced the old sentence with reviewer’s suggestion (highlighted in yellow). We appreciate the improvement.

- Introduction-3d paragraph- Suggest changing to: "The major BAF complexes in most cells include canonical BAF (cBAF, also referred to as BAF), polybromo-associated BAF (PBAF), and non-canonical or GLTSCR1-containing BAF (ncBAF or GBAF).

Answer: We thank the reviewer for their comment. We have added the improved sentence to the introduction (highlighted in yellow).

- Supplemental Fig.1 Legend: it is "this analysis" or "these analyses". Also, the Schema is panel F)

Answer: We appreciate the comment, and we have replaced "this analyses" with "this analysis" (highlighted in yellow). The number of the panel of the Schema has also been corrected (to "F", highlighted also in yellow).

- Results: The binding of BAF complexes to chromatin is responsible for many of the observed expression changes of the SS transcriptome after blocking SUMOylation
This section text and figures are hard to follow, it would gain from clearer writing and better organization of the results description and figure layout in a systemic fashion. it should be specified at the beginning that the SMARCA4 CHIP-seq was conducted in HS-SY-II cells and the SS18-SSX, KDM2B, and H3K27Ac CHIP-seqs in HS-SY-II and SYO-1 cells etc. Also, the use of imported RNA-seq data and the ones generated by the authors should be clearly explained and sorted.

Answer: While we found it difficult to rearrange the figure panels too much (Figs. 6, S4, and S5), we have now specified at the beginning of the section "*that the SMARCA4 CHIP-seq was conducted in HS-SY-II cells and the SS18-SSX, KDM2B, and H3K27Ac CHIP-seqs in HS-SY-II and SYO-1 cells etc.*" (highlighted in yellow) and clarified the source of the RNA seq data (performed for the current study or imported from Banito et al.⁴) within the text of the specific section and within the legends of Fig. 6, 9, S4, S5 and Fig. S6 (highlighted also in yellow). Additionally, the function of KDM2B in SS has been described at the beginning of the previous section ("*Blocking SUMOylation leads to disruption of the synovial sarcoma signature and to induction of mesenchymal differentiation*") (highlighted in yellow), all in efforts to make the text easier to follow. Lastly, we identified and improved some suboptimal sentence structure to increase readability. We thank the reviewer for pointing this out and agree these changes were necessary to make this section less dense and more readable.

In the same section, is it meant to be: "Additionally, 49 out of the 155 genes that had enhanced SMARCA4 signals also had enhanced SS18-SSX binding?"

Answer: The sentence has been corrected and replaced (highlighted in yellow). We appreciate the reviewer's keen eye.

- Results: TAK-981 has activity in synovial sarcoma in vitro and in vivo by inducing DNA damage

Fig.S7F in the text is S7C

Fig. S7C, S7D and S7E in the text are Fig. S7D, S7E, and S7F The corresponding legend needs same corrections

Answer: We have gone through all the figures and the figure legends of our study and corrected all the mistakes that had been made.

- Fig.S9B: the comet assay result needs some quantification and statistics. How many tumors were looked at?

Answer: As per the reviewer's request, we have now quantified the length of the tails of the comet assay performed in ASKA and HS-SY-II cells and have added statistics (**new Fig. S9B** and **S9C**). The quantification of the data has confirmed our initial visual evaluation of the images.

- Include reference for the hSS2 mice in the text.

Answer: Following the reviewer's advice, we have now added a reference for the specific mouse model (highlighted in yellow).

We thank the reviewer for helping us critically address several aspects of the study and in turn strengthen the manuscript.

Reviewer #3 - Chromatin remodeling (BAF)

Comments to the Authors

In this study, the authors aim to define how SS18-SSX fusion protein can activate the SUMOylation program, rendering these tumors sensitive to a small molecules SAE1/2 inhibitor, TAK-981. They show that TAK-981 deSUMOylates SMARCE1 that they claim then restabilizing cBAF and reduces ncBAF on chromatin. However, the major issue is that the work does not convince the reader that TAK-981 impact on SMARCE1 and cBAF (or any chromatin remodeler/regulator) is the cause of its mechanisms. Rather, it would appear clear that TAK-981 will impact MANY SUMOylation events on many cellular proteins and reduce viability of SS (and other cell) lines in this manner. It is also impossible to conclude from the data presented that besides overt inhibition of many pathways in highly proliferative cancer cells lines like SS that the mechanism is via cBAF/ncBAF as the authors claim and hypothesize, leading to significantly reduced enthusiasm for the study by this reviewer.

General Answer to this concern: We agree, and in fact have demonstrated in the manuscript (Figs. 5 and S3) that the expression of many genes/proteins are altered by an inhibitor of SUMOylation. Similarly, many genes/proteins are altered by genetic disruption of SS18::SSX in SS⁴, which should not be unexpected considering this is the primary and likely sole driver of SS.

While not direct evidence, our substantial data (Figs 5, 6, 7, S3, S4 and S6) - demonstrating the overlap between expression changes following SS18::SSX genetic disruption and TAK-981- combined with the known phenotypic consequence of SS18::SSX genetic disruption (Fig. 1B, 1C and 1D), strongly points to the redistribution of BAF complexes as the major mechanism of toxicity of TAK-981 in SS.

While of course this doesn't rule out additional contributing factors to TAK-981 toxicity in SS, which there very likely are, this does not take away from the central role of the mechanism we demonstrate and the thesis of this manuscript. To make this point clearer, we have added to the discussion "Furthermore, while our data strongly points to the redistribution of BAF complexes as the major mechanism of TAK-981 toxicity in SS, we cannot rule out other changes following deSUMOylation as contributing to toxicity" (highlighted in yellow).

Major comments:

Knockdown of the SS18-SSX fusion, in many other studies, completely reorganizes activated and repressed regions of chromatin, by mechanisms that have been well described in the literature (Kadoch lab many paper and Ana Banito lab, Kevin Jones lab, others).

Answer: Yes, we agree that it has been well documented that knockdown of SS18::SSX results in significant changes in the SS phenotype. Indeed, two out of the three researchers cited by the reviewer (Ana Banito and Kevin Jones) are co-authors on the current manuscript and as summarized above, we believe this well-demonstrated phenomenon is completely consistent with the thesis of this paper.

Many genes, including those involved in protein degradation, are both controlled by SS18-SSX on chromatin and enable protein level control. Crabtree and Kadoch even show in 2013 (Kadoch, Crabtree Cell 2013) that fixing cBAF such as by forced overexpression of BAF47 is not able to reduce or combat the effects of SS18-SSX because of likely affinity changes that I noted are also shown at biochemical and biophysical level in McBride et al NSMB 2020 paper which the authors cite.

Answer: We thank the reviewer for these comments that prompted us to be more accurate and clarify the manuscript. More specifically:

1) The critical structural role of SMARCE1, in the sustainability of the cBAF complex by affecting the connectivity of the subunits of its core module has already been reported (St. Pierre R. et al., *SMARCE1 deficiency generates a targetable mSWI/SNF dependency in clear cell meningioma*, *Nature Genetics*, 2022). “The impact of SMARCE1 on the stabilization of the cBAF complex does not depend on any effect of SS18::SSX, indicating that the presence of the fusion protein is not always necessary for any changes induced, that result in the disruption of the SS signature.”

2) We have demonstrated previously that “**SMARCB1 loss, by itself or in addition to the expression of the SS18::SSX fusion protein**, affects BAF family complexes by reducing general BAF family affinity for, and distribution across, chromatin. In addition, this alters BAF family distribution to promoters and TSSs specifically” (Li J. et al., *A Role for SMARCB1 in Synovial Sarcomagenesis Reveals That SS18::SSX Induces Canonical BAF Destruction*, *Cancer Discovery*, 2021). Therefore, while overexpressing SS18::SSX results in reduced presence of BAF47^{6,7}, its expression is sufficient but not necessary for the regulation of BAF47 levels and the stabilization of the cBAF complex.

In Figure 2, which is also very sparse, the authors identify SMARCE1 but don't show blots in these experiments for HS-SY-II or Yamato cell lines for any other cBAF or ncBAF subunit component. Where are the protein levels for all the other subunits in this experiment? This makes it very challenging for the reader to interpret; just because the mass spectrometric shows one subunit, it does not mean that that is the only subunit potentially having impact. Also, a time course set of experiments with TAK-981 for total protein as well as RNA of each BAF subunit would be needed to better interpret the data.

Answer: We apologize for not having included the abundance of the other BAF complex subunits in the original Figure 2. Following the reviewer's advice, we have now conducted a time course treatment of TAK-981 in four SS cell lines (HS-SY-II, SYO-1, YAMATO and FUJI), evaluating the protein levels as well as the mRNA levels of the SWI/SNF complex components (new Fig. 3A, 3B and 3C). The results show a significant increase in the cBAF members ARID1A, SMARCE1 and BAF47 over time, and only at the protein level, consistent with our hypothesis that blocking SUMOylation stabilizes the cBAF complex through upregulation of SMARCE1 (new Fig. 2B, 2C, 2D, 2E, 2F and 2G). We thank the reviewer for helping us improve the manuscript. **New Figure 2** now contains many BAF complex readouts and, along with the other 19 figures in the manuscript, we believe provides a robust collection of data supporting our thesis.

In the Figure 3, The authors show that several subunits are “stabilized” at the protein level. They would likely be stabilized in non-SS cells as well but these data are not shown (where are gradients and blots for many other cancer cell lines or normal cells? I anticipate that most proteins would increase in levels. One can see also that PBAF subunits (which have been shown to not be impacted by the fusion and do not contain the SS18-SSX fusion) also are majorly stabilized by TAK-981. Similarly, BRD9 of ncBAF

complex is stabilized and there looks to be even more of it with the TAK-981 treatment. Many proteins including probably chromatin repressive complexes would show all the same.

Answer: As per above, in the current study we demonstrate **that the unique cBAF component ARID1A, as well as specific subunits of the cBAF/PBAF complexes (SMARCE1 and BAF47) are being elevated in a time-dependent manner after treatment with TAK-981 in all four SS cell lines tested** (new Fig.3A and 3B). This increasing pattern was not detected in the unique ncBAF (BRD9) or PBAF (PBRM1) components in the HS-SY-II, YAMATO, or FUJI SS cell models. The exception among the four models is the SYO-1 cells, in which BRD9 and PBRM1 also show an upregulation, albeit one substantially lower in magnitude than the increases demonstrated in the cBAF components (please see quantification) (new Fig. 3A and 3B). Again, while there is a very modest increase of BRD9 in vivo in SYO.1 tumors, the increases in cBAF components are—in contrast—quite significant (Sup. Fig. 8I). Furthermore, there is no increase in GLTSCR1, another ncBAF unique member (Sup. Fig. 8I).

We would respectfully disagree with the reviewer's interpretation that the sedimentation or the salt extraction assay displays further stabilization of the GBAF (ncBAF) and the PBAF complex, or higher affinity of these complexes to the chromatin. In this study we demonstrate by glycerol gradients, that blocking SUMOylation stabilizes the cBAF complex (new Fig. 3A, 3B, 3D, 3E, 3F and 3G), which subsequently results in a significant and consistent upshift of the cBAF members SMARCE1, BAF47 and ARID1A, by approximately 3 fractions after TAK-981 treatment in SYO-1 cells and a higher expression of the same proteins in the HS-SY-II cells (new Fig.3D, 3E, 3F, 3G). On the contrary, there is no upshift, upregulation or stabilization of expression in the unique ncBAF member BRD9 or unique pBAF member PBRM1 in these experiments (new Fig. 3D, 3E).

Additionally, our salt extraction assays, that are being used to designate affinity to chromatin (like in other articles⁸), not only verify the increase in the abundance of the cBAF complex subunits, but also reveal a clear shift of the expression of BAF47, SMARCE1 and ARID1A to higher salt concentrations (indicating higher affinity to chromatin) in all three SS cell lines (with the only exception of BAF47 in YAMATO cells) **and a clear shift of the BRD9 bands to lower salt concentrations in two out of the three cell lines tested** (new Fig. 4A). We have now quantified the density of the bands of the different proteins separately in no drug treatment control (No Rx) and TAK-981 treatment cellular fractions and normalized to the expression of the strongest band within each blot, in order for conclusions not to simply (and erroneously) be based on just the change of the total protein levels of each BAF component before and after TAK-981 treatment. These data also supports our hypothesis (see new Fig. 4B, 4C and 4D and S2F). These additional measures have been added to figure legend 4A (highlighted in yellow). Regarding ARID2, a unique member of the PBAF complex (like PBRM1),

there is no consistency in its pattern of expression following TAK-981 treatment (new Fig. 4A and S2F). In all, we believe these new data support the hypothesis that the cBAF complex is selectively stabilized in SS among BAF complexes.

The panels in Fig 3D do not tell us anything except further that levels are increased of everything... if this is an attempt to show changed chromatin stability, this does not show it; this is just at the mercy of antibody detection levels/affinities and overall protein levels.

Answer: Please see our response to the previous comment where we detailed the analysis of this experiment as well as added quantification to assist interpretation.

The authors then look at gene expression in the Figure 4. Here again, the figures as presented do not help the case—clearly, TAK-981 does a lot of other things as does KD of fusion. If TAK-981 really impacted the complexes, one would expect this to look much more like SS18-SSX KD and likely other conditions like BRD9 knockout or knockdown for ncBAF.

Answer: We respectfully disagree with the reviewer that the changes caused by TAK-981 treatment in Figure 5 (old Fig. 4) “do not look like the SS18::SSX KD” changes. As discussed above, we agree that genetic disruption of SS18::SSX and blocking SUMOylation (by TAK-981) are not two identical processes. However, the genes and the pathways that are up- and down-regulated by TAK-981 in Figure 5, not only in the authors’ opinions, but by all statistical measures, remarkably overlap with the genes and the pathways that are up- and down-regulated following knocking down of SS18::SSX. Moreover, in Figure 5D, a highly statistically significant number of the 1000 most up- and down-regulated genes after TAK-981 treatment overlaps with the corresponding pathways after silencing SS18::SSX in SS (from co-author Banito’s original study, Banito et al.⁴) and further supports the significant overlap of changes. It should also be noted that this substantial overlap persists despite the KD experiment originating more than 6 years ago in a different laboratory (Banito et al, The SS18::SSX Oncoprotein Hijacks KDM2B-PRC1.1 to Drive Synovial Sarcoma, *Cancer Cell*, 2018) and persists when comparing KD in the HS-SY-II cells to TAK-981 treatment in distinct SS cells. These factors, in our minds, increase the impressiveness of the overlap of genes that are noted in the manuscript.

Additionally, all the assays we conducted in Fig. 5, 6, 7, 9, S3, S4, S5, S6, S7 and S10 (7 CHIP-seq screens in 2 SS cell lines, 2 ATAC-seq analyses, 4 RNA-seq studies), exhaustively substantiate that the most enhanced or suppressed biological processes are common to both TAK-981-treated SS cells and SS18::SSX-genetically inhibited SS cells, reported in Banito et al.⁴. Thus, these observations are not just the outcome of random changes that inhibition of SUMOylation generates in the complete proteome. Furthermore, these data confirm that “the binding of BAF complexes to chromatin is

significantly responsible for the observed expression changes of the SS transcriptome after blocking SUMOylation” (ChIP-seq data) and that “TAK-981 results in loss of chromatin accessibility to promoters of the SS18::SSX transcriptome” (ATAC-seq data), pointing, at the same time, to a highly selective mechanism of redistribution of the BAF complexes, that is triggered—at least partially—by the deSUMOylation of SMARCE1 and that is ultimately responsible for the disruption of the SS signature.

In Fig 5, the authors perform ChIP-seq but there is no comprehensive analysis of these data shown—no heatmaps, few volcano plots, few tracks over important genes, and it is not clear if indeed in their own data, there is really any alteration in cBAF and ncBAF binding to SUMOylation genes and SUMOylation target genes. Overlap in 5E is very limited and numbers of peaks are low altogether....

Answer: We respectfully disagree with the reviewer that our ChIP-seq analysis is not comprehensive. In the old Fig. 5 (now Fig. 6) we focused primarily on identifying a correlation between 1) the 100 most up- and down-regulated genes after knocking down SS18::SSX, reported in Banito. et al., 2) our SMARCA4 ChIP-seq studies and 3) our SS18::SSX ChIP-seq study. As such, adding a heatmap would not be informative the reader, nor would it be appropriate.

Instead of adding more ChIP-seq enrichment tracks for two or three randomly selected genes, to consider the totality of all the genes being altered and the consequences of such gene changes, we have performed comprehensive pathway analyses. Indeed, for all the ChIP-seq, ATAC-seq and RNA-seq experiments throughout the entire manuscript, we have now conducted in-depth analyses, including gene ontology/pathway analyses, and created Venn diagrams and volcano plots to visualize and test significance of the gene changes for each experiment. For instance, our manuscript contains 11 volcano plots in total with each offering important information that tracks with the experimental questions we are asking to test our hypothesis. For old Fig. 5E (now Fig. 6E), we have performed a Chi-square test under the assumption of 17,611 expressed transcripts. comparing significantly enhanced SS18::SSX and SMARCA4 ChIP seq signals in HS-SY-II cells, and the result is a **p-value < 4.712e-55, demonstrating a high level of significance.**

There are just as possible other mechanisms of degradation at play for SS18-SSX complexes, including very standard ones (i.e. even MG-132 has been show similar results), especially for proteins that cannot assemble into complexes because of fusion protein and what it structurally changes in complex. The fact that SMARCE1 and BAF47 levels are so much more markedly increased by the combination of TAK-981 and ifosfamide (and even ifosfamide alone!), i.e. in panel 7H, just further show that this is a global cell poisoning mechanism. If the authors had stained for BRD9 or BIRCA as ncBAF members, they would also likely show the same, indicating that ncBAF is not decreased and ncBAF-caused transcription is not the main mechanism that is

attenuated. Evaluation of other chromatin proteins and protein complexes would show similar, given these data. In fact, nowhere in this paper do the authors show that BRD9 is reduced in protein levels, and hence the authors conclusions that their TAK-981 “shifts away from SS18-SSX *ncBAF*-driven transcription” is just not well founded from the data.

Answer: 1) We agree that BRD9 and BICRA (aka GLTSCR1), specific members of the *ncBAF* complex, are not downregulated after treatment with TAK-981, and while their expression is not increased either, neither outcome is a part of our hypothesis.

In sharp contrast, the *cBAF* members ARID1A, BAF47 and SMARCE1 are consistently and markedly upregulated following TAK-981 treatment across diverse SS models (including *in vivo* in both human and mouse SS tumors) and across variable time points post TAK-981 treatment *in vitro*.

More importantly, and as pointed out above, while *cBAF* complex members clearly and repeatedly bind to chromatin with greater affinity following TAK-981

treatment across all our SS models tested, BRD9 does not, and, in fact, binds with lower affinity to chromatin after treatment (in two out of the three cell

For reviewer only Figure 2. HS-SY-II cells were treated with navitoclax (1µM, overnight treatment), radiation therapy (2Gy) or TAK-981 (100nM, 24h) and protein expression was evaluated in nuclear lysates by western blotting. TBP was used as a nuclear loading control.

lines, please see **new Fig. 4A and new Fig. S2G**).

For reviewer only Figure 3. TAK-981 treatment rebalances the relative distribution of the BAF complexes from a SS phenotype to a more normal cell phenotype.

only Fig. 2, two modes of

2) To address the reviewer’s concern, we have now stained the excised tumors from the ifosfamide/TAK-981 *in vivo* model (old Fig. 7) for BRD9 and GLTSCR1. In both cases, their expression didn’t exhibit significant changes and remained basically unaltered after ifosfamide, TAK-981 and the combination treatment (**new Fig. 8H, 8I**).

3) Regarding the comment about a global cell poisoning mechanism, it would seem, if this were true, other means to kill SS cells would include the mechanism of stabilized *cBAF* complexes. As demonstrated in **for reviewer**

killing SS cells, treatment with navitoclax⁹, and radiation (part of standard of care for SS), do not upregulate cBAF complex members (nor other BAF complexes). Cell death is confirmed in these experiments by the appearance of cleaved PARP.

4) Overall, we think the reviewer may not be fully appreciating the main hypothesis of the paper, and we apologize for not being clearer. To state it again, in order for the balance to “shift away from a more ncBAF dominant complex distribution”, which defines the SS phenotype, there does not need to be *both* an increase of cBAF complexes and a simultaneous reduction of GBAF (ncBAF) complexes. We are referring to the **relative proportions of the complexes** that change, following their redistribution (depicted in **for reviewer only Fig. 3**).

There is a difference of “ncBAF transcriptome” and “genes that are on in cancer cells” and nowhere in here do the authors specifically show any changes to ncBAF on chromatin or genes defined to be altered by BRD9 degradation or KD/KO.

Answer: Please see our detailed answer to this concern above.

If the authors were to ever convince that SUMOylation is a major factor here for these remodeling complexes then a screen would be needed (i.e. to treat cells, synovial sarcoma cells and others including normal cells without fusion) with TAK-981 and identify genes needed for TAK-981-induced killing.

Answer: While we do appreciate that the reviewer upon the first reading of the original manuscript was not convinced that deSUMOylation is leading to a redistribution of these complexes, we do hope our new data and our explanations above to the reviewer’s concern has mitigated the issue.

A second point that may be helpful to repeat, is, as we demonstrate, SS18::SSX increases SUMOylation activity in SS. The fact that the definitively driving oncogenic event in SS (SS18::SSX) directly upregulates the SUMOylation program, serves to the authors, at least, as a major indication that this is a vital and essential biological process for SS survival, and it thus follows that it would impact the most vital and essential gene expression program in SS. Hence, why SS18::SSX would “hijack” the process of SUMOylation: To provide a survival advantage through the pathway that is most important for its survival (by degrading cBAF complexes).

While performing a screen is outside the scope of this paper, it may be helpful to remind the reviewer here that SS18::SSX-containing ncBAF complexes are SS specific. Therefore, even if TAK-981 led to cBAF complex stability in other cancers (or normal cells) outside of SS, and a subsequent rebalance of BAF complexes, it is presumptuous and likely untrue that this process would be toxic to other cancers (if in fact TAK-981 is toxic at all to these other cancers). On the other side of this, how TAK-981 kills other cancer cell lines would be expected to be different than how TAK-981 kills SS cells.

Despite the points above and the data throughout the manuscript, we do agree with the reviewer's point that other mechanisms likely contribute to the total toxicity of TAK-981 in SS and have clarified and expanded on this in the discussion (yellow highlighted sections). It should be noted, that, just like in the case of most targeted therapies, there are still many genes that modify their killing (e.g. as reviewed in Hata et al. *Cancer Discovery* 2015¹⁰ or demonstrated in Crystal et al. *Science* 2014¹¹ and Faber et al. *Cancer Discovery* 2011¹²). However, that certainly does not de-value the understanding that, for instance, EGFR inhibitors inhibit mutant *EGFR* (i.e. the major target in this paradigm). Thus, we believe understanding the targets that are major contributing factors, in this instance a SMARCE1-contributing BAF complex rebalance caused by TAK-981 in SS, contains important merit on its own.

We thank the reviewer for their patience in reading through this rebuttal and do hope that the combination of new data and our improved explanations fostered more agreement.

- 1 Tasdemir, N. *et al.* BRD4 Connects Enhancer Remodeling to Senescence Immune Surveillance. *Cancer Discov* **6**, 612-629 (2016). <https://doi.org/10.1158/2159-8290.Cd-16-0217>
- 2 Aarts, M. *et al.* Coupling shRNA screens with single-cell RNA-seq identifies a dual role for mTOR in reprogramming-induced senescence. *Genes Dev* **31**, 2085-2098 (2017). <https://doi.org/10.1101/gad.297796.117>
- 3 Tordella, L. *et al.* SWI/SNF regulates a transcriptional program that induces senescence to prevent liver cancer. *Genes Dev* **30**, 2187-2198 (2016). <https://doi.org/10.1101/gad.286112.116>
- 4 Banito, A. *et al.* The SS18-SSX Oncoprotein Hijacks KDM2B-PRC1.1 to Drive Synovial Sarcoma. *Cancer Cell* **33**, 527-541.e528 (2018). <https://doi.org/10.1016/j.ccell.2018.01.018>
- 5 St Pierre, R. *et al.* SMARCE1 deficiency generates a targetable mSWI/SNF dependency in clear cell meningioma. *Nat Genet* **54**, 861-873 (2022). <https://doi.org/10.1038/s41588-022-01077-0>
- 6 Kadoch, C. & Crabtree, G. R. Reversible disruption of mSWI/SNF (BAF) complexes by the SS18-SSX oncogenic fusion in synovial sarcoma. *Cell* **153**, 71-85 (2013). <https://doi.org/10.1016/j.cell.2013.02.036>
- 7 McBride, M. J. *et al.* The SS18-SSX Fusion Oncoprotein Hijacks BAF Complex Targeting and Function to Drive Synovial Sarcoma. *Cancer Cell* **33**, 1128-1141.e1127 (2018). <https://doi.org/10.1016/j.ccell.2018.05.002>
- 8 McBride, M. J. *et al.* The nucleosome acidic patch and H2A ubiquitination underlie mSWI/SNF recruitment in synovial sarcoma. *Nat Struct Mol Biol* **27**, 836-845 (2020). <https://doi.org/10.1038/s41594-020-0466-9>
- 9 Jones, K. B. *et al.* SS18-SSX2 and the mitochondrial apoptosis pathway in mouse and human synovial sarcomas. *Oncogene* **32**, 2365-2371, 2375.e2361-2365 (2013). <https://doi.org/10.1038/onc.2012.247>
- 10 Hata, A. N., Engelman, J. A. & Faber, A. C. The BCL2 Family: Key Mediators of the Apoptotic Response to Targeted Anticancer Therapeutics. *Cancer Discov* **5**, 475-487 (2015). <https://doi.org/10.1158/2159-8290.Cd-15-0011>

- 11 Crystal, A. S. *et al.* Patient-derived models of acquired resistance can identify effective drug combinations for cancer. *Science* **346**, 1480-1486 (2014). <https://doi.org/10.1126/science.1254721>
- 12 Faber, A. C. *et al.* BIM expression in treatment-naive cancers predicts responsiveness to kinase inhibitors. *Cancer Discov* **1**, 352-365 (2011). <https://doi.org/10.1158/2159-8290.Cd-11-0106>

Overview: We have added 11 more figure panels: Fig. 2F, 2H, 2K, 7E, 7F, Fig. S2G, S6C, S6D, S6E, S6F and S6G.

All individual data points have been added to the graphs of Fig. 1E, 1G, 2G, 3C, S2G, S3J and S3L.

Additionally, the uncropped versions of all western blots of the manuscript have been included in a single PowerPoint file (“unprocessed western blots”).

Please find below our response to #4 reviewer’s comments:

The fundamental premise of the work originated from interrogation of the genome scale shRNA knockdown dependency scores generated from the DepMap portal which suggested the sumoylation pathway is a top dependency in SS. However, the authors failed to consider that key components of the sumoylation pathway including SAE1 (the sumo activating enzyme and target of TAK-981), UBE2I and SUMO2 are common essential genes as documented by DepMap CRISPR knockout dependency study. The study has not addressed the fact that the SUMO pathway, and more relevant for the present study the target of the compound the study is almost completely dependent on, is an essential gene with pleiotropic effects.

Answer:

We appreciate echoing this concern of reviewer 3 and believe we have exhaustively explained the issue of the potential toxicity in the previous rebuttal letter and would reference that here again. To the newer comment, it should be clarified that SAE is a heterodimeric protein of SAE1 and SAE2 (UBA2); thus, they are *both* targets of TAK-981. Of course, we acknowledge that some components of the SUMOylation pathway are common essential. Of course, there are large degrees of variability to the toxicity of genes in different cell lines that are deemed common essential (and as Broad is upfront, the definition itself is empirical but arbitrary). Indeed, both the redundancy to toxicity to SUMO pathway genes *and* the degree to which targeting these SUMO genes are toxic to SS, is what makes it a uniquely sensitive cancer (for example, Fig. 1A, S1). Lastly, in terms of translation, essential genes do not translate to drugs with poor therapeutic windows. An example is MEK1 and MEK2. MEK1/2 inhibitors (trametinib, cobimetinib, and binimetinib) are FDA-approved and only have efficacy (i.e. a therapeutic window) in BRAF mutant cancers and NF-1 deleted Neurofibromatosis Type 1 due to the underlying biology. We believe SAE1/2 inhibitors similarly could be one day uniquely FDA-approved in SS due to the underlying biology. Of course, there are other examples of this (CDK4/6 inhibitors) but we believe MEK inhibitors serve as a good parallel to understand why common essential in itself does not really mean anything in terms of drug development.

Additionally, as we have already mentioned in the first rebuttal letter:

“The fact that the definitively driving oncogenic event in SS (SS18::SSX) directly upregulates the SUMOylation program, serves to the authors, at least, as a major indication that this is a vital and essential biological process for SS survival, and it thus follows that it would impact the most vital and essential gene expression program in SS. Hence, why SS18::SSX would “hijack” the process of SUMOylation: To provide a survival advantage through the pathway that is most important for its survival (by degrading cBAF complexes).”

While the biochemical data regarding sumoylation of SMARCE1 is solid and convincing, the authors have not entertained other proteins in their sumoylation proteomic study such as HSP90, tubulin (table S1) that are more likely to underlie the cytotoxic phenotype of TAK-981. In fact, these targets are also most likely to explain the essentiality of the sumoylation pathway and SAE1 across the DepMap cell lines. The absence of a strong rescue experiment using SMARCE1 overexpression, knockout and complementation or similar assays are additional weak points. Taken together, the study does not convincingly show that the deSUMOylation of SMARCE1 is the primary mechanism of cytotoxicity of TAK-981.

Answer:

Obviously, some of the other targets likely do contribute to toxicity, as we have mentioned in the Discussion. However, we believe we have demonstrated pretty well that the primary mode toxicity is through inhibition of SS18::SSX-ncBAF / re-expression of cBAF complexes.

As we mentioned in the Discussion:

- 1) “Consistent with increased SMARCE1 expression ultimately resulting in TAK-981 toxicity in SS, our attempts to reconstitute wild-type SMARCE1 and SMARCE1 mutated at the sites of SUMOylation (K92 and K277) either constitutively or through an induced system resulted in significant toxicity that precluded performing functional assays.”

Although we would have liked to experimentally demonstrate SMARCE1 expression recapitulates TAK-981 toxicity, this supports the hypothesis *perhaps even better*.

- 2) We know from extensive figures throughout this paper, that the changes TAK-981 induces are consistent with the changes caused by SS18::SSX genetic

disruption. There is no doubt that SS18::SSX is the driving force in SS. So, this is also strong circumstantial evidence.

- 3) However, to further support our hypothesis, we interrogated Depmap of genome-wide RNAi screens, and, impressively, noted SS is the most sensitive cancer to RNF4 silencing among 30 subtypes of cancer (Fig. 2J and discussion). That is, further evidence that SMARCE1 stabilization, by blocking RNF4-mediated degradation of SMARCE1, is a significant contributor to TAK-981-mediated toxicity in SS.

However, RNF4 silencing is particularly toxic to SS cells compared to 30 other cancer subtypes, adding further evidence that increased SMARCE1 expression is the linchpin of the exquisite toxicity of TAK-981 in SS (yellow highlighted since it has been added to the discussion).

- 4) Additionally, we have performed the requested RNF4/TOPORS toxicity experiments. Again, in these data, we demonstrate toxicity caused by knockdown of RNF4/TOPORS across SS cell lines (Fig. 2K, S2G):

Altogether, while we of course cannot rule out the contribution of other genes, which is almost certain, we can conclude that the major toxicity, the reason for the exquisite sensitivity of SS to TAK-981, is due to the upregulation of SMARCE1 and subsequent induction of cBAF complexes and relative loss of ncBAF complexes.

Other concerns

RNF4 and TOPORS have been suggested to mediate sumo dependent ubiquitination and their knockdown was as effective as TAK-981 in stabilizing SMARCE1. Does this knockdown phenocopy TAK-981 in its cytotoxicity?

Answer:

As mentioned above, this is seen in new figures 2J, 2K, S2G.

RNF4/TOPORS KD is indeed toxic to the SS cell lines (HS-SY-II, Yamato, SYO-1) (evidenced by CV) in the 3 new panels.

We thank the reviewer for this comment as we believe these data have strengthened the MS.

Fig 3E does not convincingly show TAK-981 administration with or without shSSX shifts the BAF complex at all.

Answer:

We apologize that the reviewer may have missed some of the description of the data: In the manuscript we mention, regarding 3E:

“In the control cells (shREN knockdown), we noted a significant accumulation of ARID1A, SMARCE1 and BAF47 in the cBAF corresponding fractions following TAK-981 treatment. This is consistent with cBAF stabilization following SUMOylation inhibition. Expectedly, shSSX knockdown cells demonstrated enhanced expression of these cBAF proteins compared to control shREN knock down cells (Fig. 3E and 3G).”

We do not report nor claim “shifting” of the BAF complexes.

It should be noted:

- 1) In order to reliably compare the bands of the gradients, we ran all 8 gels simultaneously under the same conditions and exposed all 8 blots for each protein at the same time.
- 2) As mentioned above, the changes observed in the TAK-981-treated cells (HS-SY-II) recapitulate the changes observed in the SSX KD gradients (Fig. 3E).

Fig 4A, while the authors state that only components of the BAF complex show alteration in salt extraction, ARID2 (PBAF subunit) clearly shows very similar behavior to the BAF components in SYO-1 cell line. In HS-SY-II and Yamato cell lines, the BAF complex shows inconsistent and unimpressive salt extraction change by TAK-981.

Answer:

We believe the reviewer may have missed some of the description of the data: Regarding ARID2, we have written:

“ARID2 (and consequently the PBAF complex) does not follow a consistent pattern of chromatin affinity across the three cell lines” (Fig. 4A and S2I).

Yes, it shows very similar behavior to the BAF components in SYO-1 but not a similar behavior at all in HS-SY-II and Yamato cells (please see Fig. 4A and S2I-quantification of the bands).

This is the reason we quantified the bands of the salt extraction assay (as described in the previous rebuttal letter) in order to better draw conclusions:

“We have now quantified the density of the bands of the different proteins separately in no drug treatment control (No Rx) and TAK-981 treatment cellular fractions and normalized to the expression of the strongest band within each blot, in order for conclusions not to simply (and erroneously) be based on just the change of the total protein levels of each BAF component before and after TAK-981 treatment.”

Regarding the cBAF complex proteins:

We have mentioned that in all three cell lines the salt extraction assay reveals a clear shift of the expression of BAF47, SMARCE1 and ARID1A to higher salt concentrations (indicating higher affinity to chromatin) in all three SS cell lines (with the only exception of BAF47 in YAMATO cells – as we pointed out it in the previous rebuttal letter).

Based on the data, we must respectfully disagree with the reviewer that: *“In HS-SY-II and Yamato cell lines, the BAF complex shows inconsistent and unimpressive salt extraction change by TAK-981”*,

For further evidence, please see the quantification of the WB bands and the corresponding graphs in Figures 4A, 4B, 4C, 4D and S2I.

Fig 7: the BAF complex and its association with enhancers has been extensively shown to be highly relevant for tumorigenesis. Thus, it would be highly desirable to plot average peak intensities and heatmaps centered around peaks in addition to TSS centered plots.

Answer:

Thank you for this insightful point. We have now added 7 new panels to address this point:

New Fig. 7E, 7F, S6C, S6D, S6E, S6F and S6G

We thank the reviewer for giving us the opportunity to strengthen the MS.

In addition to “plot average peak intensities and heatmaps centered around peaks in addition to TSS centered plots.” (Fig. S6C, S6D), where the results are consistent to Fig. 7C (centered to TSS), we have conducted further analysis of our ATAC-seq, RNA-seq and H3K27ac ChIP-seq data in both cell lines (new Fig. 7E, 7F, S6E, S6F and S6G).

Also, as we already mentioned in the Results:

“It has been previously reported that increased cBAF complex occupancy strongly correlates with enhancer activation. We hypothesized that most of the genes that increase in expression upon de-SUMOylation are targets of cBAF enhancers. That is, TAK-981 treatment increases gene expression that is primarily due to the cBAF stabilization. To do so, we investigated the correlation between the 1000 most upregulated genes following TAK-981 treatment (detected by our RNA-seq analysis), and the changes in the accessibility of the chromatin at these specific genes (ATAC-seq analysis) and in their H3K27ac ChIP-seq signals. The 1000 most upregulated genes following de-SUMOylation also demonstrated enhanced chromatin accessibility and H3K27ac levels in both the SYO-1 and HS-SY-II cells (Fig. 7E and 7F). Expectedly, the same analysis for the 1000 most downregulated genes revealed a less consistent pattern (Fig. S6E and S6F).”

We thank the reviewer for the comments and helping us strengthen the manuscript.